



# Central Asia's spatiotemporal glacier response ambiguity due to data inconsistencies and regional simplifications

Martina Barandun[1,2] and Eric Pohl[2]

[1]Institute of Earth Observation, Eurac research, Bolzano, Italy
[2]Department of Geosciences, University of Fribourg, Fribourg, Switzerland

**Correspondence:** Martina Barandun (martina.barandun@eurac.edu)

**Abstract.** Glacier evolution across Tien Shan and Pamir is heterogeneous in space and time. This heterogeneity is believed to be mainly driven by contrasting climatic settings and changing atmospheric conditions. However, a systematic and consistent region-wide analysis of the climatic and static morphological drivers remains limited to date. Meteorological reanalysis and remote sensing products, and novel approaches to derive region-wide annual mass balance time series, all provide the basis to investigate the drivers behind the observed heterogeneous glacier response. Here, we investigate the consistency of inter-
pretations derived from available datasets through correlation analyses between climatic and static drivers with mass balance estimates in Tien Shan and Pamir. Our results show that even supposedly similar datasets lead to different and partly contradicting assumptions on dominant drivers of mass balance variability. Only when considering all glaciers in the Pamir and Tien Shan together, we find a similar picture of dominant meteorological drivers over space. Using either existing mountain subdivisions or glacier subdivisions based on mass balance variability, no consistencies can be found. Within different mass
balance and meteorological datasets the results suggest very different drivers. This conclusion is even more prominent in the temporal correlation analysis where contradicting patterns of dominant drivers result from presumably similar meteorological datasets. Clear non-climatic drivers could not be identified. Even with newly available mass balance and meteorological data, a knowledge gap about the main mechanisms behind the heterogeneous glacier response in Central Asia remains. The results highlight that apparent but false consistencies across studies using a single dataset might largely relate to the chosen dataset
rather than to the processes or involved environmental variables. As long as no glaciological, meteorological, or hydrological *in situ* observation network provides data for direct calibration and validation of extensive datasets, we cannot predict a realistic improvement in our understanding of the changing cryosphere at regional scale for Tien Shan and Pamir.

## 1 Introduction

Glaciers across the Tien Shan and Pamir, part of High Mountain Asia, have been observed to change heterogeneously (e.g., Barandun et al., 2021; Shean et al., 2020; Brun et al., 2017; Miles et al., 2021). This behaviour is principally driven by their different mass balance sensitivities to the climate (Sakai and Fujita, 2017; Wang et al., 2019). The diverse glacier responses to climate change are thus not only a result of local topographic and glacier-specific morphological characteristics (Fujita and Nuimura, 2011; Brun et al., 2019), but also relate to sharp contrasts of the local climatological settings. Changes in the weather



patterns regarding seasonality and intensities of different meteorological variables, such as precipitation and air temperature
for the last two decades are characterized in Gerlitz et al. (2020). Gerlitz et al. (2019) showed that changing regional circulation
characteristics led to increased climate variability during the winter season and explained parts of the pronounced warming
in Central Asia. Mölg et al. (2014) and Farinotti et al. (2020) related a spatially heterogeneous glacier response for selected
mountain ranges to different weather pattern constellations. Dyurgerov and Dwyer (2000) and Azisov et al. (accepted) reported

changes in accumulation and ablation patterns of selected Central Asian glaciers, resembling a shift from continental to more
maritime glacier regimes. Such shifts consequently influence the mass balance sensitivity (Wang et al., 2017) and variability
(Barandun et al., 2021). In earlier studies, climatic settings were found to be the dominant drivers of the heterogeneous mass
balance sensitivity over High Mountain Asia and explained up to 60% of its spatially contrasting glacier response (Sakai and
Fujita, 2017). Under the assumption of equal climatology, the glacier morphology was found to explain up to 36% of the mass

balance variability for the Tien Shan, 20% for the Pamir-Alay, but only 8% for the Western and Eastern Pamir (Brun et al.,
2019).

The investigation of the drivers behind the observed spatiotemporal mass balance heterogeneity has so far only received
limited attention. This is mainly due to three reasons: 1) limited glaciological measurements, 2) large uncertainties about
meteorological variables, and 3) a limited understanding of non-climatic effects on glacier mass balance.

1) Glaciological measurements are conducted predominantly at annual resolution and are limited to a few, well accessi-
ble glaciers. Geodetic methods, which have become state-of-the-art to assess glacier mass balances, have limited temporal
resolution. Remote sensing provides a powerful tool to study inaccessible glaciers from space, however robust mass change
assessments remain limited to intervals of five years or more (e.g., Kääb et al., 2015; Brun et al., 2017; Wang et al., 2017;
Shean et al., 2020; Wouters et al., 2019; Hugonnet et al., 2021). Barandun et al. (2018) developed an approach to reduce the

uncertainties related to conventional mass balance modelling by incorporating observations of transient snowlines. Barandun
et al. (2021) has applied this approach in combination with geodetic mass changes and highlight, for the first time, hot-spots
of spatiotemporal heterogeneity and increasing mass balance variability in the different mountain ranges of the Tien Shan and
Pamir at annual temporal resolution.

2) The identification of possible climatic drivers for mass balance variability is strongly complicated by prevailing uncer-
tainties in climatic state variables due to a lack of station data in the remote and largely inaccessible terrain. These data are
crucially needed for validation and adjustment of gridded datasets (Zandler et al., 2019). Precipitation products, in particular,
from reanalysis, interpolation, and remote sensing can show up to 1000% difference in these remote locations (Palazzi et al.,
2013; Pohl et al., 2015; Immerzeel et al., 2015) and can barely cover the large range of orographic processes that affect e.g.
small scale precipitation events (Roe et al., 2003). Problems in remote sensing snow retrieval, and precipitation in general, over

complex topography render reanalysis products in most cases more suitable for capturing precipitation seasonality and spatial
patterns in the Pamir albeit overall intensities might not be captured well (Zandler et al., 2019; Pohl et al., 2015). The spa-
tiotemporal comprehensiveness of reanalysis data facilitates including various climatic variables at global scale in correlation
analysis that are otherwise not even available from simple meteorological stations or remote sensing/interpolation data prod-



ucts. This allowed, for example Hugonnet et al. (2021), to derive matching patterns of decadal glacier mass balance variability
with precipitation and temperatures in a global scale analysis.

3) Many glaciers in Central Asia are heavily debris-covered in their ablation areas and debris thickness can range consider-
ably (Kraaijenbrink et al., 2017). A scale-dependent debris-cover mass balance relationship and the lack of region-wide debris
thickness assessments limit the explanatory power of debris cover for region-wide glacier mass balance patterns in Central
Asia (Brun et al., 2019; Miles et al., 2022). Both the Tien Shan and Pamir are known to host numerous surge-type glaciers
(Mukherjee et al., 2017; Kotlyakov et al., 2008; Gardelle et al., 2012; Guillet et al., 2022). After a surge, the mass balance
regime of the glacier changes abruptly due to non-climatic reasons. After such pronounced advance, melt rates might increase
strongly, uncoupled from current local climate conditions. Guillet et al. (2022) show that there is no significant difference in
mass balance between surge-type and non-surge-type glaciers. However, they use two different geodetic mass balance estimates
that already show pronounced differences.

With the ultimate aim to better understand the climatic and non-climatic induced spatiotemporal glacier mass balance vari-
ability for the Tien Shan and Pamir previously reported in literature (e.g., Brun et al., 2017, 2019; Barandun et al., 2021;
Hugonnet et al., 2021), we aim to provide conclusive and accurate results and interpretations on the drivers of the glacier re-
sponse to climate change for Central Asia. Our analysis benefits from newly available and advanced high temporally resolved
mass balance estimates and new reanalysis products. Given the often unconstrained uncertainties in climatological / meteoro-
logical datasets for data sparse regions, we consider the analysis of the consistency of the different meteorological and mass
balance data sets as a fundamental and eminently needed first step. In this study, we therefore rigorously analyse different
products to pinpoint similarities and differences in the identified drivers behind the glacier mass balance changes in Pamir
and Tien Shan. We follow a systematical approach for testing three different reanalysis products that are or have been used
extensively in the region. Additionally, due to existing differences in glacier mass balance time-series, we also incorporate two
mass balance estimates. One are the snowline-aided estimates by Barandun et al. (2021), and one are geodetic mass balances by
Hugonnet et al. (2021). These are related to the most commonly used climatic variables temperature and precipitation from the
reanalysis datasets, and to the glacier specific topographic and morphological characteristics. Finally, to account for possible
regional data issues, the analyses are performed at different spatial subsets.

In short, our analysis follows the objectives 1) to reveal dominant drivers for glacier mass balance and associated uncertain-
ties resulting from dataset choices, and 2) to reveal the limitations connected to the use of gridded climate data products as
only estimates in the absence of ground truthing.





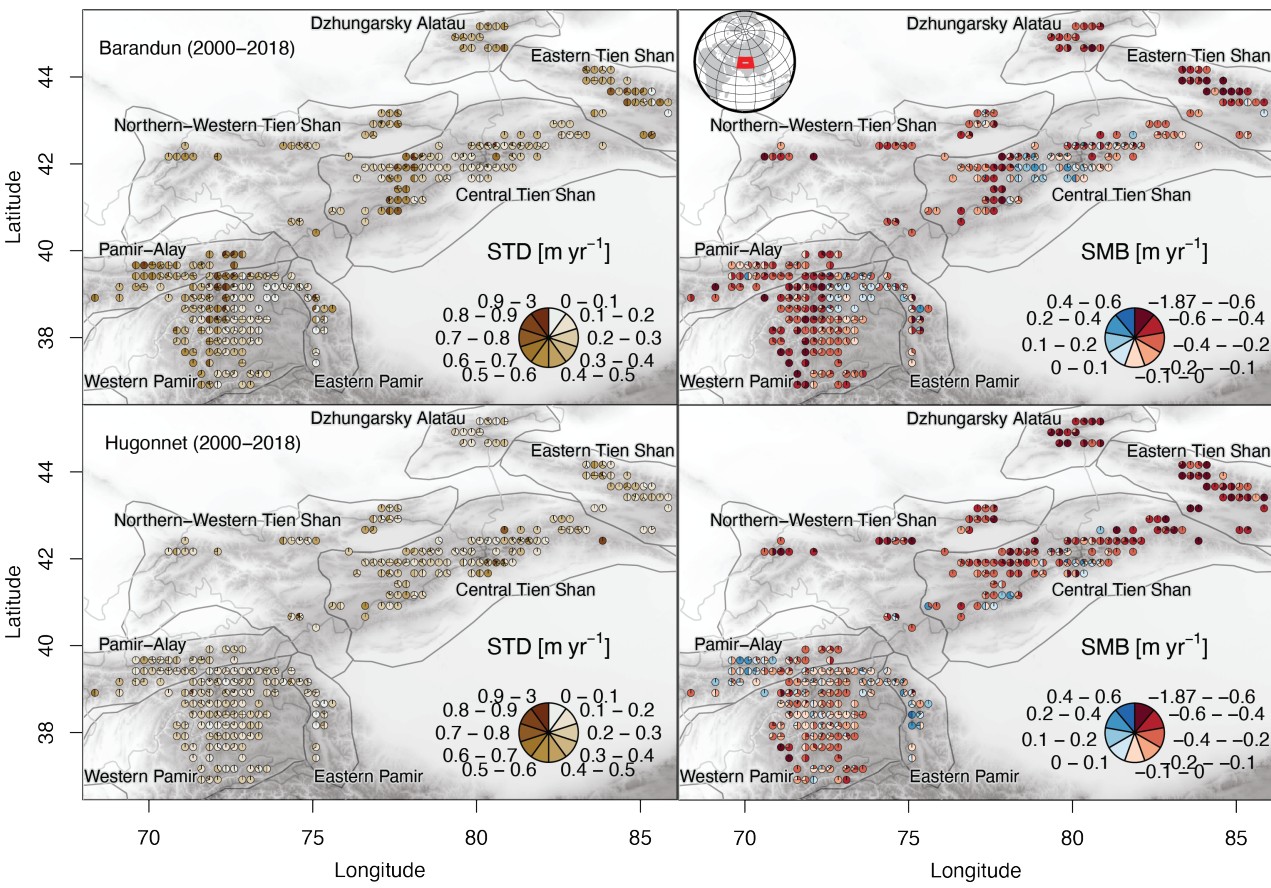

**Figure 1.** Overview map of the study region, the glacier mass balance variability in terms of standard deviation (STD; left) and average surface mass balance estimates (SMB; right) from the work of Barandun et al. (2021) (top) and Hugonnet et al. (2021) (bottom). Estimates from Hugonnet et al. (2021) only include the glaciers for which the transient snowline constrained modelling of Barandun et al. (2021) provides estimates. The pie charts aggregate values per glacier into classes and the relative class frequencies. Pies are not scaled to glacier area. The seven subregions of the HiMAP classification (Bolch et al., 2019) are shown in grey outlines.

## 2   Study Site and Data

### 2.1   Study sites

#### 2.1.1   Climate setting and variability

The following shall provide some insights on the complex climate settings of High Mountain Asia that reanalysis products are expected to depict. These settings range from humid, maritime, to continental and hyper-arid (Bohner, 2006; Schiemann et al., 2007; Yao et al., 2012; Maussion et al., 2014). Central Asia is a mostly arid to semi-arid region (Barry, 1992) with high





seasonal precipitation variability due to its continentality (Haag et al., 2019). Synoptic large-scale meteorological conditions over Central Asia are the result of the main direction of the zonal flow of the air masses from west to east. A deflection of

westerly trade winds (Westerlies) to the north and south at the western orogen margin of Tien Shan and Pamir cause intense precipitation in the west and the barrier effect creates increasingly arid conditions towards the central and eastern part of the main mountain ranges (Pohl et al., 2015; Aizen et al., 2009, 1995). Meridional airflow can occur when tropical air masses enter from south and south-west or when north-westerly, northerly and sometimes even north-easterly cold air masses intrude into Central Asia (Schiemann et al., 2008, 2007).

The climate variability depends strongly on how and when the different weather types interact (Zhao et al., 2014; Wei et al., 2017; Gerlitz et al., 2020), guided principally by the position and strength of the jet stream. Schiemann et al. (2008) investigated in detail the seasonal cycle of the Central Asian climate to show that the jet stream is situated over the north of Central Asia during the summer months and that it moves towards the south in autumn, creating atmospheric instabilities. Resulting precipitation occurs mainly at the western margin until mid-January. Subsequently, the influence of the jet stream

weakens over Central Asia and the Siberian high-pressure system creates clear and calm winter weather, especially reducing winter precipitation in the north and east (Aizen et al., 1997). By the end of February, the jet stream returns northwards and reaches the southern edge of Central Asia carrying warm and moist air. This creates a temperature contrasts between the north and the south and strengthens the cyclonic activity over Central Asia (Schiemann et al., 2008). Consequently, the highest amount of precipitation, characterised by heavy showers and thunderstorms, occurs in March and April and culminates in the

western parts of the Tien Shan and the Pamir. While the jet stream continues northwards during May, precipitation maxima are reached in the Northern Tien Shan in June (Aizen et al., 2001). At the beginning of the summer, the cyclonic activity weakens and heat lows start to form again. During summer, the Siberian anticyclonic circulation provides cold and moist air masses in Northern, Central and Eastern Tien Shan, resulting in frequent spring or summer precipitation (Aizen et al., 1997). The most dominant moisture source at the southern margins of the Pamir are heavy rainfalls provided by the Indian Summer Monsoon

(e.g., Cadet, 1979). Orographic shielding at the south and south-eastern margin of Central Asia's mountain ranges however strongly reduce this moisture supply and lead to very dry conditions in the central parts of the Pamir (Boos and Kuang, 2010; Haag et al., 2019). The Tibetan anticyclone influences additionally the local climate along the eastern margin of the Pamir (Archer and Fowler, 2004), leading to summer cooling (Forsythe et al., 2017) and summer rainfall (Aizen et al., 1997).

### 2.1.2 Topography and glaciation

The Tien Shan and Pamir are the two main mountain ranges of Central Asia in the North of the Karakoram and Hindu Kush. Here, we chose a subdivision of the regions based on the commonly-used HiMAP regional division suggested in Bolch et al. (2019) into: Western / Northern Tien Shan, Eastern Tien Shan, Central Tien Shan, Dzhungarsky Alatau, Pamir-Alay, Western Pamir and Eastern Pamir (Fig. 1). The Tien Shan hosts almost 15,000 glaciers, covering a surface area of $\approx 12'300\,km^2$ (according to the Randolph Glacier Inventory Version 6.0 (RGIv6.0, RGI, 2017). The Pamir including Pamir-Alay (also

Hissar-Alay) hosts around 13,000 glaciers covering similarly a surface area of $\approx 12'000\,km^2$. Highest mountain ranges are found in the Central Tien Shan and Western and Eastern Pamir. Barandun et al. (2021) showed similar mass loss for the





Tien Shan and Pamir. Mass balance estimates of Barandun et al. (2021) show the least negative mass balances from 2000 to 2018 for the Central Tien Shan ($-0.13\pm0.37\,m\,w.e.\,yr^{-1}$) and Eastern Pamir ($-0.12\pm0.37\,m\,w.e.\,yr^{-1}$). Dzhungarsky Alatau ($-0.46\pm0.37\,m\,w.e.\,yr^{-1}$) and Eastern Tien Shan ($-0.48\pm0.37\,m\,w.e.\,yr^{-1}$) were the subregions with the strongest

mass loss and the mass balances for the Pamir-Alay ($-0.32\pm0.37\,m\,w.e.\,yr^{-1}$), Western Pamir ($-0.25\pm0.37\,m\,w.e.\,yr^{-1}$) and the Northern / Western Tien Shan ($-0.30\pm0.37\,m\,w.e.\,yr^{-1}$) are close to the region-wide average from 2000 to 2018.

## 2.2 Data

### 2.2.1 Annual mass balance time-series

We use two annually-resolved and glacier specific datasets for glacier mass balance estimates; one based on transient snow line

observations and the other one on digital elevation model (DEM) differences.

The first MB$_{Barandunetal}$ are annual time series provided by Barandun et al. (2021) who used a mass balance model combining transient snowlines, as a proxy for surface mass balance, together with derived geodetic mass changes to provide annual mass balance time series. 255,000 automatically mapped snowline observations (Naegeli et al., 2019) from 2000 to 2018 for the Tien Shan and Pamir were used for model calibration. The snowlines were mapped for each glacier individually on over

3,000 Landsat surface reflectance scenes with cloud cover of <50%. The transient snowlines are used to directly calibrate a temperature-index and distributed accumulation model driven with ERA-interim reanalysis (Dee et al., 2011) data for each glacier and year separately (Barandun et al., 2021, 2018). In a second step, semi-decadal to decadal geodetic mass balances were integrated into the model calibration (Barandun et al., 2021). The geodetic mass balances were derived from Advanced Spaceborne Thermal Emission and Reflection Radiometer (ASTER) scenes and High Mountain Asia DEMs (Girod et al., 2017;

Shean et al., 2017). A total of over 25,000 individual geodetic estimates were homogenized to a common reference period from 2000 to 2018 using long-term glaciological measurements (Zemp et al., 2019). These glacier-specific decadal to semi-decadal geodetic mass changes were then used to constrain the modelled mass balance time series in order to reach agreement between the two observational datasets. Thus, Barandun et al. (2021) provided annual mass balance time-series, closely tied to direct observations, for roughly 60% of the glaciers larger than $2\,km^2$. For more details on the methodological approaches for mass

balance determination the reader is referred to Barandun et al. (2018, 2021), for the automatic snowline mapping to Naegeli et al. (2019), and for the geodetic estimates to Girod et al. (2017) and McNabb et al. (2019).

The second dataset MB$_{Hugonnetetal}$ are geodetic mass balance estimates by Hugonnet et al. (2021). Their estimates rely on DEM differences and filtering techniques. The predominant input is the 20-year-long archive of stereo images from the ASTER used to derive time-series of DEMs. Their estimates were validated at all glaciers world-wide with intersecting laser

altimetry and optical elevations and show good agreement at the global scale although at local and regional scale biases can persist. Geodetic methods are more accurate at longer time-scales as the signal-to-noise ratio improves. However, Hugonnet et al. (2021) also provide annual estimates, which we use in this work. We select only the same glaciers from this dataset for which we have estimates from Barandun et al. (2021). This serves for a consistent analysis of dominant drivers and at the same





time highlight differences between the two mass balance estimates.


### 2.2.2 Glacier morphological characteristics

We use glacier outlines and areas from the RGIv6 (RGI, 2017), which were kept unchanged over time, and the freely available, void-filled Shuttle Radar Topography Mission (SRTM, Jarvis et al., 2008) digital elevation model to derive glacier topography, aspect and slope as morphological parameters. We use further the information provided in Scherler et al. (2018) on percentage

of debris-cover on individual glaciers for the Tien Shan and Pamir and the data on surge activity from (Guillet et al., 2022). The latter provides multiple statistics on surge activity. We chose to simply use occurrence of surge activity provided in this dataset to test if there is a significant difference between surge and non-surge type glacier mass balances in $\mathrm{MB}_{Hugonnetetal}$ and $\mathrm{MB}_{Barandunetal}$.

### 2.2.3 Meteorological data

In order to investigate possible relationships of the mass balance with the key meteorological drivers, precipitation and air temperature, we use three different reanalysis dataset and a remote sensing snow cover product. The reanalysis products are 1) the European Centre for Medium Range Weather Forecast (ECWMF) fifth generation atmospheric reanalysis ERA5 (Hersbach et al., 2020), 2) the Climatologies at High resolution for the Earth's Land Surface Areas (CHELSA) time-series version 2.1

(Karger et al., 2017, 2021), and 3) the High Asia Refined analysis (HAR) in 30 km spatial resolution HAR30 version 1.4 (Maussion et al., 2011). The snow cover product is the Moderate resolution Imaging Spectroradiometer (MODIS) monthly snow cover product MOD10CM version 6 (Hall and Riggs, 2015). We chose the time period 2000–2018, as well as 2000–2014 for all dataset to match the limited temporal availability of HAR30 v1.4 (Maussion et al., 2011) for which only the second period was analysed.

**ERA5** monthly averaged data on single levels from 1979 to present is an atmosphere reanalysis dataset (Hersbach et al., 2020) with a $0.25° \times 0.25°$ native spatial resolution. ERA5 incorporates a multitude of *in situ* and remote sensing data in an integrated forecasting system to produce hourly outputs of atmospheric variables. We obtained the monthly means (sums) product from the ECMWF data store for temperature (precipitation). ERA5 data have not been thoroughly validated in hydrological and glaciological model applications in Pamir and Tien Shan. A significant overestimation of snow depth (Wang et al.,

2021; Orsolini et al., 2019) is problematic for these applications and is also assumed to alternate the energy fluxes related to overestimated albedo (Wang et al., 2021). This renders the parameters for in- and outgoing energy fluxes problematic, and we focus only on the two variables precipitation (P) and 2 m air temperature (T). Consequently, we focus on P and T for the other datasets as well.

**CHELSA** time-series version 2.1 is a processed and downscaled version of the ERA5 dataset. It incorporates directional

wind speeds and cloud cover observations to address precipitation biases, as well as wind- and leeward distributions (Karger et al., 2017). The final downscaled product has a ≈1 km spatial and monthly temporal resolution.





**HAR30** version 1.4 (Maussion et al., 2011) has a 30 km spatial and daily temporal resolution. The data resulted from a dynamical downscaling of global analysis data (Final Analysis data from the Global Forecasting System (National Centers for Environmental Prediction, National Weather Service, NOAA, 2000); data set ds083.2) using the Weather Research and

Forecasting (WRF-ARW) model (Skamarock and Klemp, 2008). The data set shows high consistency with temperature, precipitation, and snow cover measurements in several regions of High Mountain Asia, where uncertainties are particularly large due to limited or non-existent meteorological stations. It captured climatic extremes in the greater Pamir region that lead to floods and droughts (Pohl et al., 2015). It was used for glacier studies in the greater Himalayan region (Maussion et al., 2011; Mölg et al., 2013; Curio et al., 2015), and it was applied in hydrological studies (Pohl et al., 2015, 2017; Biskop et al., 2016).

While there was a need to bias-correct HAR precipitation intensities in the Pamir (Pohl et al., 2015) and the Tibetan Plateau (Biskop et al., 2016), the dataset showed superior correlation with *in situ* measurements than interpolated and remote sensing dataset in Pamir (Pohl et al., 2015). Due to large uncertainties in gridded meteorological dataset, we believe that the HAR dataset, which has been validated at least to some degree in High Mountain Asia in several previous studies (e.g. Pohl et al., 2015, 2017; Biskop et al., 2016; Maussion et al., 2014), should be included in our analysis.

**MOD10CM** is a monthly average snow cover dataset on a regular climate modelling grid with approximately 5 km spatial resolution (Hall and Riggs, 2015). The snow cover is calculated using the normalized differences snow index (NDSI) constrained to positive values, providing snow cover values in the range between 0 to 100%. From the original snow cover fraction values we also derive the temporal changes between consecutive time steps as a measure for snow accumulation or depletion events. The two variables are simply referred to as snow cover (SC) in the following.


## 3 Methods

The mass balance estimates of Hugonnet et al. (2021) and Barandun et al. (2021), provide average long-term values for the two periods (2000–2014 and 2000–2018), but more importantly, also annual time-series for individual glaciers. In turn, this allows us to run two kinds of analysis to determine possible drivers explaining the mass balance variability: 1) a temporal

analysis for each glacier individually by correlating the mass balance and meteorological time series, and 2) a spatial analysis relating mean glacier mass balance with statistics of meteorological and morphological data, e.g. long-term averages, trends, or the standard deviation as a measure for mass balance or meteorological variability. Both types of analysis are based on multiple linear regression analysis to identify significant predictors in the group of meteorological or morphological variables. An overview of this workflow is given in Fig. 2. In this study, we focus strictly on the determination of significant predictors in

multiple linear regression analysis and do not report, nor discuss obtained model coefficients or coefficients of determination in detail. Our reasoning is based on the very different outcomes of the analyses, based on the choices of meteorological dataset and mass balance estimates questioning the decisiveness of the results, and to streamline the scope of the work.

In order to systematically identify all significant (always at a significance level $\alpha$=0.05) and independent variables while accounting for some complexity in glacier accumulation and ablation processes, we use all possible combinations of two





independent predictors for each meteorological and morphological dataset, respectively. For the meteorological analysis, we constrain these combinations to individual dataset products, e.g. only ERA5, but extend this analysis also to include the MODIS snow cover data in a second step. This allows exploring if the observational snow cover dataset adds more explanatory power than the reanalysis.

We do not run a separate collinearity test to identify and remove correlating predictors even though this is expected, for ex-
ample, in the case of different temporal temperature aggregates (Fig. 2). As our focus is to show possible inconsistent outcomes depending on chosen glacier mass balance and meteorological dataset, we focus only on identifying significant predictors rather than model coefficients. A possible impact in case of collinearity is a non-significance of one, or both of the two correlating predictors (Vatcheva et al., 2016). Due to the large number of combinations for the meteorological dataset (n=1480 for the spatial analysis including MODIS snow cover), we expect that randomly omitted predictors would, however, show up in a
different combination and not be completely omitted.

### 3.1  Spatial correlation

The spatial correlation is using the derived average mass balances of the 1220 glaciers, either in their entirety or divided into different regions according to two different classifications; 1) HiMAP after Bolch et al. (2019) according to main mountain
range names (Fig. 1), and 2) to the mass balance variability found in Barandun et al. (2021). The latter is determined using k-means clustering with 3 fixed classes (Fig. 3). This division is done to see if there are different dominant predictors for these "hot spots" of glacier mass balance variability.

We use either meteorological or morphological parameters as predictors. For meteorological predictors, we use calculated average, standard deviation, and trend or tendency (derived slope from a temporal regression), each for the different seasons
spring, summer, autumn, winter, and the entire hydrological year (October-September) (Fig. 2). As static morphological predictors, we include the variables latitude, longitude, slope over glacier tongue, median elevation, aspect, and glacier area from the RGI, as well as debris cover from Scherler et al. (2018). For aspect, we use the sine and cosine of the original values given in the range between 0 and 360° to have consistent value ranges in zonal and meridional directions. We showcase, additionally, any possible relationship with surge activity using separate histograms and boxplots where we simplify the complex informa-
tion on surge activity given in Guillet et al. (2022) as either having experienced surge activity (1) or not (0) during the 2000 to 2018 study period.

### 3.2  Temporal correlation

In addition to the spatial analysis, we also run multiple linear regression analysis on the glacier mass balance time-series per glacier to identify significant predictors over the time domain. The annual glacier mass balance time series are the dependent
variable and combinations of meteorological variables from individual reanalysis datasets are the predictors. As this yields one result per glacier, we aggregate the results visually in form of maps. We identify both, significant variables, as well as



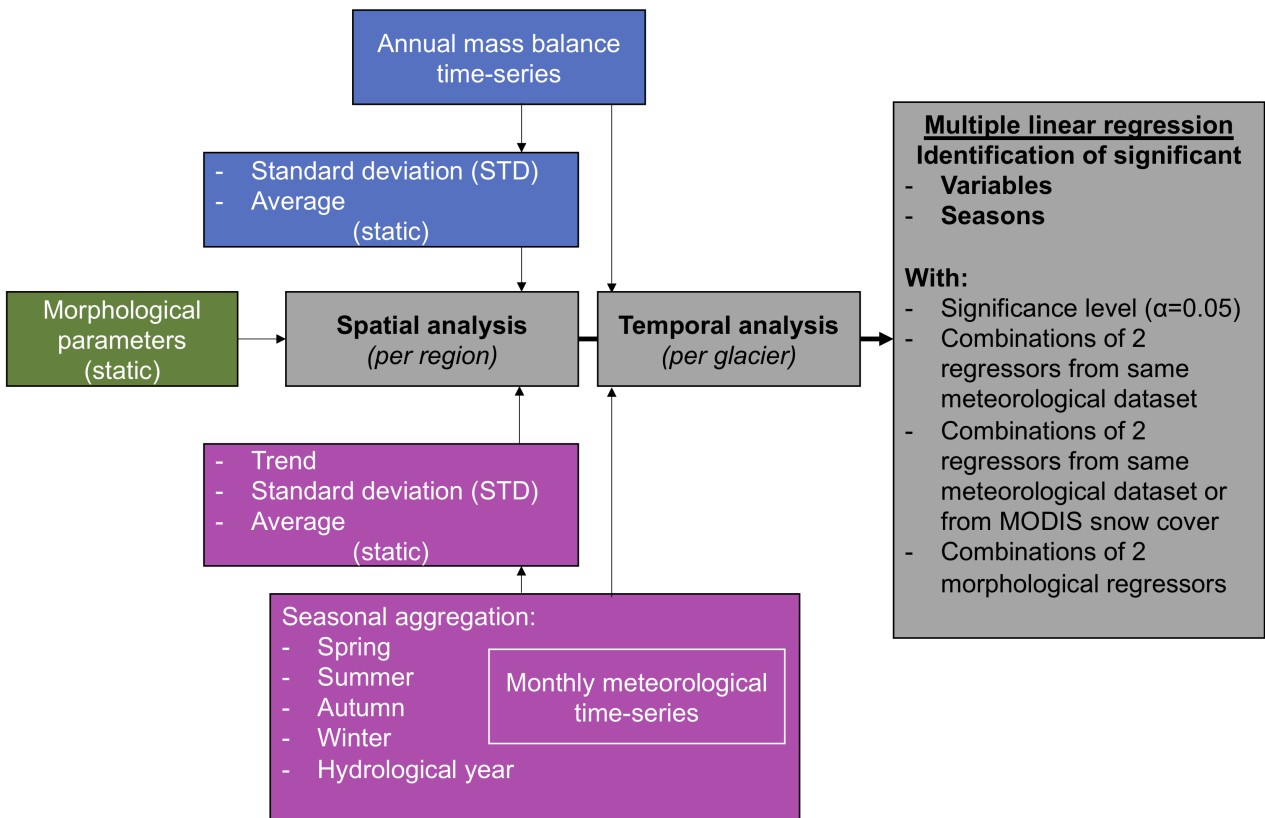

**Figure 2.** Workflow of spatial and temporal analysis using the mass balance estimates from Barandun et al. (2021) and Hugonnet et al. (2021), and the meteorological data from the three different reanalysis products and MOD10CM snow cover, as explained in the data section. Focus in this study is the identification of significant relationships rather than explained variance or other statistics. Therefore, identification of important variables and seasons is solely based on derived *p-values* from multiple linear regression analyses.

significant seasons in this way to identify possible patterns and differences resulting from the use of a particular reanalysis or mass balance dataset.

# 4 Results

## 4.1 Spatial correlation

Despite a relatively high number of significant correlations between mass balances and meteorological variables (70-90%), the variability between the different dataset is high, leaving an interpretation on dominant drivers for the Tien Shan and Pamir uncertain (Fig. 4). Obtained coefficients of determination ($R^2$) are in the range of 5% for the entire Pamir and Tien Shan per





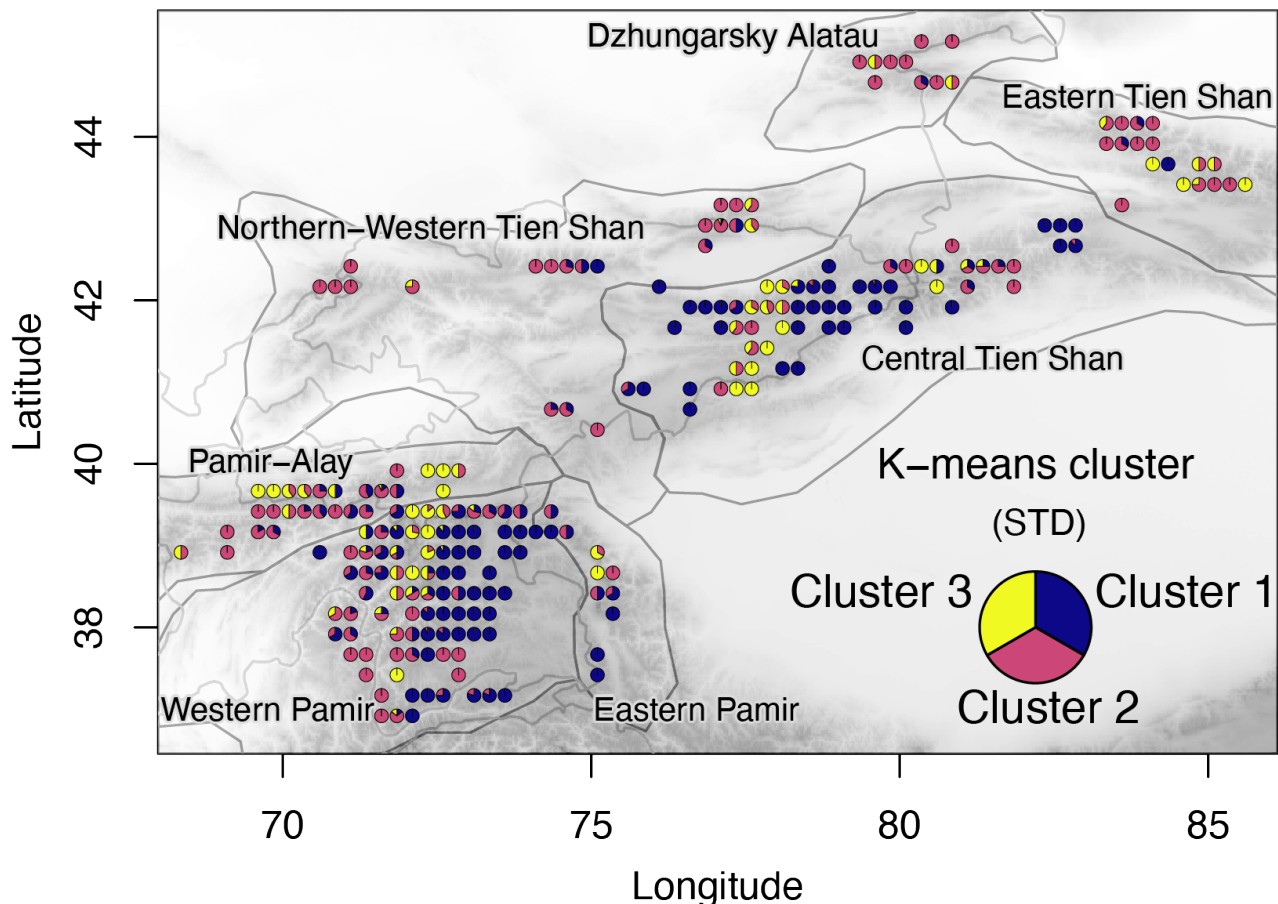

**Figure 3.** K-means clusters according to mass balance variability (standard deviation - STD) for the 2000–2018 period from Barandun et al. (2021).

variable and increase to up to 20% for sub-regions (Fig. A1). When including remote sensing snow cover (SC) in addition to
the reanalysis meteorological variables (T and P), SC constitutes about half of the significant correlations. Without SC, an akin
number of significant correlations indicates that SC is likely explaining a similar portion of the variance in the mass balance,
but slightly more significantly than the variables from the reanalysis. The $R^2$ remains low (Fig. A1). No dominant season can
be identified in the analysis for the Tien Shan and Pamir. The variability in the meteorological variables within each season or
for the hydrological year are equally important (Fig. 4; middle).

Of the tested morphological parameters, 60 to 70% showed a significant correlation with the mass balances for the entire
Tien Shan and Pamir (Fig. 4; bottom), and no individual morphological variable stands out as the most dominant one (Fig. 4).
Each was determined in around 10% of combinations as significant predictor. Debris cover is not identified in any of the cases
using either mass balance dataset for the short or long time period. This only changes in the regional division where debris



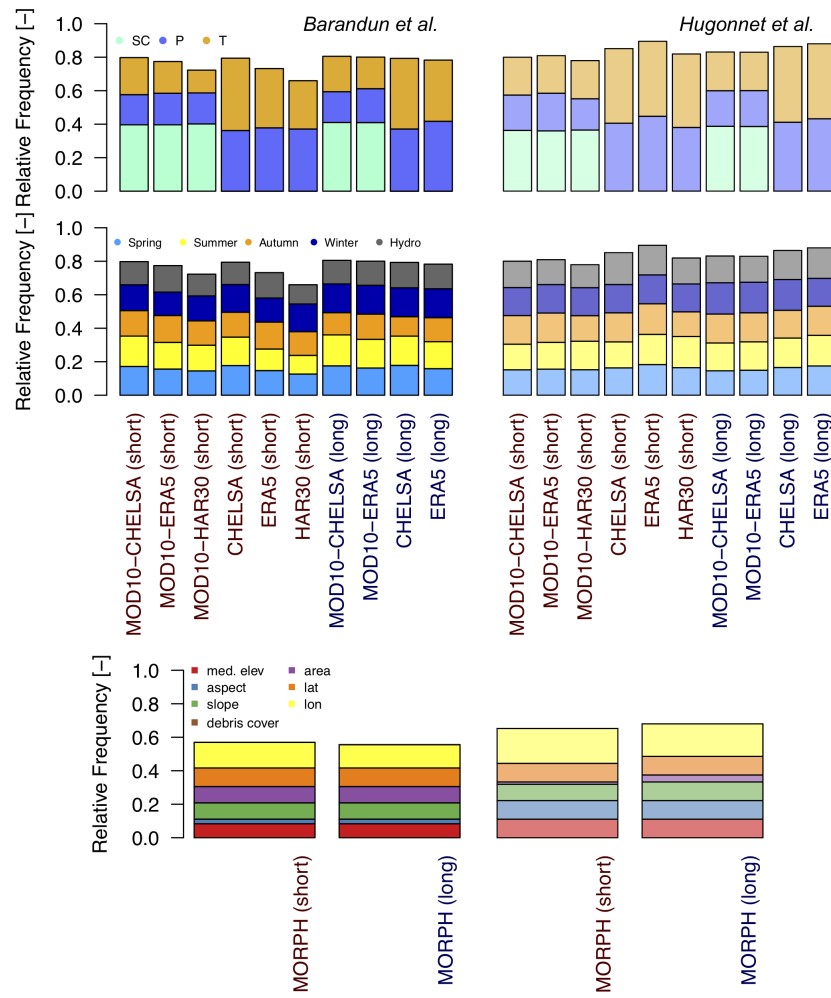

**Figure 4.** Importance of climatic variables (top), season (middle), and morphological parameters (bottom) for the entire Tien Shan and Pamir. Solid colors (left) refer to MB$_{Barandunetal}$ and semi-transparent colors (right) refer to MB$_{Hugonnetetal}$.

cover is identified for some sub-regions. However, depending on the mass balance estimates and sub-region, the identified

morphological parameters can change entirely (Fig. A2). The mass balance distribution indicates no significant difference between surge and non-surge type glaciers (Fig. A4). The small number of surge type glaciers and uneven distribution across the regions (Fig. A3) prevents a conclusive result.

Considering the different HiMAP regions (Bolch et al., 2019), the number of significant correlations between mass balance and meteorological variables varies strongly, and is depending also on the mass balance dataset (Fig. 5). Overall, the total num-

ber of identified important variables are similar between the two mass balance estimates. However, strong differences are apparent regarding which variable and temporal aggregation (seasons) is dominant. Likewise, the choice of a different reanalysis dataset results in different variable and temporal aggregation importance. For example, there are many significant correlations



for the Western Pamir and Northern/Western Tien Shan for $MB_{Barandunetal}$ and for Central Tien Shan for $MB_{Hugonnetetal}$; a medium amount of significant correlations for the Eastern and Central Tien Shan for $MB_{Barandunetal}$ and for the Western

Pamir and Pamir-Alay for $MB_{Hugonnetal}$ (Fig. 5). For all other regions, the importance of meteorological variables is low and very variable from one dataset to another. For the majority of cases, the number of significant correlations is higher for CHELSA than for ERA5, despite CHELSA being based on ERA5 data as input. Regardless of contributing to at least half of the significant combinations for most regions, snow cover increases the total amount of significant correlations only for the Central Tien Shan for $MB_{Barandunetal}$ and for the Eastern Tien Shan, Pamir-Alay Western and Eastern Pamir for $MB_{Hugonnetetal}$.

For all other regions, temperature and precipitation provide more significant correlations when not considering snow cover. Combinations with snow cover barely showed significant correlations in the Eastern Pamir and the Dzungarsky Alatau (Fig. 5). Overall, we found clearly more significant combinations including precipitation than temperature for the Western Pamir, and the Pamir Alay. Temperature seemed more important for the Eastern and Northern/Western Tien Shan and to some degree also for the Central Tien Shan. For the Dzhungarsky Alatau the results varied strongly between the different datasets, and

prevented a conclusion on a dominance of one of the meteorological variables. For the Western Pamir and the Eastern Tien Shan, winter and spring changes contributed to most of the significant combinations. For the Pamir Alay, the autumn season, and for the Central and Northern/Western Tien Shan, the summer and autumn seasons showed most significant correlations. For the Eastern Pamir and the Dzhungarsky Alatau, the different seasons contributed more or less equally to the amount of significant combinations.

Besides the importance of the meteorological variables, the morphological parameter importance is relatively high for the Western Pamir (approx 50%; Fig. A2). However, no dominant morphological parameter could be identified and the variability across the different HiMAP regions as well as for the different mass balance estimates is relatively high (Fig. A2). The only region in which both morphological and meteorological parameters showed a high frequency of significant correlations is the Western Pamir. In contrast, for Dzungarsky Alatau and Eastern Pamir, neither category of parameters showed a high number

of significant correlations with either of the mass balance estimates.

  Within the k-means derived "hot spot" regions of different mass balance variability, there is always a high number of significant correlations (Fig. 6), different to the regional HiMAP classification. We found for over 80% of meteorological variables significant correlations in cluster 1, slightly less in cluster 3 (60–80%), and the lowest amount, with high variability depending on the meteorological dataset used, for cluster 2 (40–70%). Including snow cover did not increase the overall number of sig-

nificant combinations for any of the 3 clusters, but it is the most important (relative highest number of correlations) variable for clusters 1 and 3. For cluster 1 (mainly positive and more balanced "hot spot" and low glacier mass balance variability in $MB_{Barandunetal}$), precipitation seems slightly more important than air temperature, and vice-versa for cluster 3 (negative "hot spot" and high mass balance variability in $MB_{Barandunetal}$), temperature is slightly more dominant (Fig. 6). Cluster 2 (average mass balance with low variability, mainly along the Western/North-Western margins) is temperature dominated, and

snow cover showed the lowest number of significant correlations among all clusters.

  For cluster 1, no clear dominance of either season is visible (Fig. 6). In contrast, for cluster 2 and 3 there are some dominant seasons, however, dependent on the mass balance dataset. Generally, there is a more even distribution in seasonal importance





**Figure 5.** Importance of meteorological variables (left) and seasons (right), as the number of significant correlations, identified using mass balance estimates from Barandun et al. (2021) (solid color) and from Hugonnet et al. (2021) (semi-transparent color) for the HiMAP regions.



for MB$_{Hugonnetetal}$ than for MB$_{Barandunetal}$. The highest variability of dominant seasons was found for cluster 2 based on the reanalysis dataset used, with particularly strong variability in spring and winter importance. Regarding the two mass

balance estimates, MB$_{Barandunetal}$ showed dominance of summer and autumn, contrasting winter and spring dominance for MB$_{Hugonnetetal}$. The morphological variables showed less significant correlations than the meteorological variables, especially for MB$_{Barandunetal}$ in cluster 2 and 3 (Fig. A2).

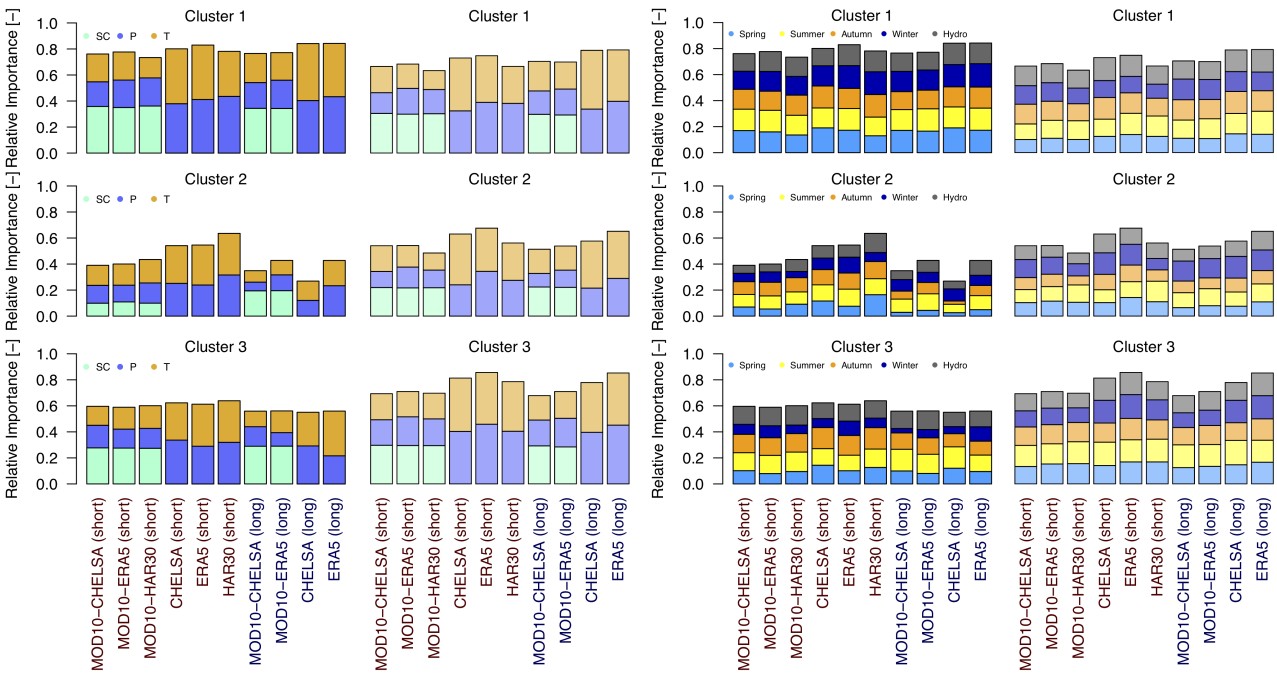

**Figure 6.** As Fig. 5 but with glaciers aggregated according to similar glacier mass balance standard deviations (k-means clusters) found in Barandun et al. (2021).

## 4.2 Temporal correlation

The explanatory power of the meteorological variables for the glacier-wide mass balance time series per individual glacier

is between 30 and 50% (Fig. A5). However, similar to the spatial analysis, regional and dataset-dependent differences are apparent.

The importance of the variables depends strongly on the dataset used, and our analysis led to opposing results using the different products. This concerns the different reanalysis products but also the different mass balance time series. As an example, for the Pamir-Alay snow cover is the dominant variable for MB$_{Hugonnetetal}$, whereas temperature and precipitation seem more

important for MB$_{Barandunetal}$. A similar discrepancy is found for the different reanalysis datasets (Fig. 7). For example, for MB$_{Hugonnetetal}$, the central and southern parts of the Western Pamir are dominated by temperature using the HAR dataset, by





snow cover using ERA5, and by precipitation when using CHELSA. Even stronger visible is this discrepancy when excluding snow cover in the analysis, only using temperature and precipitation as predictor for the mass balance time series (Fig. A6).

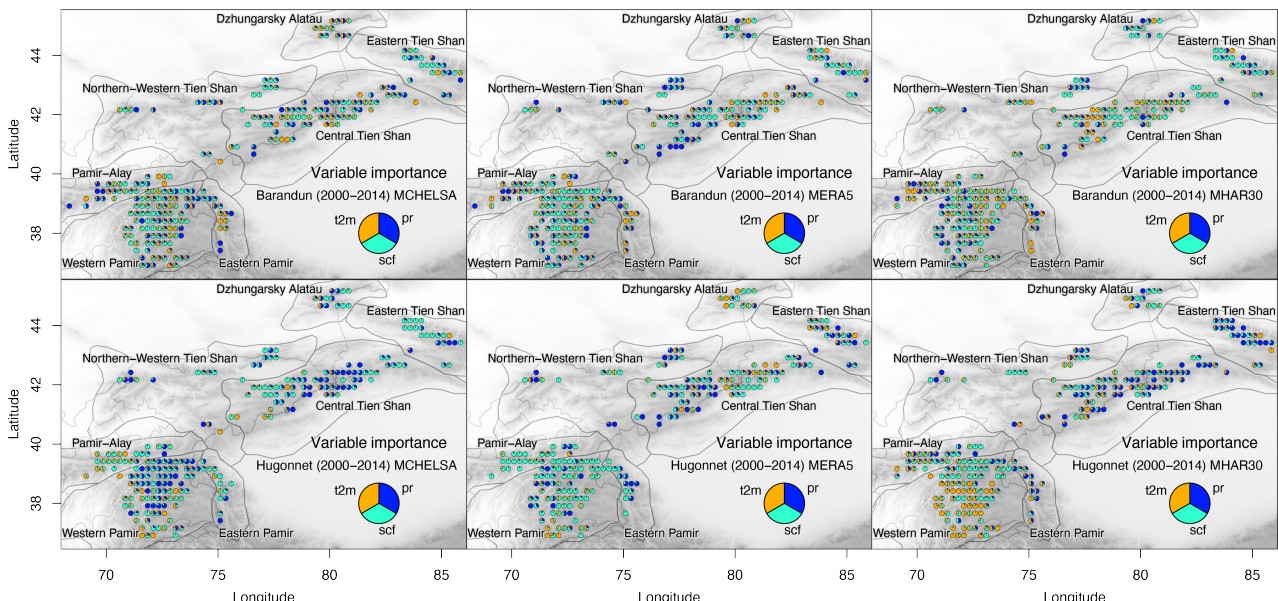

**Figure 7.** Importance of meteorological variables for the temporal analysis between annual mass balance estimates from Barandun et al. (2021)(top) and Hugonnet et al. (2021)(bottom), and two meteorological variables from the different reanalysis datasets, with and without snow cover as indicated in the figure legends.

Similarly, for the seasonal importance, results vary largely for the different glaciers depending on the dataset used and provide a more detailed view than the spatial analysis (Fig. 8). No consistency could be found either for the same mass balance time series with different reanalysis products, or for different mass balance time series with the same reanalysis product. When including snow cover, the pattern of variable importance becomes slightly more consistent in regions where snow cover replaces other variables as most important variable (Fig. 8) without any apparent change in explained variance (Fig. A5).

## 340   5   Discussion

The correlation analyses between the two mass balance estimates and different reanalysis products show foremost that a clear identification of dominant drivers of glacier evolution is not possible in Pamir and Tien Shan with the currently available reanalysis datasets. Even slight differences in P and T, for example, between CHELSA and ERA5, where the former is a result of downscaled ERA5, lead to partly opposing results (Fig. 7 and Fig. A6). This finding is independent of what mass balance esti-
mate is used or if the short (2000–2014) or long period (2000–2018) is considered. Even though the two utilized mass balance estimates mainly agree at the regional scale, the differences at local scale amplify the inconsistencies. Different conclusions can





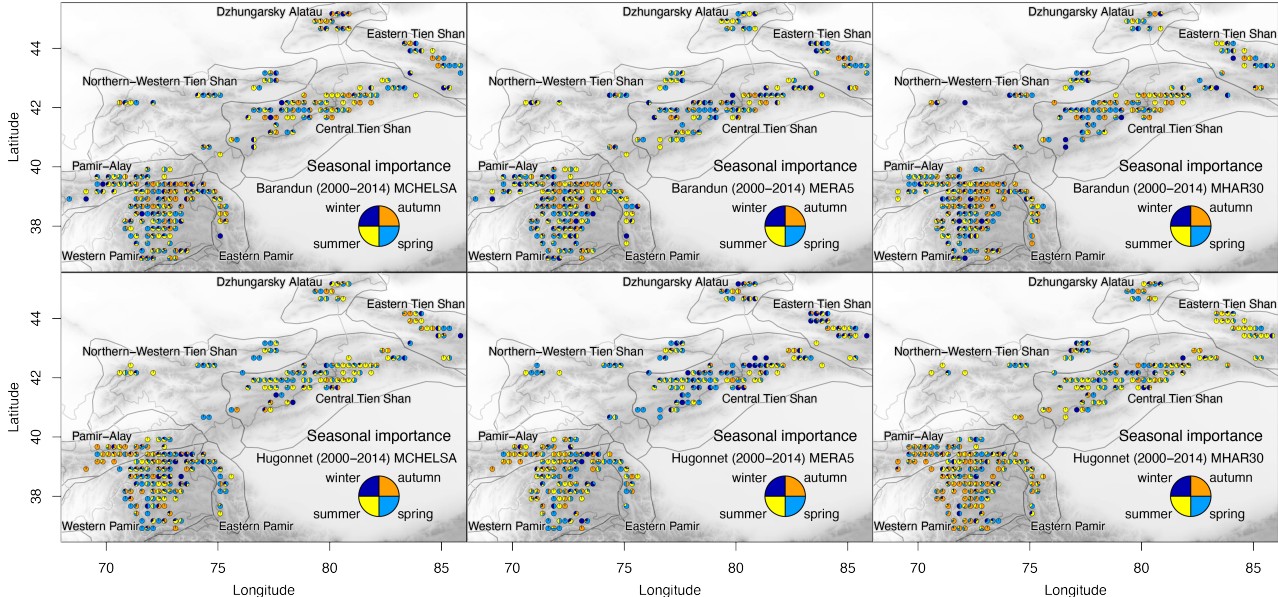

**Figure 8.** Seasonal importance from the temporal analysis between annual mass balance estimates from Barandun et al. (2021)(top) and Hugonnet et al. (2021)(bottom) and two meteorological variables from the different reanalysis datasets and snow cover as indicated in the figure legends.

also be drawn depending on whether the spatial or temporal domain builds the basis for the analysis as presented in Sections 4.1 and 4.2. The temporal analysis presents a very detailed overview of where and which meteorological variables correlate with mass balance time-series. On the other hand, long-term average mass balance estimates, which are the standard output

from geodetic mass balance studies, prevent such an analysis. Consecutive aggregation or dividing the total number of glaciers into adequate regional clusters are somehow arbitrary and can significantly influence the outcome as shown in Section 4.1. However, the presented results, in particular from the temporal analysis, also allow to find similarities that suggest consistent meteorological representations in some regions by identifying the same variable and seasonal importance and their associated glaciological processes.


## 5.1 Spatial Analysis

Our results reveal, largely independent of mass balance estimate and reanalysis products, a high number of significant correlations between T, P and mass balance for the entire Pamir and Tien Shan region (Fig. 4). The relative number of significant correlations remains similar when introducing SC, suggesting that the meteorological parameters from the reanalysis and snow

cover relate in the same way to the spatial variability in glacier mass balances.





When dividing the glaciers regionally, either according to the HiMAP classification or according to the k-means temporal mass balance variability found in Barandun et al. (2021), the amount of significant correlations between T, P, snow cover and mass balance drops. It can also be seen that identified variables depend strongly on the datasets used (Fig. 5 and 6). Using the division based on the k-means clustering shows consistently more significant correlations, which might be related to the

more even number of glaciers per division class in comparison to the HiMAP regions. The decreasing number of significant correlations for certain subregions can have multiple reasons. (1) The provided meteorological variables are not sufficient to represent the relevant processes responsible for the mass balance variability. For example, at high elevations, the relevance of temperature and long-wave radiation can decrease in favor of short wave radiation which could likewise explain lower numbers of significant correlations for T in Eastern Pamir where the highest glaciers are located (Sicart et al., 2011; Yang et al., 2011). (2)

Especially in Eastern Pamir and Dzhungarsky Alatau, a relatively low number of modelled glaciers adds to the difficulty to find a correlation with the meteorological variables. (3) Varying mass balance variability, sensitivity or mass balance gradients and regimes that are not well represented within one subregion can further complicate finding consistent relationships. Typically, the Tien Shan and Pamir are classified as subcontinental (western and northern part of the Tien Shan, and the Pamir-Alay) to continental glacier regimes (central part of the Tien Shan, and central and eastern part of the Pamir) (Wang et al., 2019).

Under ongoing climate change, glacier response to meteorological conditions are undergoing important changes due to a changing mass balance sensitivity and variability (Azisov et al., accepted; Dyurgerov and Dwyer, 2000), partly related to a shift from continental to more sub-continental conditions. Small scale processes (e.g. changes in pore space) close to the equilibrium line altitude can change accumulation patterns to which such glaciers react sensitively (Kronenberg et al., 2022). This adds to the heterogeneous mass balance response and renders finding direct correlations with climatic drivers difficult.

Based on station data, Wang et al. (2019) showed that subcontinental glaciers react mostly to air temperature variability, whereas the continental glaciers, generally located at higher elevations, are sensitive to both air temperature and precipitation. Faster changing accumulation and ablation processes or a rotation and steepening of the mass balance gradients (increased ablation and accumulation) will change the mass balance sensitivity and variability on the short to long term (Kronenberg et al., 2022; Dyurgerov and Dwyer, 2000; Kuhn, 1980, 1984). This will, hence, shift the dependence on specific meteorological

variables. In addition to these changes, already slight uncertainties in meteorological estimates might complicate finding a relationship with sensitive glacier responses. This is especially important in sub-continental regions, where sensitivity is highest and small variations at various spatial scales over a glacier are invisible in the coarsely resolved reanalysis products. In contrast, at high altitudes and more continental climate regimes, larger changes are needed for a glacier response. These more important changes are easier to capture in reanalysis datasets and could explain the higher number of significant correlations found.

The clusters roughly coincide with a gradient from a sub-continental (cluster 2) to a more continental climate regime (cluster 1) and higher glacier elevations from the north-western to the central and south-eastern parts of the study region. Cluster 2 with medium temporal mass balance variability and average mass balances describes most of the outer orogene margins (Fig. 3) and glaciers within this cluster are spatially more heterogeneously distributed. This is the cluster with the lowest amount of significant correlations. In combination with the high variability within the datasets, our results suggest that glacier mass

balance variability does not result from the investigated meteorological drivers. In other words, this finding suggests that a





similar mass balance variability is a response to different drivers within this cluster. In contrast, anomalous high (cluster 3) and low (cluster 1) mass balance variability are better correlated. Cluster 1 and cluster 3 are spatially much more aggregated and the investigated meteorological and morphological parameters represent better the subregional mass balance variability.

What is responsible for the high variability within cluster 2 remains speculation in this data scarce region at the moment. It could be imprecise precipitation estimates at the glacier scale, resulting e.g. from the coarse resolution of reanalysis products, insufficiently represented localized precipitation events such as orographic precipitation effects (Roe et al., 2003), or different microclimatic settings. An interesting finding is the lower number of correlations when including snow cover for both mass balance estimates. This is especially pronounced for cluster 2 (Fig. 6), where a low fraction of variable combinations identifies snow cover in combination with $MB_{Barandunetal}$. This could indicate that a crucial meteorological component is missing in our analysis and that the T and P estimates are a surrogate for a process not related to snow dynamics. For these cases, glacier response might be more related to radiation, physical snow properties, or the amount of snow, which in total are better represented by simply using T and P, rather than the qualitative information of SC. In contrast, a similar number of correlations would indicate that T and P explain processes similar to SC.

We have not identified a strong seasonal importance of the meteorological variables in our spatial analysis (Fig. 5 and Fig. 6). Our results indicate a tendency of stronger summer and autumn influence along the northwestern margin and to some extend also for the negative "hot spot". This contrasts the autumn to spring importance for the positive mass balance anomalies in the Western Pamir and Central Tien Shan. We suggest that the latter might relate to changing snow cover dynamics and / or changes in solid-liquid precipitation ratios, supported by the high amount of significant correlations including precipitation. In contrast, the summer and autumn importance could be linked to either summer snowfall events or changing melt rates due to changes in air temperature.

## 5.2 Temporal Analysis

From our results, a very heterogeneous picture of dominant variables and seasons from 2000 to 2014(2018) becomes apparent, that is strongly dependent on the choice of datasets. Looking at an individual dataset might allow interpreting patterns backed-up by literature. At the same time the interpretations are contradicted by another set of data, which is similarly well supported by literature. In the following, we show two, somewhat contradicting but plausible interpretations about meteorological drivers for glacier mass balance changes based on the here presented temporal analysis. With this we aim to highlight the persistent difficulty to shed light onto the glacier-climate interaction in a heterogeneous and data sparse region such as Central Asia.

### 5.2.1 Analysis I: HAR dataset and $MB_{Barandunetal}$

All mass balances in this section are $MB_{Barandunetal}$ and all drivers refer to the identified HAR meteorological variables (limited to the 2000 to 2014 period) in our results. From the combined analysis of these data and previous studies, significant trends towards warmer annual air temperatures for large parts of Central Asia from 1950 to 2016 (Haag et al., 2019) seem to be reflected by the overall negative mass balance throughout the Tien Shan and Pamir. Air temperature is the dominant driver for most glaciers in the Tien Shan, most dominantly in spring and autumn (Fig. 7). This is possibly related to a strong increase



in temperatures by about 0.1 to 0.2° C per decade from 1960 to 2007 (Aizen et al., 1996; Kutuzov and Shahgedanova, 2009;
Kriegel et al., 2013).

A pronounced winter warming and an increase in winter and autumn precipitation in the northern and eastern parts of the Tien
Shan have been reported (Haag et al., 2019). Snow cover and air temperature in spring and summer are the dominant drivers
identified for the high mass loss in the Eastern Tien Shan and Dzungarsky Alatau (Fig. 7, 8). This might relate to a pronounced
snow cover decrease reported for the Eastern Tien Shan Notarnicola (2020), as a result of increased air temperatures causing
faster snow depletion in late spring and early summer. In line with that, Sakai and Fujita (2017) concluded that climatic settings
represented by the three factors (i) summer temperature, (ii) temperature range, and (iii) summer precipitation ratio, are the
most important factors for mass balance variability.

Precipitation trends for Central Asia are less significant and play a minor role for the mass balance evolution in Tien Shan.
Haag et al. (2019) reported a slight positive trend in summer, winter and autumn from precipitation anomaly time-series for the
period 1950 to 2016 for entire Central Asia. However, in our analysis most of these changes were not identified as individually
dominant drivers for the mass balances at the regional scale (Fig. 4). More locally, increasing spring and summer precipitation
in combination with an autumn and summer cooling in the Central Tien Shan (e.g. Li et al., 2022) align with the cluster
of slightly positive / close-to-zero mass balances. A localized temperature decrease over parts of the Central Tien Shan in
summer increases the frequency of solid precipitation events at the location of the positive "hot spot". Such summer snowfalls
can contribute to the mass gain and more importantly lower melt rates due to a positive albedo effect of fresh snow (e.g.,
Kronenberg et al., 2016). This is reflected by the importance of a changing snow cover and the seasonal importance ranging
from spring to autumn for the positive "hot spots" in the Central Tien Shan (Fig. 8, 7). This is in line with heterogeneous and
sharply contrasting snow cover changes reported by Notarnicola (2020), showing a positive snow cover change matching the
location of the positive mass balance anomaly in Barandun et al. (2021). The neighbouring negative mass balance anomaly in
the Central Tien Shan shows predominantly a spring and air temperature importance, where a sharp decrease in snow cover
duration was reported (Notarnicola, 2020). Hence elevated spring temperature enhances the snow depletion in the region,
favoring an early start of the ablation period.

Seasonal temperature trends are heterogeneous for the Pamir. Knoche et al. (2017) indicated a summer cooling for the
Northern Pamir but detected a warming trend for the Southern Pamir. This North to South gradient is reflected in the mass
balance of the glaciers located in the eastern part of the Western Pamir, driven mainly by air temperatures (Fig. 7).

Despite non-significant precipitation trends for the entire Pamir (Pohl et al., 2017), the Pamir-Alay received an increase in
winter precipitation since 1950 (Haag et al., 2019; Kronenberg et al., 2021). Winter precipitation changes, however, play a
subordinate role for the glacier mass balance of the Pamir-Alay, whereas spring and summer changes dominate (Fig. 7, 8). Fig.
7 and 8 also suggest that only glaciers with less negative mass balances show a relation with winter precipitation and that most
glaciers in the region relate to temperature, and more to the eastern part of the Pamir-Alay, also to snow cover changes.

The negative mass balance "hot spot" in the Western Pamir is dominated by winter and spring precipitation and snow cover.
At this location, where year-to-year mass balance variability is high, temperature seems to play a minor role. This, however,
changes for the positive "hot spot" in the Western Pamir, with low year-to-year variability, were the mass balance is clearly





controlled by the autumn temperature and snow covers (Fig. 7, 8). The positive and negative mass balance clusters in Western
Pamir agree well with decreasing and increasing snow cover fractions reported by Notarnicola (2020), respectively.

### 5.2.2    Analysis II: ERA5 dataset and MB$_{Hugonnetetal}$

The interpretation presented in section 5.2.1 is strongly contrasted by the following analysis (II), where mass balance data
are based on MB$_{Hugonnetetal}$ and all drivers refer to the identified ERA5 meteorological variables in our results. Air tem-
perature increased by about 0.1 to 0.2° C per decade in the Tien Shan during 1960 to 2007 (Aizen et al., 1996; Kutuzov and
Shahgedanova, 2009; Kriegel et al., 2013). Precipitation trends for Central Asia are less significant than temperature trends.
However, precipitation and snow cover are more important drivers than temperature for the mass balance of glaciers located in
the entire Tien Shan (Fig. 5, 7). Whilst in the southwestern part, spring is the most important season, winter is more important
in the northeastern part of Tien Shan, where glacier mass loss is especially pronounced (Fig. 8). In the Eastern Tien Shan, the
main driver for the negative mass balance seems to be precipitation and snow cover (Fig. 7). In contrast, for the Dzhungarsky
Alatau, the mass loss seems mainly driven by temperature and snow cover (Fig. 7).

At the western margin of the Central Tien Shan, less negative mass balances are in line with reported positive precipitation
changes (Aizen et al., 1996; Kutuzov and Shahgedanova, 2009; Kriegel et al., 2013). Haag et al. (2019) reported a slightly
positive trend in summer, winter and autumn precipitation for the Tien Shan that might influence the close-to-zero and slightly
positive mass balances at the southern margin of the Tien Shan. This is in agreement with the increase of snow cover fraction
reported in Notarnicola (2020). Therefore, the main driver seems to be precipitation and snow cover in spring and summer
(Fig. 7, 8).

For the Pamir-Alay, summer and autumn are the dominant seasons for the positive mass balances found at the western
margin of the subregion. Notarnicola (2020) reported a longer snow cover duration from 2000 to 2018. Thus, the observed
changes seem to be related to increased snow cover and decreasing temperatures during the transition from winter to summer,
allowing snow to persist longer, and earlier snowfall in autumn shortens the ablation season, acting favorable on the glacier
mass balances.

Haag et al. (2019) showed that despite a non-significant trend of annual precipitation amounts, summer precipitation had
significantly increased by around 5 mm per decade for the Western and Eastern Pamir. In combination with an important cooling
in summer reported for the nearby Karakorum during the past decadesFowler and Archer (2006); Mölg et al. (2014); Forsythe
et al. (2017), summer and early autumn precipitation becomes the most important driver for the positive mass balances in the
Eastern Pamir. The summer snowfall acts on the one hand directly as mass contributor and on the other hand lowers melt rates
due to a positive albedo effect (e.g. Kronenberg et al., 2016).

### 5.3    Implications from the spatial and temporal analysis

The resulting importance of the seasons and meteorological variables from the temporal and spatial analysis are strongly de-
pendent on the dataset, and suggest sometimes even a contradicting relationship with glacier response. In addition to that, clear
non-climatic drivers can not be identified. The interpretation and analysis is made more difficult because derived interpretations





(from Analysis I and II) align with other reported findings in the regions, as shown in Section 5. Unfortunately, available time series are also rather short to provide more robust evidence in the correlation analysis. This is due to a high meteorological and mass balance variability compared to rather slight trends and tendencies in these time series. Independent of the different

results, using different meteorological, and mass balance estimates, our study highlights in all cases a highly complex glacier response to climate variability and change. The current lack of sufficiently detailed and qualitatively satisfying data prevents elucidating the complex relationship between glacier mass balance and especially climatic drivers in the Tien Shan and Pamir.

At the current stage of research, we are not able to rate the different, partly contrasting products. Due to the lack of regional-wide systematic cryospheric and atmospheric monitoring, validation of these datasets for scientific applications is strongly

limited. This results in an elevated uncertainty for mass balance modelling and interpretation on the underlying processes of the glacier response to climate change. Highlighted with the two scenarios in Section 5, any derived understanding of the glacier-climate interactions depends strongly on the dataset used. The limitations of these datasets cascade to an increase of uncertainties when a thorough quality assessment of the different reanalysis products for further application cannot be assured. This concerns in particular the modelling of future glacier response, hydrological modelling and the equifinality problem

related to too many unknowns in the cryo-hydrological cycle (e.g. Beven, 2006; Farinotti et al., 2015), or reconstruction of glacier mass change. Many studies are typically based on a single reanalysis dataset, either used for bias correction or as model input directly. Direct calibration data are generally limited to a few individual location, which is insufficient for regional applications.

## 6   Conclusions

The extreme scarcity of in situ observations for both meteorological variables and glacier mass balance in Central Asia leaves much space for different interpretations on how glaciers may evolve in this region within a changing-climate scenario. As a result, water availability assessment for the growing population is uncertain. Our study shows in particular that even supposedly similar datasets such as ERA5 and its derivative CHELSA lead to different and partly contradicting assumptions on drivers for mass balance variability. Ease of use and great availability of datasets such as ERA5 might lead to a one-sided use of certain

datasets in disfavor of those less comprehensive but more thoroughly validated ones such as HAR v1.4, whose shortcomings are better known of. Our study points to a trend in which apparent but false consistencies across studies using a single dataset might largely relate to the chosen dataset rather than to the processes or involved environmental variables. We accordingly showcase how determined important variables change with the use of the two different mass balance datasets even though said datasets agree at regional scale: differences at individual glacier scale are stark. Obviously, regionally aggregated mass

balance estimates are useful for summarizing and reporting results, e.g., following the HiMAP classification or our glacier mass balance ($\text{MB}_{Barandunetal}$) standard deviation-based k-means clusters. However, the downside is that derived interpretations about important drivers carry a large subjective and arbitrary aspect as these interpretations will change significantly based on the chosen clustering. This effect is largely remediated in the temporal analysis where each glacier–meteorological relationship





is preserved. Such analysis, however, requires mass balance time series that geodetic methods cannot provide by default and
with the trade-off of increasing uncertainty at short time scales.

In summary, we find that the aspects from highest to lowest impact on deriving conclusive results are:

– Differences in meteorological data

– Differences in mass balance data

– Regional classifications and aggregations

Finally, for the present work, we were completely unable to arrive to a conclusion on the driving meteorological and morphological variables for mass balance variability in Tien Shan and Pamir. In this data-scarce region, where meteorological or mass balance datasets cannot be rigorously validated with ground truth data, we believe the only honest option from a scientific point of view is to "suggest" rather than "state" the existence of found relationships and inferred dependencies.

*Code and data availability.* The meteorological data for HAR (Maussion et al., 2011) can be obtained from the Technical University of
Berlin via https://data.klima.tu-berlin.de/HAR/V1/[Last access:2022-06-09], CHELSA data (Karger et al., 2021) can be obtained via https://doi.org/10.16904/envidat.228.v2.1[last access:2022-06-09], and ERA5 data (Hersbach et al., 2020) via the Copernicus Climate Data Store (Hersbach et al.)[Last access:2022-06-09].

The mass balance data MB$_{Barandunetal}$ is published in Barandun et al. (2021) and annual mass balance time series are provided via zenodo open-access repository https://doi.org/10.5281/zenodo.4782116[Last access:2022-06-09]. The mass balance data MB$_{Hugonnetetal}$
is published in Hugonnet et al. (2021) and annual mass balance are publicly available at https://doi.org/10.6096/13[Last access:2022-06-09].

Debris cover information (Scherler et al., 2018) was obtained via http://doi.org/10.5880/GFZ.3.3.2018.005[Last access: 2022-06-09]. Surge type glaciers inventory from Guillet et al. (2022) is available at https://doi.org/10.5281/zenodo.5524861 [Last access:2022-06-09]. Glacier outlines and glacier morphological characteristics have been taken from the Randolph glacier inventory accessible here https://www.glims.org/RGI/[Last access:2022-06-09].
The code to reproduce the results can be found on github (https://github.com/pohleric/barandun_pohl_tsl-correl). The required main input files, containing the morphological and meteorological data are deposited on a zenodo repository (https://doi.org/10.5281/zenodo.6631963).

*Author contributions.* Both authors have contributed equally to the manuscript.

*Competing interests.* We declare no competing interest.

*Acknowledgements.* This study is supported by the Swiss National Science Foundation (SNSF) grant 200021_155903 and the CICADA
(Cryospheric Climate Services for improved Adaptation; contract no. 81049674) project funded by the Swiss Agency for Development and



the University of Fribourg and the project Cryospheric Observation and Modelling for Improved Adaptation in Central Asia (CROMO-ADAPT), contract no. 81072443, between Swiss Agency for Development and Cooperation and the University of Fribourg. This project received funding from the Swiss Polar Institute (project number SPI-FLAG-2021-001). This study is supported by Snowline4DailyWater. The project Snowline4DailyWater has received funding from the Autonomous Province of Bozen/Bolzano – Department for Innovation,

Research and University in the frame of the Seal of Excellence Programme. We thank Christian Hauck and Martin Hoelzle for their feedback on an early version of the manuscript.



**Figure A1.** Coefficient of determination ($R^2$) for variables in the spatial analysis. Shown $R^2$ is the average for all variable combinations in which the displayed meteorological variable (SC, P, T) is contained. Mass balance estimates of Barandun et al. (2021) (left) and Hugonnet et al. (2021) (right).





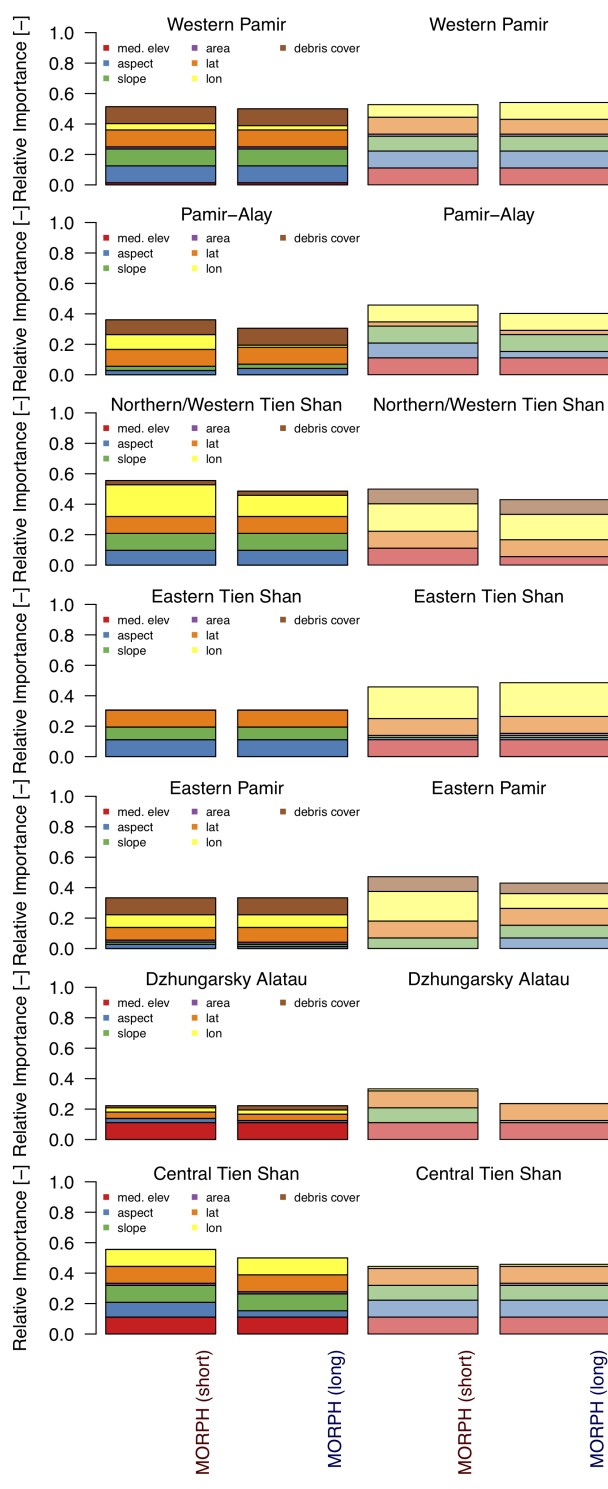

**Figure A2.** Frequency of significant morphological variables for the HiMAP regions and for the two mass balance estimates by Barandun et al. (2021) (left) and Hugonnet et al. (2021) (right). .



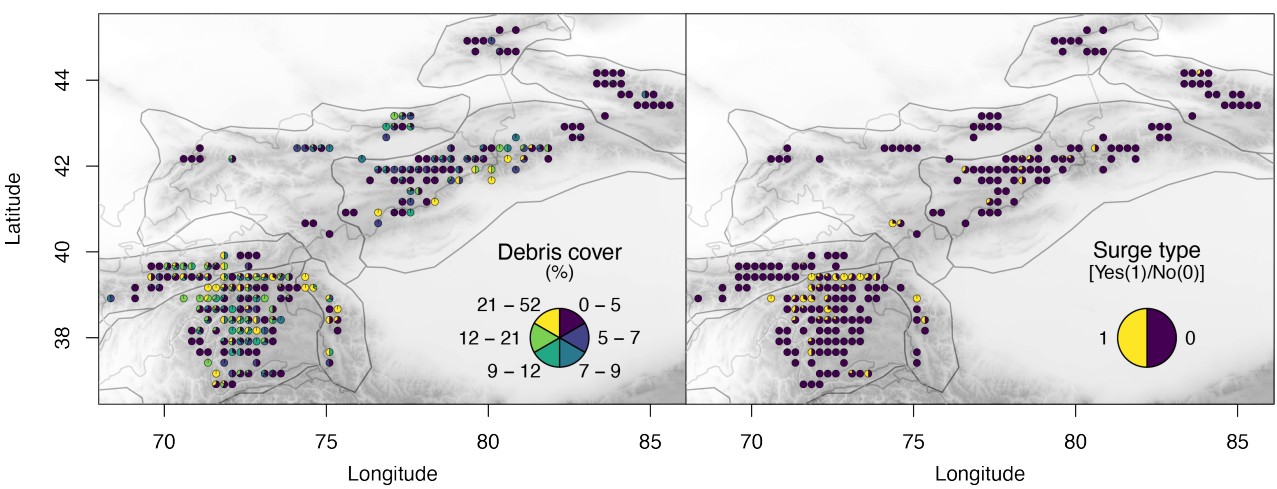

**Figure A3.** Surge type glacier (Guillet et al., 2022) and debris cover (Scherler et al., 2018) distribution in the Tien Shan and Pamir

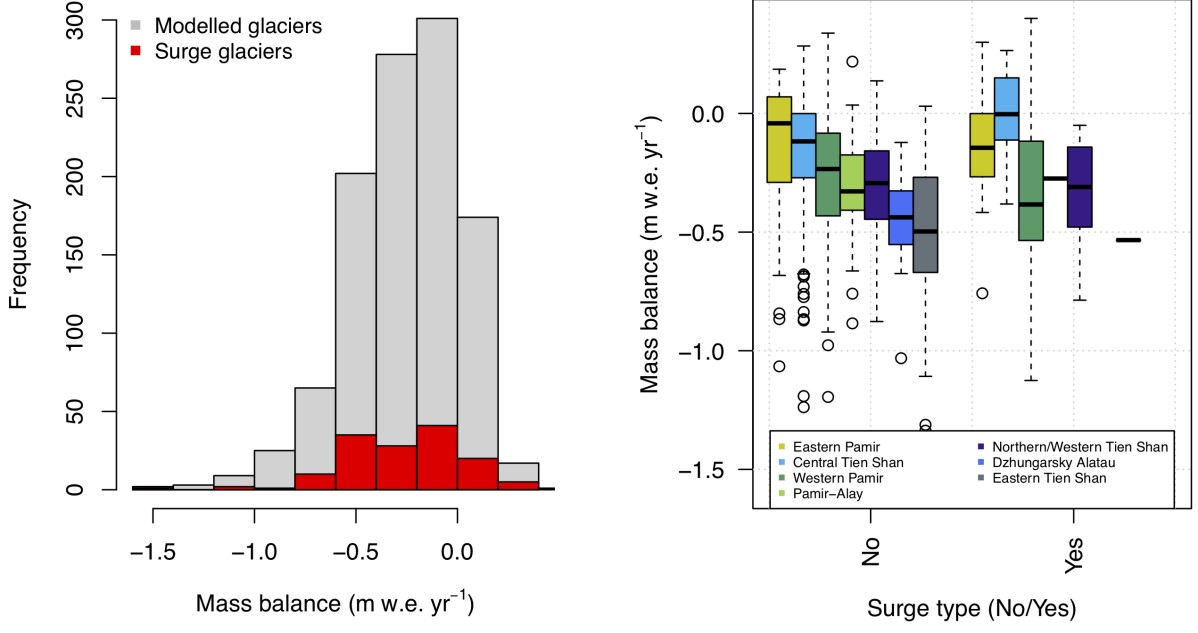

**Figure A4.** Distribution of surge type glaciers and mass balance range of all glaciers (left) and for the HiMAP regions (right).





**Figure A5.** Coefficient of determination ($R^2$) of temporal analysis between annual mass balance estimates from Barandun et al. (2021) and Hugonnet et al. (2021) and two meteorological variables from the different reanalysis datasets as indicated in the figure legends. Upper two panels without, and lower two panels including snow cover.



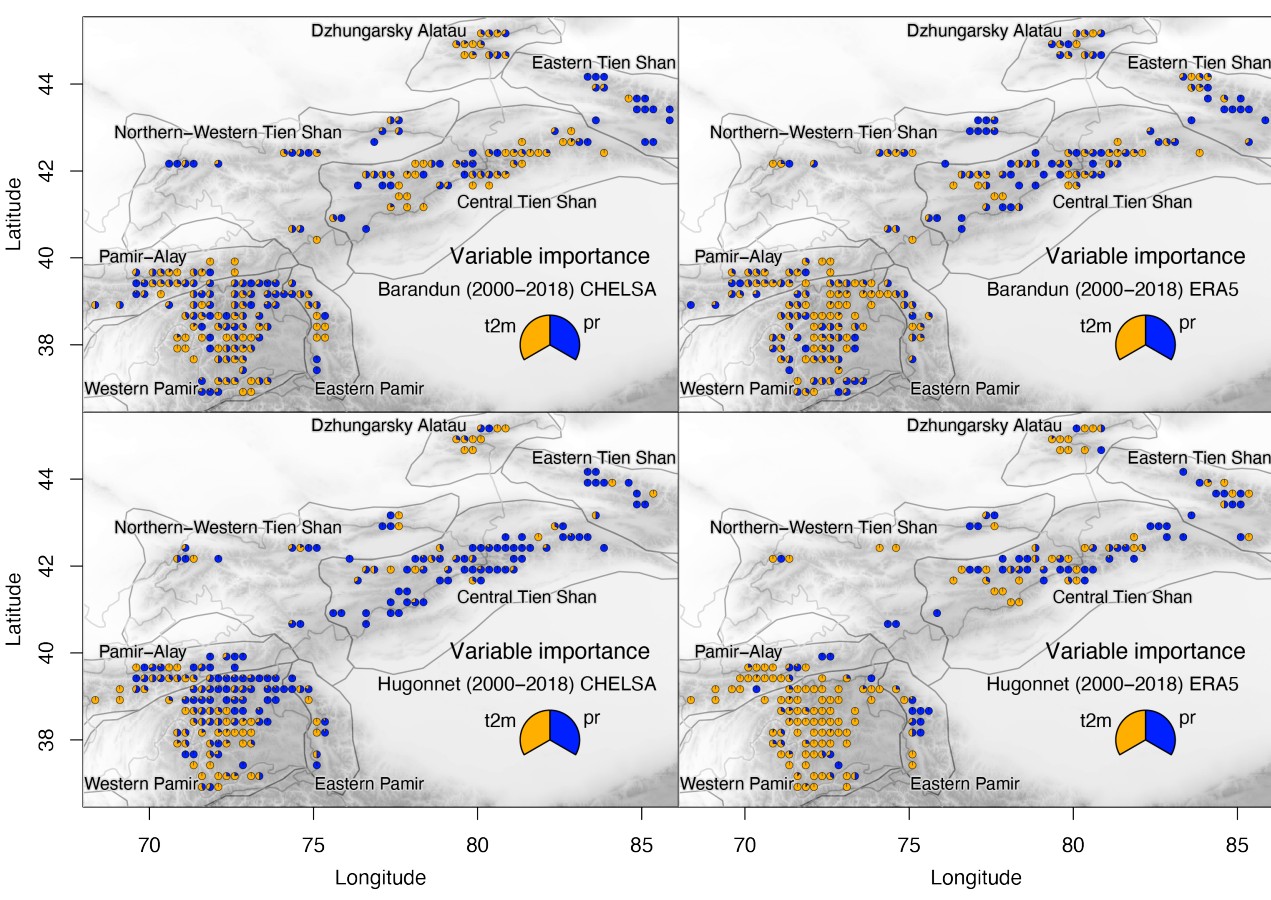

**Figure A6.** Importance of meteorological variables for temporal analysis between annual mass balance estimates from Barandun et al. (2021)(top) and Hugonnet et al. (2021)(bottom), and two meteorological variables (T and P) from the different reanalysis datasets.



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
