# Peer review of "Central Asia's spatiotemporal glacier response ambiguity due to data inconsistencies and regional simplifications"

_The Cryosphere, 2022_

## Referee Comment (RC1)

**Review of *'Central Asia's spatiotemporal glacier response ambiguity due to data inconsistencies and regional simplifications'* by Barandun and Pohl, 2022**

This study by Barandun and Pohl (2022) explores the links between climate re-analysis datasets and glacier mass balance in Central Asia. The authors use annual mass balance data of individual glaciers in the Pamir and Tien Shan extracted from the datasets by Hugonnet et al. (2021) and Barandun et al. (2021). They link the temporal and spatial patterns with temperature and precipitation from 1/ three different reanalysis products (ERA5, CHELSA and HAR), as well as the snow cover product from MODIS and 2/ morphological attributes of the different glaciers using multiple variable regression. The authors show that the correlations and their interpretations can change considerably depending on which mass balance and reanalysis products are used, thus highlighting the limitations of such an approach in a data scarce region.

This study highlights some very important limitations of the analysis of regional datasets in a particularly important region for the study of water resources. I commend the authors for their scientific rigor that led them to explore such a problematic. This 'non-result' is actually a valuable demonstration that calls for more work on the controls of glacier mass balance in Central Asia. However, I have a number of comments that would need to be addressed for this manuscript to be considered for publication.

**Major comments**

**Direct comparison of datasets:** Looking at how the different reanalysis datasets correlate with the glacier mass balance datasets is interesting but I feel that it adds quite some complexity in the interpretation. I am missing in this manuscript a separate comparison of all mass balance products together and all climatic data together. This would likely help understand the multiple linear regressions better.

**Barandun et al. (2021) mass balance:** It strikes me that this mass balance model is based on ERA-interim data, which is also a reanalysis product similar to ERA5. It has been calibrated with snowlines and geodetic mass balance, but I suspect the results are likely influenced by the climate data as well. Have the authors compared the ERA-interim with the ERA5 to look for possible changes? This is likely to influence the regressions and result in some circularity – do the regressions agree or not with the results of Barandun et al. (2021)?

**Uncertainties in mass balance data:** I am concerned about the use of yearly glacier mass balance, especially as it is not clear to me how representative of the actual mass balance. This is relatively well explained for the Barandun et al. (2021) dataset, but less for the Hugonnet et al. (2021) – are these actual geodetic measurements made on a yearly basis or are they extracted from the general trend? In either case, I expect the uncertainties on this data to be quite high relative to the glacier mass balance values, especially for glaciers in Central Asia, which tend to not lose mass very quickly. This seems to be confirmed by the fact that the Western regions have higher significant correlation frequency (also the ones with clearer mass balance signal). I am therefore wondering how valid it is to take yearly data and whether taking decadal trends would not be a better avenue to analyze the spatio-temporal patterns. At the very least a discussion about these uncertainties would be necessary to include in the manuscript.

**Downscaling of the reanalysis data:** I was surprised to see that the different reanalysis datasets had not been downscaled, especially considering that their respective resolution varies a lot. Is there not a risk that this will introduce elevation biases between subregions? Why has this not been considered in the study?

**Dependence of snow cover on temperature and precipitation:** Does the fact that snow cover is dependent on temperature and precipitation not affect the regressions?

**Summarizing the scenarios in the discussion:** These potential scenarios are interesting but lengthy and difficult to follow for readers not familiar with the particularities of the region. Could these be synthetized in a figure and streamlined? A short summary of the main differences would also be welcome.

**Moving forward:** This is very briefly mentioned in the abstract only (as far as I can tell). I was a bit frustrated that there were not more discussions on this – how should one then proceed to interpret the mass balance patterns? What possible other tools could be used, what additional data should be collected?

**Minor comments**

**Abstract**

Overall, I find that the abstract could be streamlined and the main message made clearer.

L3-4: 'Meteorological analysis, remote sensing products and novel approaches … all provide …'

L9: 'only … do we find'

L13-14: This feels like a repeat from above

L16-18: this part is barely mentioned in the discussion and could be developed more.

**Introduction**

In general, the introduction is interesting and well written but I think it would benefit from some reorganization efforts and some streamlining to make the message clearer.

L25-26: This sentence does not bring much and could be removed. Are the two references to Gerlitz et al., 2019, 2020 really needed here?

L32-34: This feels out of place

L20-36: There are lots of ideas in this first paragraph but the logical links are missing. These need to be reorganized/structured.

L46: 'Barandun et al. (2021) have applied'

L44-48: I am not sure that many details are necessary here, especially as these are described in length in the methods.

L59-60: Are these details really necessary here? A simple reference to Hugonnet et al. (2021) is likely enough

L61: something wrong with the English at the end of this sentence

L63: There are actually some region-wide debris thicknesses assessments. See Rounce et al. (2021) and McCarthy et al. (2022)

Rounce, D.R., Hock, R., McNabb, R.W., Millan, R., Sommer, C., Braun, M.H., Malz, P., Maussion, F., Mouginot, J., Seehaus, T.C., Shean, D.E., 2021. Distributed global debris thickness estimates reveal debris significantly impacts glacier mass balance. Geophys. Res. Lett. e2020GL091311. https://doi.org/10.1029/2020GL091311

McCarthy, M., Miles, E., Kneib, M., Buri, P., Fugger, S., Pellicciotti, F., 2022. Supraglacial debris thickness and supply rate in High-Mountain Asia. https://doi.org/10.31223/X5WW5B

L67: references missing. A recent one could be the work by Glasser et al. (2022) - 10.1016/j.geomorph.2022.108291

L67-69: Not sure these details are necessary here

L69: Have the authors considered avalanching as a possible morphological control? It feels like for some of the steeper ranges of the region that could actually play a significant role (Brun et al., 2019)?

**Data**

L87: Study site should be plural. No need to capitalize the nouns in the titles.

L121: I don't think this acronym has been defined before in the main text.

L124: Simple reference to RGI v6.0 is enough here (text can be shortened)

L134: The Barandun et al. (2021) dataset also uses geodetic mass balance products to calibrate their model (second order calibration).

L135: Suggest adding reference to Figure 1 here.

L134-159: Are all the details provided here really needed considering that these are already published approaches?

L141: (Dee et al., 2021) should come after 'data'

L148: 'observational' is not really true for Barandun et al. (2021) MB, as it comes from modeling.

L155: can you explain a bit better the sentence 'local and regional scale biases can persist'? A reference might be needed here.

L181: Could you also provide the resolution in km for consistency with the other datasets?

L205-209: I would recommend putting this in a separate subsection.

L240-241: I struggle a bit with this k-mean clustering. Could you give a few more details?

**Results**

L276: A3 should come before A4 in the text

**Discussion**

L357: remove the comma

L359-360: this makes sense as they are likely related

L434: snow cover decrease reported by

L489-490: missing parenthesis

**Conclusion**

L543-544: Use lower cases for Barandunetal and Hugonnetetal

**Figures**

The panels of the different figures need to be numbered.

Figure 1

- Instead of a globe, a map of HMA would be sufficient and more informative here.
- Shouldn't the mass balance should be in m.w.e? Also for consistency with the text.

- In general I don't really like the term 'surface' mass balance here – at least for Hugonnet et al., 2021, these are geodetic mass balance measurements. I would suggest sticking to 'glacier' mass balance.

Figure 2

- Why are the spatial and temporal analysis linked in the figure? Aren't they done independently from one another?
- It would be good to distinguish the data boxes and the methods boxes. Having the seasonal aggregation in the same box as the monthly meteorological time-series is confusing

Figure 3

- Could you specify what each cluster corresponds to in the figure? There are likely many ways of clustering this data, which criteria were used here and why?

Figure 4

Most of the comments here hold for the next figures as well:
- Suggest writing out SC, P, T
- Do these meteorological correlations relate then to trend, mean, STD?
- Which variables do the seasons refer to? This would need to be specified in the caption, and maybe even in a supplementary figure?
- Specify in the caption that 'short' and 'long' refer to the 2000-2014 and 2000-2018 periods
- It needs to be specified also that this is an aggregation of all glaciers in the region

Figure 7

These comments also hold for figure 8:
- Stay consistent with acronyms in figures
- The explanatory diagram does not need to be repeated in every subplot if it stays the same
- Where are the subplots with and without snow cover? I cannot find the legend.

Figure A4

Would there not be a way to weight the results of the right panel by the number of glaciers?

---

## Referee Comment (RC2)

**Review of Barandun and Pohl (tc-2022-117): "Central Asia's spatiotemporal glacier response ambiguity due to data inconsistencies and regional simplifications" – The Cryosphere Discussions**

**General Comments**

The manuscript by Barandun and Pohl highlights the inconsistencies of glacier mass balance responses to different meteorological and morphological drivers in Central Asia's high mountain regions. The authors provide a systematic, region-wide spatial and temporal analysis of two glacier mass balance products against three reanalysis/gridded atmospheric products and open source spatial data on glacier characteristics to this end. The authors find that inconsistencies in glacier response were evident across different glacier or regional/mountain sub-divisions, suggesting that extreme care should be given to statements regarding the main drivers in such a region with limited in-situ validation data.

I think that the authors present a valuable perspective on a common problem in glacier response attribution, specifically applied to a sparsely monitored and highly heterogeneous region, where such drivers are used to theorize about past and future patterns of glacier change in the absence of widespread monitoring. I think that the paper is generally well written and I particularly liked the approach to include specific case studies (Analysis I + II) of the discussion section. Unfortunately, I feel that the comparison of the different reanalysis products and the resultant temporal analyses are not overly convincing or perhaps not robust enough in places. I think the manuscript can be of sufficient quality to be published in The Cryosphere, but a few elements certainly need to be addressed first.

- I think the authors need to restructure their objectives to be something they more clearly address: Objective 1 seems to encompass two separate research questions about the drivers and physical processes themselves, and then the degree to which this is simply a result of which data are used to explore the problem. Objective 2 is related to the latter part of the first objective in my opinion and additionally implies that the authors explicitly compared ground truth data for correlation analysis. I'm not convinced that the authors really test enough the limitations of the gridded data in this manuscript (so objective 1, part 1), or at least not from the data/figures presented.

- It is clear that different reanalysis datasets will produce a different correlation with the same mass balance product. While the authors perform a nice assessment of these expected differences, I'm left wondering how comparable the forcings really are in the first place. The authors apply ERA5 (approx.. 36km), HAR30 (30km) and CHELSA (1km, using ERA5 to downscale). While the first two are closer in terms of spatial resolution and the processes they can represent at the surface, CHELSA represents a major effort to include finer scale topographical features, such as windward/leeward precipitation adjustments, even though it likely remains with significant seasonal biases. Accordingly, I would not expect to find a similar relationship to glacier mass balance (spatially or temporally) when comparing to ERA5 or HAR30 (e.g. L286-7, L343-4), which cannot represent precipitation dynamics at the scales relevant to glacier processes (e.g. L401). The authors have given some consideration to these limitations in their discussion section, but I'm not convinced that it is a fair test and fully supports the conclusions about an ambiguity to glacier response in the region.
I think that redoing analysis with additional datasets would also not necessarily advance the findings of this manuscript and perhaps cloud the main messages. However, I do think that the authors need to address these discrepancies and perhaps even remove CHELSA from the analysis. More importantly, I would like to see some level of comparison between the different products for air temperature and precipitation estimates (especially if CHELSA

remains included). What explains the presence or lack of correlations in given subregions, for example? Some supplementary figure(s) would be useful to that end.

- Building on the previous point, I think the authors also need to address any potential circular issues related to the use of ERA-Interim for the derivation of the $MB_{Baradunetal}$ mass balance results and its comparison to the next generation of the ECMWF reanalysis.

- The temporal analyses produce clearer and more interpretable plots for this manuscript (which I liked), but I have concerns about the robustness of these results given their short temporal scale (14 years for correlations using HAR30) and high uncertainties for mass balances given annual data series. While the ideal minimum sample size required can vary based on the type of observations, it has generally been considered that sample sizes less than 30 can produce imprecise correlation estimates. I'm curious why the authors did not consider a few longer term reanalysis/gridded products of similar spatial resolution (e.g. HARv2, ERA5Land, WRF9km) that all provide data since 1980 at least. I guess there are also temporal limitations for the mass balance data period, but this should at least be discussed more clearly.

- Plotting such large datasets can be challenging and I think the authors have done a great job in many respects. I think there are several areas where they might still be improved for clarity and to help the message of the manuscript. I have given some specific comments below.

- As mentioned, the work is generally well written. However, there are several instances where the wording is unclear or with grammatical errors. I have tried to provide some examples of this below, but the authors should carefully check the entire manuscript at the next submission.

**Specific Comments**

L90+ :  I think that the authors provide some very valuable and interesting information about the climatic setting in this paragraph, but I do not feel that the discussion reflects enough upon these interesting variations. Building upon my main point about comparisons of the gridded datasets, the authors should present spatial maps of mean (summed) temperature (precipitation) from different products, or better yet, the spatial trends of some of these variables (where are they statistically significant for ERA5/HAR30?).  Exact comparisons can be challenging due to differences in spatial scale, but the authors could consider spatially binning the grids as presented, for example, in de Kok et al. (2020).

L185-188: The authors state that snow depth over-estimation in ERA5 renders use of radiation variables "problematic", and rely upon only temperature and precipitation data. However, there are well reported cold biases for air temperature at high elevation regions (such as those in this manuscript) which also result from this albedo-effect (e.g. Wang et al. 2020).

L192: Given that the authors test the latest products from ECMWF (ERA5) and CHELSA, I'm surprised that they did not also consider the HARv2 product which is openly accessible and available back to 1980 (thus eliminating the shorter temporal focus and affecting the strength of correlations).  I think a line should be added somewhere to justify the use of these three datasets specifically and why they are considered comparable, especially given the points from my main comment above.

L205: I'm not convinced that the MOD10CM product at a 5km resolution does much to aid the analysis presented here.  Given its spatial resolution, can it reasonably represent anything meaningful at the scale of the glaciers? Please add some additional information.

L215-216 (and 243-245): I do not understand clearly how the authors relate the spatial means of mass balance with trends of meteorological variables. From the trend you will have just one value per gridded pixel, correct? So the authors combine all glaciers in a given binned region (circles in Figure 1) or just the entire region, and regress all the trends in a given variable (say temperature) at the corresponding pixel of each glacier against the mean or STD of mass balance at that glacier? I believe the authors need to more clearly explain this. Unfortunately, Figure 2 does not help much with this interpretation.

L221-222: Please re-write this sentence to improve syntax. It makes sense, but is not well written.

L225-226: This is also a little unclear from the writing. Do the authors mean that they do not mix variables from different products in their analyses (e.g. temperature from ERA5 and precipitation from CHELSA)?

L242: Please redefine here what is meant by these 'hotspots'. I note that this comes from the author's previous published work, but it would be better to clarify explicitly for the reader what is meant by a hotspot of mass balance variability. I interpret that as the areas in Figure 1 where STD is greatest, for example.

L253: The authors consider the ambiguity of the glacier response here as being largely due to different meteorological or mass balance datasets considered, but what about the uncertainties stemming from the annual-series of geodetic mass balances themselves, which are, to the authors own admission (e.g. L156), with less accuracy, and I assume, less precise? Again, it would be ideal to be able to assess the comparison of the different gridded meteorological and mass balance products independently to identify where the largest differences in the datasets lie, not just the correlations/significance.

L264: I think the authors should attempt to signpost some of the key findings a little better in the figures and simplify some of the main figures where possible (see specific figure comments below). I find it a little hard to navigate from text to figure to follow the storyline.

L344: The authors do not consider all available reanalysis datasets or atmospheric data for their analysis. Please rephrase this.

L346: "at the local scale…" Please check the text for many of these small grammatical issues or minor mistakes.

L346-8: This sentence does not make sense to me. Please rephrase to make your point clear.

L350: but they do not prevent this analysis here, the authors have an annual mass balance value for temporal analysis. I think this first paragraph needs to be restructured slightly to highlight what the authors are able to show, which is normally not considered/available.

L369-379: I think the authors should rewrite this segment as their point is not particularly clear upon reading it. Do the authors mean that their HiMAP regions encompass multiple sub-climatic regions and associated atmospheric processes which therefore limits the usefulness of these defined regions for such analyses?

L382: Please define what "rotation" of mass balance gradients means.

L385: reword "already slight uncertainties…" It does not make sense. Small uncertainties? What estimates? Which variables?

L387: How "invisible"? Are simply not resolved by the coarse reanalysis grid scale?

L388-9: Not clear. Do the authors suggest that coarse grid scale products can capture changes in these regions because there is less sub-grid variability in topography and its associated meteorological complexity?

L392: What do the authors mean with "outer orogene margins"?

L399: "remains unclear…"

L406: please provide examples of which physical snow properties that the authors refer to and what variables missing from the analysis might influence the results.

L410: "extent"

L434: Citation requires parenthesis, check formatting.

L489: Check formatting.

L497-8: Related to the general comments, 14 years does not produce an ideal correlation period. Some discussion on this is needed.

L503: The authors cannot evaluate the biases and patterns of reanalysis datasets due to lack of ground data, but should still highlight how the products relate to each other. Although long term in situ observations are rare, and I understand that it is not the goal of the manuscript to identify the 'best' product to use, but I would be interested to see how the different products compare to even short term measurements from sporadic AWS measurements familiar to the authors from previous work (e.g. Abramov Glacier , Golubin Glacier).

**Figures**

Figure 1: The inset map in the upper right panel should ideally be of High Mountain Asia to provide a better spatial context as to where the study site is. If able, the authors should attempt to neaten the location and intersect of the legend numbering as there is overlap in places.

Figure 4: Please clarify in the caption how variable importance is defined and why this is given as a relative freq. in the plot labels. Also ideally provide indices (a,b,c etc) to aid navigation for the reader from the main text. I don't find a 'short' analysis period as meaningfully different to the long one for the lower panels. Such morphological characteristics (e.g. debris) won't greatly change over 4 years. Consider removing it to streamline and simplify the figure.
In general I find it very hard to interpret the information given in these figures (4,5,6, A1, A2). Any attempt to simplify it or remove parts would aid interpretation.

Figure 5: I think this figure has far too much information and becomes difficult to interpret. Please consider streamlining the most crucial information where possible, perhaps just showing the short periods, or those with MOD10. For upper panels, one has to struggle to align the x axis label and interpret what is the take-home information. legends should be increased in size, but could easily be kept for just one panel for the whole left and right side.

Figure 6: Similar considerations to the above.

Figure 7: These plots are more clear than the previous ones. Perhaps the authors can re-structure the panels to be 3 rows and two columns, so that the size on the page can be enhanced and the reader can more easily interpret the small circles. Why using 'M' in front of the meteorological datasets? I think it would be clearer without. I do not understand from the figure caption the "with or without snow

cover…" as all show scf as a variable importance coloured on the map. Please clarify or correct this. Please use indices for the subplots to help the reader. For Figure 8 as well.

References

de Kok, R. J., Kraaijenbrink, P. D. A., Tuinenburg, O. A., Bonekamp, P. N. J., & Immerzeel, W. W. (2020). Towards understanding the pattern of glacier mass balances in High Mountain Asia using regional climatic modelling. The Cryosphere, 14(9), 3215–3234. https://doi.org/10.5194/tc-14-3215-2020

Wang, X., Tolksdorf, V., Otto, M., & Scherer, D. (2020). WRF-based dynamical downscaling of ERA5 reanalysis data for High Mountain Asia: Towards a new version of the High Asia Refined analysis. International Journal of Climatology, May, 1–20. https://doi.org/10.1002/joc.6686

---

## Author Comment (AC1)

**Author Comments**

**Central Asia's spatiotemporal glacier response ambiguity due to data inconsistencies and regional simplifications.**

Martina Barandun and Eric Pohl

We would like to thank the two reviewers for the positive feedback and the constructive comments on the submitted manuscript. We agree on most of the critique and provide in the following our suggestions on how to implement the changes to the manuscript for all the issues pointed out. As several issues were raised by both reviewers we post this combined response identically to both RCs. In cases where we disagree, we present our reasoning and arguments, which we would then implement in an edited version. In order to further improve the clarity, we would also make use of a professional proofreader if the proposed changes should satisfactorily resolve the current issues with the manuscript.

Below, we respond to all comments, and state how we plan to account for them in the revised version of the manuscript. The responses (normal font style) to the reviewers' comments are written directly after the reviewer comments (displayed in italic font style).

**1 Reviewer #1**

**1.1 Major revision**

**Direct comparison of datasets:** Looking at how the different reanalysis datasets correlate with the glacier mass balance datasets is interesting but I feel that it adds quite some complexity in the interpretation. I am missing in this manuscript a separate comparison of all mass balance products together and all climatic data together. This would likely help understand the multiple linear regressions better.

We will add a comparison between the different mass balance and reanalysis products. See also answer to comment of Reviewer 2. We follow the design by de Kok et al. (2020) for the comparison of reanalysis products, and add a map in the style of the distributed pie chart maps for the mass balance comparison.

Regarding a comparison with meteorological station data, we do not think this is helpful for the present manuscript. There are multiple studies that have investigated new data products for precipitation. There is, however, no real baseline to compare any product against because most of the few available meteorological stations are located in valleys and thus there is limited to no ground truthing at higher elevation. We can see the motivation for the reviewers' request. Reviewer two has pointed also out that the stated objectives should be adjusted to better match the presented results and discussion. We agree on this and adjust the objective description to more accurately match the presented work. Regarding the presentation of meteorological data, we think that the overview in the style of de Kok et al. (2020) is providing the important information on spatially different estimates by the products. The issue of missing ground truthing data is referenced in the text. We are only planning to provide the comparisons of the annual data for temperature and precipitation, instead of producing two (variables) times five multi-panel figures as in de Kok et al. (2020). For the main text, we are planning to include the precipitation comparison and refer to the Annex/Supplementary for the temperature comparison. That is because these figures occupy almost a page.

Regarding the mass balance data, we show an overview map in the same style as for all other presented maps with pie charts per 25 km by 25 km regions to make differences easier to spot with the eye. The strongest differences can be seen at the short period and we would include this figure in the main text. We would provide the map for the long period in the Supplementary.

**Barandun et al.** (2021) mass balance: It strikes me that this mass balance model is based on ERA-interim data, which is also a reanalysis product similar to ERA5. It has been calibrated with snowlines and geodetic mass balance, but I suspect the results are likely influenced by the climate data as well. Have the authors compared the ERA-interim with the ERA5 to look for possible changes? This is likely to influence the regressions and result in some circularity – do the regressions agree or not with the results of Barandun et al. (2021)?

This point has been raised by both reviewers and we understand their concern. However, we believe the use of the two datasets is justified in the way they have been used. The approach presented in Barandun et al. (2021) fortunately does not rely on precise meteorological input

(which simply does not exist extensively in the high mountains of Central Asia). Barandun et al. (2018) showed that using transient snowlines for model calibration reduces the sensitivity to the meteorological input data. The sensitivity analysis in Barandun et al. (2018) uses an average temperature and precipitation dataset without any year-to-year variability. Obtained mass balances using the transient snowline approach still provided results with an RMSE of less than 0.2 m w.e.  $yr^{-1}$  in comparison with the direct measurements. This highlights that the snowline observations used for calibration, for each glacier and year individually, are primarily responsible for the modelled mass balances. For more details, we believe it is sufficient to refer to the previous work Barandun et al. (2018) that provides an in-depth analysis of the model sensitivity which the current manuscript builds upon.

In Barandun et al. (2018) ERA-interim was used becauseOrsolini et al. (2019) showed the superior performance of ERA-interim for mountainous regions like High Mountain Asia over ERA5, mainly due to the assimilation of *in situ* station data in ERA-interim that is omitted in ERA5. The products have thus significant difference in the generated precipitation fields that are independent from each other. This provides the opportunity for further analysis of possible correlations between the mass balance time series and the climatic drivers with ERA5 or other independent reanalysis datasets. Currently ERA5 is one of the most used reanalysis products due to its superior spatial resolution. However, often the above outlined shortcomings are not considered in correlation analyses. Despite the low model sensitivity we did not to use ERA-interim for the correlation analysis. We are planning to add a justification of the dataset in the manuscript, explaining also the insensitivity of the mass balance time series to the meteorological input data.

Uncertainties in mass balance data: I am concerned about the use of yearly glacier mass balance, especially as it is not clear to me how representative of the actual mass balance. This is relatively well explained for the Barandun et al. (2021) dataset, but less for the Hugonnet et al. (2021) – are these actual geodetic measurements made on a yearly basis or are they extracted from the general trend? In either case, I expect the uncertainties on this data to be quite high relative to the glacier mass balance values, especially for glaciers in Central Asia, which tend to not lose mass very quickly. This seems to be confirmed by the fact that the Western regions have higher significant correlation frequency (also the ones with clearer mass balance signal). I am therefore wondering how valid it is to take yearly data and whether taking decadal trends would not be a better avenue to analyze the spatio-temporal patterns. At the very least a discussion about these uncertainties would be necessary to include in the manuscript.

This is completely true. The idea behind using more then one mass balance time series is to show the uncertainties not just in the reanalysis datasets but also in the mass balance time series currently available. Hugonnet et al., also point out that the data should not really be used in an annual study – even though they provide the values. The annual values are mass conserving with respect to the long-term calculated geodetic values and thus the provided mean annual values are usable. Our reasoning for using the annual estimates anyways is that it is so far the only other mass balance dataset at annual resolution available in the region. We will point this out more clearly in the reworked manuscript.

Both mass balance time series use geodetic estimates calculated from the same dataset (ASTER) but use two different processing approaches, leading to very different glacier-specific mass balance estimates. This highlights the uncertainties in geodetic methods and how these relate into uncertainties further down the processing chain.

We believe the case study in Barandun et al. (2018) provides an extensive assessment of the model sensitivity and uncertainty and eventually provides a very conservative estimate. Barandun et al. (2021) adopted the mean uncertainties provided in Barandun et al. (2018) ( $\pm 0.32 \text{ m w.e. yr}^{-1}$ ) associated with the snowline-constrained mass balance model and combined it with the error estimate from the geodetic surveys. The estimated uncertainty of  $\pm 0.37 \text{ m w.e. yr}^{-1}$  does not assume independence of the errors from year to year. We believe that this is a fair estimate of uncertainties and openly show the limitation of such approaches for regional mass balance estimates. At the current stage, this is what is available for Central Asia.

We already present the standard deviation map to showcase that the two mass balance datasets are very different at an annual resolution. We additionally will provide a difference map of the mean annual mass balance to show also the inherent difference of the two mass balance times series at decadal scale. We do not want to judge the performance of the two mass balance estimates but just show their disagreements. Finally, we will highlight the uncertainty of the annual mass balance time series in the data section and add a statement in the discussion.

We use the datasets intentionally in their original spatial resolution. Any downscaling method will be subject to another level of uncertainty and subjectivity. In this work, we do not want to answer the question of "How should precipitation (temperature) data be downscaled?". We agree that there should be an elevation bias. However, we believe that our two types of analyses and our choice of derived statistics can account for this. First, our temporal analysis investigates the correlation and thus any systematic bias in form of a simple offset (fixed lapse rate) does not affect the outcome. Second, in the spatial analysis, we incorporate statistics like trend, and standard deviation as explanatory variables, which both are in the same way not dependent on the systematic offset. A simple downscaling that we could potentially performed with our expertise would also not help to resolve this problem. A downscaling using a linear function from a coarse to a finer pixel size would keep the trend and variability of the original coarse pixel time-series.

By incorporating the CHELSA data, we provide a means to see how a state-of-the-art downscaling affects the correlation analysis. CHELSA is maybe the most advanced downscaling of ERA5, incorporating various meteorological fields to cover assumptions about precipitation distributions in complex high mountains. The most important message from this exercise is finally that a downscaling can have a severe impact on our interpretation.

Following the critical tenor of this paper, this shows the "danger" of applying downscaling without ground truth data as we create a reality that might fit "better" our story.

**Downscaling of the reanalysis data:** I was surprised to see that the different reanalysis datasets had not been downscaled, especially considering that their respective resolution varies a lot. Is there not a risk that this will introduce elevation biases between subregions? Why has this not been considered in the study?

We will add a paragraph at the beginning of the data section to condense these thoughts and provide a justification for our data choice and our choice to not downscale the data.

**Dependence of snow cover on temperature and precipitation:** Does the fact that snow cover is dependent on temperature and precipitation not affect the regressions?

We strongly assume this, and we think that this is also reflected in our Figures 5 and 6, where we show that the incorporation of snow cover does not increase the number of significant correlations overall. Instead, a shift of important variables from precipitation and temperature to snow cover occurs. And we discuss this using the same argumentation (see Section 5.1, L.355 f.; L.399 ff.). We also mention the collinearity that results from this and is maybe the biggest concern of the comment. While not filtering the results for collinear occasions, the fact that there is this distinct shift from precipitation and temperature to snow cover seems a good proof that the collinearity is not biasing the overall outcome. The referenced paper by Vatcheva et al. (2016) (L. 233) states that the anticipated effect of collinearity results in masking out (removing) significant variables, rather than including more. The results can thus be understood as a more conservative identification of significant variables.

We will directly mention this example in the relevant paragraph.

Summarizing the scenarios in the discussion: These potential scenarios are interesting but lengthy and difficult to follow for readers not familiar with the particularities of the region. Could these be synthetized in a figure and streamlined? A short summary of the main differences would also be welcome.

We will provide a figure summarizing the two analyses in the form of a map. The map will show the most important drivers identified for the specific region for each analysis. We will shorten the text accordingly and synthesize instead what the figure shows more accessibly.

**Moving forward:** This is very briefly mentioned in the abstract only (as far as I can tell). I was a bit frustrated that there were not more discussions on this – how should one then proceed to interpret the mass balance patterns? What possible other tools could be used, what additional data should be collected?

We will add a more elaborated part on possible solutions to improve the current situation. This will cover in more detail instrumentation, advanced techniques for downscaling, holistic energy and mass conserving modelling approaches, as well as remote sensing validation data from independent data sources.

**1.2 Minor revision**

**1.2.1 Abstract**

Overall, I find that the abstract could be streamlined and the main message made clearer.

We will change the abstract to provide a clearer and better streamlined main message.

L3-4: 'Meteorological analysis, remote sensing products and novel approaches ... all provide ... ' We will adjust this accordingly.

L9: 'only ... do we find'

We will adjust this accordingly.

L13-14: This feels like a repeat from above

We will shorten the statement.

L16-18: this part is barely mentioned in the discussion and could be developed more.

We agree with this comment. Please see our answer and suggested modification outlined in the major comment above.

Introduction

In general, the introduction is interesting and well written but I think it would benefit from some reorganization efforts and some streamlining to make the message clearer.

We will restructure the introduction as shown below in the specific comments.

L25-26: This sentence does not bring much and could be removed. Are the two references to Gerlitz et al., 2019, 2020 really needed here?

We will restructure the first paragraph of the introduction.

L32-34: This feels out of place.

We will rephrase the statement (see comment above).

L20-36: There are lots of ideas in this first paragraph but the logical links are missing. These need to be reorganized/structured.

We will reorganize this part (see suggestion two comments earlier).

L46: 'Barandun et al. (2021) have applied'

We will adjust this accordingly.

L44-48: I am not sure that many details are necessary here, especially as these are described in length in the methods.

We will shorten this part.

L59-60: Are these details really necessary here? A simple reference to Hugonnet et al. (2021) is likely enough.

We will shorten this as suggested.

L61: something wrong with the English at the end of this sentence

This will be rephrased.

L63: There are actually some region-wide debris thicknesses assessments. See Rounce et al. (2021) and McCarthy et al. (2022)

We will rephrase the statement and add the suggested reference.

A scale-dependent debris-cover mass balance relationship and limited observation-based distributed debris thickness estimates hamper the explanatory power of debris cover for region-wide glacier mass balance patterns in Central Asia

L67: references missing. A recent one could be the work by Glasser et al. (2022) - 10.1016/j.geomorph.2022.10829

We will add the reference.

L67-69: Not sure these details are necessary here.

We will shorten the statement.

L69: Have the authors considered avalanching as a possible morphological control? It feels like for some of the steeper ranges of the region that could actually play a significant role (Brun et al., 2019)

This is a good point. Unfortunately, it is not straightforward to accurately quantify the avalanche contribution to mass balance and hence its importance as morphological driver. We will add this point to the sections where we introduce the potential morphological drivers and point out in the method section that we do not integrate it in our analysis.

Data

L87: Study site should be plural. No need to capitalize the nouns in the titles.

We will adjust this accordingly.

L121: I don't think this acronym has been defined before in the main text.

We will introduce the acronym in the text.

L124: Simple reference to RGI v6.0 is enough here (text can be shortened)

We will adjust this accordingly.

L134: The Barandun et al. (2021) dataset also uses geodetic mass balance products to calibrate their model (second order calibration).

We will rephrase this statement.

L135: Suggest adding reference to Figure 1 here.

We will adjust this accordingly.

L134-159: Are all the details provided here really needed considering that these are already published approaches?

We agree and will shorten this paragraph and add a few lines on the uncertainties.

L141: (Dee et al., 2021) should come after 'data'.

We will adjust this accordingly.

L148: 'observational' is not really true for Barandun et al. (2021) MB, as it comes from modeling.

We will remove this statement due to the shortening of the paragraph.

L155: can you explain a bit better the sentence 'local and regional scale biases can persist'? A reference might be needed here.

We will reword this. The sentence is supposed to say that the approach is sometimes performing badly at the single glacier scale (Hugonnet et al., 2021; e.g. Extended Data Fig. 5d) and shows varying uncertainties at the regional scale (e.g. Hugonnet et al., 2021; e.g. Extended Data Fig. 5d).

L181: Could you also provide the resolution in km for consistency with the other datasets? We will include this.

L205-209: I would recommend putting this in a separate subsection.

We will add a subsection.

L240-241: I struggle a bit with this k-mean clustering. Could you give a few more details.

We will add some more details on the k-mean clustering method with a standard reference.

Results L276: A3 should come before A4 in the text.

Figure order will be changed.

Discussion L357: remove the comma

We will adjust this accordingly.

L359-360: this makes sense as they are likely related

Yes, we agree. This is also discussed later in the manuscript, as mentioned in one of the earlier comments.

We will adjust this accordingly.

L434: snow cover decrease reported by

L489-490: missing parenthesis

We will adjust this accordingly.

Conclusion

L543-544: Use lower cases for Barandunetal and Hugonnetetal

We will adjust this accordingly.

Figures

The panels of the different figures need to be numbered.

We will adjust this accordingly.

Figure 1

- Instead of a globe, a map of HMA would be sufficient and more informative here.
- Shouldn't the mass balance should be in m.w.e? Also for consistency with the text.
- In general I don't really like the term 'surface' mass balance here at least for Hugonnet et al., 2021, these are geodetic mass balance measurements. I would suggest sticking to 'glacier' mass balance.

We agree on all points and adjust labels in figures, and all occurrences in the text accordingly.

Figure 2

- Why are the spatial and temporal analysis linked in the figure? Aren't they done independently from one another?
- It would be good to distinguish the data boxes and the methods boxes. Having the seasonal aggregation in the same box as the monthly meteorological time-series is confusing.

They are independent and we are thankful for the suggestions and will change the figure according to the suggestions.

**Figure 3**

• Could you specify what each cluster corresponds to in the figure? There are likely many ways of clustering this data, which criteria were used here and why?

As mentioned briefly in the text this is based on a k-means clustering algorithm. It is an unsupervised classification, meaning that there is not a precise association of a class with e.g. a process. The k-means clustering is based on the standard deviation of the mass balance time series of Barandun et al. Therefore, classes represent glaciers with different variability. We will clarify this in the figure caption and in the text (see also comment above).

Figure 4,5 & 6 Most of the comments here hold for the next figures as well:

- Suggest writing out SC, P, T
- Do these meteorological correlations relate then to trend, mean, STD?
- Which variables do the seasons refer to? This would need to be specified in the caption, and maybe even in a supplementary figure?
- Specify in the caption that 'short' and 'long' refer to the 2000-2014 and 2000-2018 periods.
- It needs to be specified also that this is an aggregation of all glaciers in the region.

We will write out the abbreviations and increase their size in the figure. The correlations are indeed incorporating all derived statistics. As explained also in the downscaling section, this allows to find relationships where there is e.g. an altitudinal bias. It also allows us to use the meteorological data in both the temporal and in the spatial analysis. The latter requires a single value for any given glacier, whereas the former utilizes the time-series as it is. We will clarify this and the other points in the text and also in the figure captions.

- Stay consistent with acronyms in figures
- The explanatory diagram does not need to be repeated in every subplot if it stays the same
- Where are the subplots with and without snow cover? I cannot find the legend.

Figure 7 & 8 These comments also hold for figure 8:

We will remove all explanatory pie diagrams but one and change the labels to be consistent with the other figures. The subplot without the snow cover was moved to the Appendix and we missed changing the caption accordingly. The figures are in A6. We will adjust the figure caption.

Figure A4

• Would there not be a way to weight the results of the right panel by the number of glaciers?

The only thing we want to show with this figure is that we did not identify any different relationship between surge- and non-surge-type glaciers with mass balance among the different subregions. It would certainly be possible to weigh the mass balance estimates but the main aim is really to show that there is not a systematical higher or lower value for surge-type glaciers. We will clarify this in the caption.

**2 Reviewer #2**

**2.1 Major revision**

I think the authors need to restructure their objectives to be something they more clearly address:

Objective 1 seems to encompass two separate research questions about the drivers and physical processes themselves, and then the degree to which this is simply a result of which data are used to explore the problem.

Objective 2 is related to the latter part of the first objective in my opinion and additionally implies that the authors explicitly compared ground truth data for correlation analysis. I'm not convinced that the authors really test enough the limitations of the gridded data in this manuscript (so objective 1, part 1), or at least not from the data/figures presented.

We agree that the description of the objectives was not well chosen. As it was written, one would indeed expect a more detailed analysis, and more importantly, conclusive results on the interplay between climate and mass balance. Instead, we finally provided what the reviewer correctly points out as second part of Objective 1. The motivation of our work is, nevertheless, a better understanding of the climate-mass balance relationship. This is why we chose the unfortunate wording in L.70 ff. We will reword in particular this section to differentiate motivation and resulting objective. The objective will more clearly state that rather than revealing dominant factors, we will explore the relationships using the newly available datasets. This will then address the question if we can identify consistent relationships between mass balance and climatologic/morphologic drivers.

We will address the limitations critique by providing a pre-analysis of the datasets used. This will be done following the suggested graphs in the style of de Kok et al. (2020), and by comparing the mass balance datasets using the already introduced aggregated pie chart maps.

It is clear that different reanalysis datasets will produce a different correlation with the same mass balance product. While the authors perform a nice assessment of these expected differences, I'm

left wondering how comparable the forcings really are in the first place. The authors apply ERA5 (approx. 36km), HAR30 (30km) and CHELSA (1km, using ERA5 to downscale). While the first two are closer in terms of spatial resolution and the processes they can represent at the surface, CHELSA represents a major effort to include finer scale topographical features, such as windward/leeward precipitation adjustments, even though it likely remains with significant seasonal biases. Accordingly, I would not expect to find a similar relationship to glacier mass balance (spatially or temporally) when comparing to ERA5 or HAR30 (e.g. L286-7, L343-4), which cannot represent precipitation dynamics at the scales relevant to glacier processes (e.g. L401).

The authors have given some consideration to these limitations in their discussion section, but I'm not convinced that it is a fair test and fully supports the conclusions about an ambiguity to glacier response in the region. I think that redoing analysis with additional datasets would also not necessarily advance the findings of this manuscript and perhaps cloud the main messages. However, I do think that the authors need to address these discrepancies and perhaps even remove CHELSA from the analysis. More importantly, I would like to see some level of comparison between the different products for air temperature and precipitation estimates (especially if CHELSA remains included). What explains the presence or lack of correlations in given subregions, for example? Some supplementary figure(s) would be useful to that end.

A more detailed answer to this valid and important point regarding the spatial resolution can be found in an answer to reviewer 1 about downscaling. We argue that using the relatively coarse datasets alongside the fine resolution CHELSA dataset is a valuable and valid approach for what we want to show. Reviewer 1 was concerned about the elevation bias that results from not downscaling the data. The coarse data provide ultimately the same data for multiple glaciers. Here we argue twofold: 1) a systematic offset in the independent (meteorological) variable will have no impact on the outcome of the temporal analysis (significant relationship or not) because we perform a correlation analysis, where a determined significance is independent of a linearly scaled variable; 2) this holds true for the spatial analysis because the statistics derived from the meteorological time-series that serve as independent variable reflect temporal components (standard deviation, trend) and thus the same applies as for point (1).

However, this does not eliminate the problem that there are the same time-series or the same derived statistic for multiple glaciers encompassed by a single grid cell. Finding a relationship might be prevented if the glaciers within the extent of a grid cell show a stronger variability in mass balance than between different grid cells. This is, however, not the case for the entire study region, or the larger regional subsets, as we show in our study (e.g. Fig. 4,5,6). For smaller regional subsets, as shown particularly in Fig.5, this might have actually been the reason for the lack of significant correlations. However, because we also use CHELSA, we can see that this is not necessarily a spatial resolution problem. We argue that this can be related to changing variable importance that we do not cover in our study, e.g. a transition into more radiation-dependent processes (Sec. 5.1, L.367 f.). This assessment is supported by having CHELSA as an additional fine spatial resolution dataset in our analysis.

We do agree that having a better idea of how the different datasets compare to each other would provide a much-appreciated basis for any reader. The additional figure about meteorological datasets will show the differences and correlations now.

We will add the outlined considerations in the Data and Discussion sections.

Building on the previous point, I think the authors also need to address any potential circular issues related to the use of ERA-Interim for the derivation of the  $MB_{Barandunetal}$  mass balance results and its comparison to the next generation of the ECMWF reanalysis.

This point has been raised by both reviewers and we understand their concern. However we believe the use of the two datasets is justified in the way they have been used. The approach presented in Barandun et al. (2021) fortunately does not rely on precise meteorological input (which simply do not exist extensively in the high mountains of Central Asia). Barandun et al. (2018) showed that using transient snowlines for model calibration reduces the sensitivity to the meteorological input data. The sensitivity analysis in Barandun et al. (2018) uses an average temperature and precipitation dataset without any year-to-year variability still provided results with an RMSE of less than 0.2 m w.e.  $yr^{-1}$  in comparison with the direct measurements. This highlights that the snowline observations used for calibration, for each glacier and year individually, are primarily responsible for the modelled mass balances. For more details, we believe it is sufficient to refer to the previous work Barandun et al. (2018) that provides an in-depth analysis of the model sensitivity which the current manuscript builds upon.

In Barandun et al. (2018) ERA-interim was used becauseOrsolini et al. (2019) showed the superior performance of ERA-interim for mountainous regions like High Mountain Asia over ERA5, mainly due to the assimilation of *in situ* station data in ERA-interim that is omitted in ERA5. The products have thus significant difference in the generated precipitation fields that are independent from each other. This provides the opportunity for further analysis of possible correlations between the mass balance time series and the climatic drivers with ERA5 or other independent reanalysis datasets. Currently ERA5 is one of the most used reanalysis products due to its finer spatial resolution and often shortcomings as outlines above are not considered in correlation analysis. Despite the low model sensitivity we did not use ERA-interim for the correlation analysis. We will add a justification of the dataset in the manuscript, explaining also the insensitivity of the mass balance time series to the meteorological input data.

The temporal analyses produce clearer and more interpretable plots for this manuscript (which I liked), but I have concerns about the robustness of these results given their short temporal scale (14 years for correlations using HAR30) and high uncertainties for mass balances given annual data series. While the ideal minimum sample size required can vary based on the type of observations, it has generally been considered that sample sizes less than 30 can produce imprecise correlation estimates. I'm curious why the authors did not consider a few longer term reanal-ysis/gridded products of similar spatial resolution (e.g. HARv2, ERA5Land, WRF9km) that all provide data since 1980 at least. I guess there are also temporal limitations for the mass balance data period, but this should at least be discussed more clearly.

This is a good point and we would have liked to use longer time-series. We agree fully that it would certainly make the analyses more robust. Our work is unfortunately restricted by the length of the mass balance estimates rather than the meteorological datasets. But, we were also keen to use the HAR version 1 dataset as it is one of the best performing datasets in the region that has been shown to actually resolve the discharge variability over the greater Pamir domain (Pohl et al., 2017), and also over the Tibetan plateau (Biskop et al., 2016).

Comparison between HAR version 1 and version 2 (not shown here) show that these datasets are very different. This is not surprising as version 2 uses ERA5 as input data. We certainly intend to raise some awareness about what looks like a tendency to more and more use ERA5 and ERA5-land directly - or as basis for further downscaling (HAR v2, CHELSA)- because it is so convenient with its long temporal coverage and good spatial resolution. With our choice of datasets we were hoping to provide simultaneously two critical views on the climate-glacier interaction topic. As the comparison between CHELSA and ERA5 shows, the downscaling has a significant impact on the obtained results. By incorporating HAR version 1, with a similar spatial resolution as ERA5, we also have a comparison between different forcing datasets for the downscaling. Without that, the main deductible conclusion would be that downscaling affects the outcome.

We will add an additional paragraph on our justification for using the indeed rather short HAR version 1.

Plotting such large datasets can be challenging and I think the authors have done a great job in many respects. I think there are several areas where they might still be improved for clarity and to help the message of the manuscript. I have given some specific comments below.

Thank you for the comment. We will address the specific comments below and refer to the answers given to each specific comment.

As mentioned, the work is generally well written. However, there are several instances where the wording is unclear or with grammatical errors. I have tried to provide some examples of this below, but the authors should carefully check the entire manuscript at the next submission.

We went through the specific comments and will include the changes in the updated manuscript. We have also decided to ask for a professional proofreader to eliminate unclear wording. ———

**2.2 Specific Comments**

L90+: I think that the authors provide some very valuable and interesting information about the climatic setting in this paragraph, but I do not feel that the discussion reflects enough upon these interesting variations. Building upon my main point about comparisons of the gridded datasets, the authors should present spatial maps of mean (summed) temperature (precipitation) from different products, or better yet, the spatial trends of some of these variables (where are they statistically significant for ERA5/HAR30?). Exact comparisons can be challenging due to differences in spatial scale, but the authors could consider spatially binning the grids as presented, for example, in de Kok et al. (2020).

As mentioned in other comments, we plan to provide such a figure.

L185-188: The authors state that snow depth over-estimation in ERA5 renders use of radiation variables "problematic", and rely upon only temperature and precipitation data. However, there are well reported cold biases for air temperature at high elevation regions (such as those in this manuscript) which also result from this albedo-effect (e.g. Wang et al. 2020).

For any type of correlation analysis, a systematic and linearly scaled over-/underestimation does not pose a problem. Our main reason for the choice of the two main variables temperature and precipitation is, however, that they are widely available. For reanalysis with meteorological station data assimilation these two variables should also be better constrained than others, which are rarely or not at all measured at meteorological stations or not retrieved by other means.

We will clarify this.

L192: Given that the authors test the latest products from ECMWF (ERA5) and CHELSA, I'm surprised that they did not also consider the HARv2 product which is openly accessible and available back to 1980 (thus eliminating the shorter temporal focus and affecting the strength of correlations). I think a line should be added somewhere to justify the use of these three datasets specifically and why they are considered comparable, especially given the points from my main comment above.

This point is largely answered in a previous comments. We will address this in the data section with more rigour.

L205: I'm not convinced that the MOD10CM product at a 5km resolution does much to aid the analysis presented here. Given its spatial resolution, can it reasonably represent anything meaningful at the scale of the glaciers? Please add some additional information

Even though the resolution of MOD10CM is not matching the individual glacier scale it provides a robust small (10km) scale snow cover evolution time-series. We chose this product for the robustness (averaging out falsely classified pixels and cloud cover issues) with the intention to have a general overview of snow patterns. Similar to the coarse (30km) ERA5 and HAR products, we use either the time-series directly in the temporal analysis or the statistics that provide information derived from the temporal domain (standard deviation, trend, and also snow cover change per time step) for the spatial analysis. Many times the high spatial resolution MODIS snow cover products provide either time-invariant signals at the highest resolution (250 m and 500 m), or slightly varying values suggesting changes in winter where these changes are more likely to reflect measurement or classification uncertainties than actual changes.

In our analysis we use snow cover mainly to check the results for plausibility, i.e. if we can substitute an interplay of precipitation and temperature (leading to snow cover) with an actual estimate for snow cover. As such, the resolution closer to the coarse meteorological datasets is likely better suitable than a fine resolution dataset that would capture small scale processes that certainly are not represented in the meteorological datasets.

An advantage of the coarse resolution is certainly that changes over time are more clearly captured. This is because the averaging over a larger area will detect a decline or increase in snow cover due to the more variable evolution at lower elevations. With other words, we obtain a better signal-to-variability ratio. In the end this is a trade-off. But we think that the use is justified for the presented work and our idea of testing the obtained variable importance for plausibility.

We will incorporate these thoughts into the Data and Discussion sections.

L215-216 (and 243-245): I do not understand clearly how the authors relate the spatial means of mass balance with trends of meteorological variables. From the trend you will have just one value per gridded pixel, correct? So the authors combine all glaciers in a given binned region (circles in Figure 1) or just the entire region, and regress all the trends in a given variable (say temperature) at the corresponding pixel of each glacier against the mean or STD of mass balance at that glacier? I believe the authors need to more clearly explain this. Unfortunately, Figure 2 does not help much with this interpretation.

We will explain this in more detail in addition to the proposed changes from the previous comments about the coarse reanalysis resolution. We do not calculate a mass balance mean but instead use each glacier as one data point. This will - explained in the other comments - indeed lead to often multiple glaciers sharing the same meteorological data point. Regarding Figure 2, we do not have a direct solution for how to incorporate these ideas in the figure but we think that with our argumentation of why we think that our approach is sound (explanations in previous comments), we can provide a much clearer discussion on the this issue.

L221-222: Please re-write this sentence to improve syntax. It makes sense, but is not well written.

We will rewrite the statement.

L225-226: This is also a little unclear from the writing. Do the authors mean that they do not mix variables from different products in their analyses (e.g. temperature from ERA5 and precipitation from CHELSA)?

Yes, exactly. We will clarify this in the text.

L242: Please redefine here what is meant by these 'hotspots'. I note that this comes from the author's previous published work, but it would be better to clarify explicitly for the reader what is meant by a hotspot of mass balance variability. I interpret that as the areas in Figure 1 where STD is greatest, for example.

We will introduce what we mean with "hot spots" in the introduction.

L253: The authors consider the ambiguity of the glacier response here as being largely due to different meteorological or mass balance datasets considered, but what about the uncertainties stemming from the annual-series of geodetic mass balances themselves, which are, to the authors own admission (e.g. L156), with less accuracy, and I assume, less precise? Again, it would be ideal to be able to assess the comparison of the different gridded meteorological and mass balance products independently to identify where the largest differences in the datasets lie, not just the correlations/significance.

In accordance with the comment of reviewer #1, we will outline the differences between the annual time series and highlight the usefulness of using the two different mass balance time series. We add also a statement on the uncertainties related to both mass balance time series. For a detailed answer on the uncertainties and proposed changes in the manuscript please also refer to the answer to reviewer #1 (major revision comment #2).

L264: I think the authors should attempt to signpost some of the key findings a little better in the figures and simplify some of the main figures where possible (see specific figure comments below). I find it a little hard to navigate from text to figure to follow the story-line.

We will improve the figures of the paper and refer here to the answer of to specific comments.

L344: The authors do not consider all available reanalysis datasets or atmospheric data for their analysis. Please rephrase this.

We will rephrase the statement.

L346: "at the local scale..." Please check the text for many of these small grammatical issues or minor mistakes.

We will adjust this accordingly.

L346-8: This sentence does not make sense to me. Please rephrase to make your point clear.

We will adjust the entire paragraph to account also for the next three comments.

L350: but they do not prevent this analysis here, the authors have an annual mass balance value for temporal analysis. I think this first paragraph needs to be restructured slightly to highlight what the authors are able to show, which is normally not considered/available.

This comment is not fully clear to us. The temporal analysis provides us with the means to avoid any arbitrary aggregation and instead look at each glacier individually. We will rephrase the entire paragraph, so that it becomes more clear to the reader (see comment above).

L369-379: I think the authors should rewrite this segment as their point is not particularly clear upon reading it. Do the authors mean that their HiMAP regions encompass multiple sub-climatic regions and associated atmospheric processes which therefore limits the usefulness of these defined regions for such analyses?

There are various points addressed that encompass spatial and temporal aspects. But yes, this is often the problem when using standard regional divisions for correlation analysis. Apart from this, the heterogeneity of glacier response to changing atmospheric settings that again leads to changes in sensitivity can create a very complex picture within one subregion. This is then difficult to relate to the reanalysis datasets. We will try to be clearer with our statement and rewrite and restructure the section.

L382: Please define what "rotation" of mass balance gradients means.

We will explain this better.

L385: reword "already slight uncertainties..." It does not make sense. Small uncertainties? What estimates? Which variables?

We will rephrase the statement

L387: How "invisible"? Are simply not resolved by the coarse reanalysis grid scale?

This is indeed what we meant to say. We will clarify.

L388-9: Not clear. Do the authors suggest that coarse grid scale products can capture changes in these regions because there is less sub-grid variability in topography and its associated meteorological complexity?

We meant that due to the different mass balance sensitivities of glaciers in continental or more maritime settings, their reaction to a change will be different in amplitude. For glacier at high altitudes and more continental climate regimes, with lower mass balance sensitivity, larger changes are needed for a glacier response. These more important changes are probably easier to capture in reanalysis datasets than small scale changes and might explain the higher number of significant correlations found for more continental settings. However for glaciers in sub-continental regions, where sensitivity is highest, small variations at various spatial scales over a glacier might remain unresolved in the coarsely resolved reanalysis products, leading to the lower correlation. This adds to the heterogeneous mass balance response and renders finding direct correlations with climatic drivers difficult.

We will try to be more clear and better structured with the argumentation within the entire paragraph (see also changes above).

L392: What do the authors mean with "outer orogene margins"?

We will adjust this. The "outer" was not necessary.

L399: "remains unclear..."

We will adjust this accordingly.

L406: please provide examples of which physical snow properties that the authors refer to and what variables missing from the analysis might influence the results.

We will adjust this accordingly.

L410: "extent".

We will adjust this accordingly.

L434: Citation requires parenthesis, check formatting.

We will adjust this accordingly.

L489: Check formatting.

We will adjust this accordingly.

L497-8: Related to the general comments, 14 years does not produce an ideal correlation period. Some discussion on this is needed.

We will add more discussion on this as explained also in an earlier comment.

LL503: The authors cannot evaluate the biases and patterns of reanalysis datasets due to lack of ground data, but should still highlight how the products relate to each other. Although long term in situ observations are rare, and I understand that it is not the goal of the manuscript to identify the 'best' product to use, but I would be interested to see how the different products compare to even short term measurements from sporadic AWS measurements familiar to the authors from previous work (e.g. Abramov Glacier, Golubin Glacier).

We will include the dataset comparison. The comparison with AWS or meteorological station data in general comes with a lot of culprits that we do not want to address in this paper. At least for the HAR dataset there is a comparison available in Pohl et al. (2015). For Abramov there is already the problem that precipitation is not measured by the modern AWS (Kronenberg et al., 2021) and data gaps are very common at other sites as well. We think that trying to find a suitable method for a sound comparison would - although interesting - not change the conclusions in the present work. We hope this is acceptable for the reviewer. We will link the two references mentioned before and look for and include references from other sites where there has been an assessment made. Either way, a good or a bad correlation should only locally increase our trust in any of the products but not over the whole study domain.

Figures

Figure 1: The inset map in the upper right panel should ideally be of High Mountain Asia to provide a better spatial context as to where the study site is. If able, the authors should attempt to neaten the location and intersect of the legend numbering as there is overlap in places.

We will include a HMA inset map and neaten the legend.

We use the term "Relative Importance" which at this point is only explained in the methods. We will add this information in the caption with a simpler wording and reference to the main text to aid understanding the graphs.

Figure 4: Please clarify in the caption how variable importance is defined and why this is given as a relative freq. in the plot labels. Also ideally provide indices  $(a,b,c\ etc)$  to aid navigation for the reader from the main text. I don't find a 'short' analysis period as meaningfully different to the long one for the lower panels. Such morphological characteristics (e.g. debris) won't greatly change over 4 years. Consider removing it to streamline and simplify the figure. In general I find it very hard to interpret the information given in these figures (4,5,6, A1, A2). Any attempt to simplify it or remove parts would aid interpretation.

Figure 5: I think this figure has far too much information and becomes difficult to interpret. Please consider streamlining the most crucial information where possible, perhaps just showing

the short periods, or those with MOD10. For upper panels, one has to struggle to align the x axis label and interpret what is the take-home information. legends should be increased in size, but could easily be kept for just one panel for the whole left and right side.

This is a difficult point. We agree that this is a lot of information provided in the figure. However, we also think it is well suited to show the differences per sub-regions, the changing variable importance with and without snow cover, and between the two considered time periods. As we take reference to these figures, we would work on improving the readability by adding helping lines and by highlighting bars that we reference later on on the text.

Figure 6: Similar considerations to the above.

We will address this in the same way as pointed out for the previous comment.

Figure 7: These plots are more clear than the previous ones. Perhaps the authors can re-structure the panels to be 3 rows and two columns, so that the size on the page can be enhanced and the reader can more easily interpret the small circles. Why using 'M' in front of the meteorological datasets? I think it would be clearer without. I do not understand from the figure caption the "with or without snow cover..." as all show scf as a variable importance coloured on the map. Please clarify or correct this. Please use indices for the subplots to help the reader. For Figure 8 as well.

We will homogenize the different abbreviations and write out the data combinations (M for MODIS) to avoid this confusion. We will try to optimize the figure readability by the proposed  $3 \ge 2$  format to increase the individual panel sizes.

**References**

- Barandun, M., Huss, M., Usubaliev, R., Azisov, E., Berthier, E., Kääb, A., Bolch, T., and Hoelzle, M.: Multi-decadal mass balance series of three Kyrgyz glaciers inferred from modelling constrained with repeated snow line observations, The Cryosphere, 12, 1899–1919, 2018.
- Barandun, M., Pohl, E., Naegeli, K., McNabb, R., Huss, M., Berthier, E., Saks, T., and Hoelzle, M.: Hot spots of glacier mass balance variability in Central Asia, Geophysical research letters, 48, e2020GL092084, 2021.
- Biskop, S., Maussion, F., Krause, P., and Fink, M.: Differences in the water-balance components of four lakes in the southern-central Tibetan Plateau, Hydrology and Earth System Sciences, 20, 209–225, 2016.
- de Kok, R. J., Kraaijenbrink, P. D., Tuinenburg, O. A., Bonekamp, P. N., and Immerzeel, W. W.: Towards understanding the pattern of glacier mass balances in High Mountain Asia using regional climatic modelling, The Cryosphere, 14, 3215–3234, 2020.

- Kronenberg, M., Machguth, H., Eichler, A., Schwikowski, M., and Hoelzle, M.: Comparison of historical and recent accumulation rates on Abramov Glacier, Pamir Alay, Journal of Glaciology, 67, 253–268, 2021.
- Orsolini, Y., Wegmann, M., Dutra, E., Liu, B., Balsamo, G., Yang, K., de Rosnay, P., Zhu, C., Wang, W., Senan, R., et al.: Evaluation of snow depth and snow cover over the Tibetan Plateau in global reanalyses using in situ and satellite remote sensing observations, The Cryosphere, 13, 2221–2239, 2019.
- Pohl, E., Knoche, M., Gloaguen, R., Andermann, C., and Krause, P.: Sensitivity analysis and implications for surface processes from a hydrological modelling approach in the Gunt catchment, high Pamir Mountains, Earth Surface Dynamics, 3, 333–362, https://doi.org/ 10.5194/esurf-3-333-2015, 2015.
- Pohl, E., Gloaguen, R., Andermann, C., and Knoche, M.: Glacier melt buffers river runoff in the Pamir Mountains, Water Resources Research, 53, 2467–2489, 2017.
- Vatcheva, K. P., Lee, M., McCormick, J. B., and Rahbar, M. H.: Multicollinearity in regression analyses conducted in epidemiologic studies, Epidemiology (Sunnyvale, Calif.), 6, 2016.

---

## Author Response (AR1)

**Author Comments**

**Central Asia's spatiotemporal glacier response ambiguity due to data inconsistencies and regional simplifications.**

Martina Barandun and Eric Pohl

We would like to thank the two reviewers for the positive feedback and the constructive comments on the submitted manuscript. We agree on most of the critique and provide in the following our suggestions on how to implement the changes to the manuscript for all the issues pointed out. As several issues were raised by both reviewers we post this combined response identically to both RCs. In cases where we disagree, we present our reasoning and arguments, which was implement in the edited version. In order to further improve the clarity, we made use of a professional proofreader.

Below, we respond to all comments, and state how we plan to account for them in the revised version of the manuscript. The responses (normal font style) to the reviewers' comments are written directly after the reviewer comments (displayed in italic font style). The revised sentences in the manuscript are given in quotation marks.

**1 Reviewer #1**

**1.1 Major revision**

***Direct comparison of datasets:*** *Looking at how the different reanalysis datasets correlate with the glacier mass balance datasets is interesting but I feel that it adds quite some complexity in the interpretation. I am missing in this manuscript a separate comparison of all mass balance products together and all climatic data together. This would likely help understand the multiple linear regressions better.*

We added a comparison between the different mass balance and reanalysis products. Regarding the mass balance data, we show an overview map in the same style as for all other presented maps with pie charts per 25 km by 25 km regions to make differences easier to spot with the eye. The strongest differences can be seen at the short period and we include this figure in the main text. We provide the map for the long period in the Supplementary. We follow the design by de Kok et al. (2020) for the comparison of reanalysis products. Regarding the presentation of meteorological data, we think that the overview in the style of de Kok et al. (2020) is providing the important information on spatially different estimates by the products. We are only planning to provide the comparisons of the annual data for temperature and precipitation, instead of producing two (variables) times five multi-panel figures as in de Kok et al. (2020). For the main text, we are planning to include the precipitation comparison and refer to the Annex/Supplementary for the temperature comparison. That is because these figures occupy almost a page. See also answer to comment of Reviewer 2.

Regarding a comparison with meteorological station data, we do not think this is helpful for the present manuscript. The issue of missing ground truthing data is now better referenced in the text. There are multiple studies that have investigated new data products for precipitation. There is, however, no real baseline to compare any product against because most of the few available meteorological stations are located in valleys and thus there is limited to no ground truthing at higher elevation. We can see the motivation for the reviewers' request. Reviewer two has pointed also out that the stated objectives should be adjusted to better match the presented results and discussion. We agree on this and adjust the objective description to more accurately match the presented work.

l.154:"This serves for a consistent analysis of dominant drivers and to highlight the differences between the two mass balance estimates (Fig. 2)."

l.35: "Meteorological measurements are sparse and often discontinuous even for the most monitored glaciers in Central Asia, such as Abramov or Golubin Glacier (Kronenberg et al., 2021; Azisov et al., accepted). Replacement of old meteorological stations with modern sensors often lacks precise homogenization. Regional extrapolation from station data and use of existing time series as validation datasets for gridded products are thus problematic."

l.287ff: "A comparison of the different reanalysis datasets is shown for precipitation in Figure 3 and for temperature in Figure A2. Only CHELSA provides a spatial resolution to resolve intramountain range variability. Differences in precipitation amount and different trends exist independently of the spatial resolution (Fig. 3). However, for temperature, CHELSA is the temperature-conserving downscaled ERA5 product with matching trends and a correlation of one (Fig. A2i). For precipitation, CHELSA and ERA5 show deviating distribution

patterns and trends, highlighting the nonlinear downscaling method (Fig. 3). The use of the datasets in their original spatial resolution in consecutive analyses allow for an assessment of how downscaling affects the correlation with mass balance data."

———

**Barandun et al. (2021) mass balance:** *It strikes me that this mass balance model is based on ERA-interim data, which is also a reanalysis product similar to ERA5. It has been calibrated with snowlines and geodetic mass balance, but I suspect the results are likely influenced by the climate data as well. Have the authors compared the ERA-interim with the ERA5 to look for possible changes? This is likely to influence the regressions and result in some circularity – do the regressions agree or not with the results of Barandun et al. (2021)?*

This point has been raised by both reviewers and we understand their concern. However, we believe the use of the two datasets is justified in the way they have been used. The approach presented in Barandun et al. (2021) fortunately does not rely on precise meteorological input (which simply does not exist extensively in the high mountains of Central Asia). Barandun et al. (2018) showed that using transient snowlines for model calibration reduces the sensitivity to the meteorological input data. The sensitivity analysis in Barandun et al. (2018) uses an average temperature and precipitation dataset without any year-to-year variability. Obtained mass balances using the transient snowline approach still provided results with an RMSE of less than 0.2 m w.e. yr$^{-1}$ in comparison with the direct measurements. This highlights that the snowline observations used for calibration, for each glacier and year individually, are primarily responsible for the modelled mass balances. For more details, we believe it is sufficient to refer to the previous work Barandun et al. (2018) that provides an in-depth analysis of the model sensitivity which the current manuscript builds upon.

In Barandun et al. (2018) ERA-interim was used becauseOrsolini et al. (2019) showed the superior performance of ERA-interim for mountainous regions like High Mountain Asia over ERA5, mainly due to the assimilation of *in situ* station data in ERA-interim that is omitted in ERA5. The products have thus significant difference in the generated precipitation fields that are independent from each other. This provides the opportunity for further analysis of possible correlations between the mass balance time series and the climatic drivers with ERA5 or other independent reanalysis datasets. Currently ERA5 is one of the most used reanalysis products due to its superior spatial resolution. However, often the above outlined shortcomings are not considered in correlation analyses. Despite the low model sensitivity we did not use ERA-interim for the correlation analysis. We added a justification of the dataset in the manuscript, explaining also the insensitivity of the mass balance time series to the meteorological input data.

l.136: "The model driven with ERA-interim reanalysis data (Dee et al., 2011) for each glacier and year separately (Barandun et al., 2021, 2018). ERA-interim was chosen because, unlike other reanalysis products (e.g., ERA5), *in situ* observations in mountain regions are assimilated (Orsolini et al., 2019). Barandun et al. (2021) provided annual mass balance time series, closely tied to direct observations, with low sensitivity to meteorological input for roughly 60% of the glaciers larger than 2 km$^2$ in the data-sparse Tien Shan and Pamir. For more details on the methodological approaches for mass balance determination and model sensitivity the reader is referred to Barandun et al. (2018) and Barandun et al. (2021); for the automatic snowline mapping, to Naegeli et al. (2019); and for the geodetic estimates, to Girod et al. (2017) and McNabb et al. (2019). Differences in ERA-interim and

ERA5 performance and output at high altitudes are highlighted in Orsolini et al. (2019) and Liu et al. (2021), showcasing the independence of the two datesets."

————

*Uncertainties in mass balance data: I am concerned about the use of yearly glacier mass balance, especially as it is not clear to me how representative of the actual mass balance. This is relatively well explained for the Barandun et al. (2021) dataset, but less for the Hugonnet et al. (2021) – are these actual geodetic measurements made on a yearly basis or are they extracted from the general trend? In either case, I expect the uncertainties on this data to be quite high relative to the glacier mass balance values, especially for glaciers in Central Asia, which tend to not lose mass very quickly. This seems to be confirmed by the fact that the Western regions have higher significant correlation frequency (also the ones with clearer mass balance signal). I am therefore wondering how valid it is to take yearly data and whether taking decadal trends would not be a better avenue to analyze the spatio-temporal patterns. At the very least a discussion about these uncertainties would be necessary to include in the manuscript.*

This is completely true. The idea behind using more then one mass balance time series is to show the uncertainties not just in the reanalysis datasets but also in the mass balance time series currently available. Hugonnet et al., also point out that the data should not really be used in an annual study – even though they provide the values. The annual values are mass conserving with respect to the long-term calculated geodetic values and thus the provided mean annual values are usable. Our reasoning for using the annual estimates anyways is that it is so far the only other mass balance dataset at annual resolution available in the region. We point this out more clearly in the reworked manuscript.

Both mass balance time series use geodetic estimates calculated from the same dataset (ASTER) but use two different processing approaches, leading to very different glacier-specific mass balance estimates. This highlights the uncertainties in geodetic methods and how these relate into uncertainties further down the processing chain.

We believe the case study in Barandun et al. (2018) provides an extensive assessment of the model sensitivity and uncertainty and eventually provides a very conservative estimate. Barandun et al. (2021) adopted the mean uncertainties provided in Barandun et al. (2018) ($\pm 0.32\,\mathrm{m\,w.e.\,yr^{-1}}$) associated with the snowline-constrained mass balance model and combined it with the error estimate from the geodetic surveys. The estimated uncertainty of $\pm 0.37\,\mathrm{m\,w.e.\,yr^{-1}}$ does not assume independence of the errors from year to year. We believe that this is a fair estimate of uncertainties and openly show the limitation of such approaches for regional mass balance estimates. At the current stage, this is what is available for Central Asia.

We already present the standard deviation map to showcase that the two mass balance datasets are very different at an annual resolution. We now additionally provide a difference map of the mean annual mass balance to show also the inherent difference of the two mass balance times series at decadal scale. We do not want to judge the performance of the two mass balance estimates but just show their disagreements. Finally, we highlight the uncertainty of the annual mass balance time series in the data section and add a statement in the discussion.

l.156ff: "Both datesets are tied to elevated uncertainties of the annual mass balance estimates. Barandun et al. (2021) adopted mean uncertainties provided in Barandun et al. (2018) ($\pm 0.32\,\mathrm{m\,w.e.\,yr^{-1}}$) associated with the

snowline-constrained mass balance modelling and combined them with the error estimate from the geodetic surveys. This resulted in a rather conservative uncertainty of $\pm 0.37\,\mathrm{m\,w.e.\,yr^{-1}}$ and does not assume independence of the errors from year to year. Hugonnet et al. (2021) reported uncertainties of up to $\pm 0.1\,\mathrm{m\,w.e.\,yr^{-1}}$ for mean annual mass balance values of 5-year periods. Hugonnet et al. (2021) provide uncertainties in mass changes for periods shorter than 5 years only for the global or near-global estimates. Global annual mass balance uncertainties are reported to be higher, with around $\pm 0.2\,\mathrm{m\,w.e.\,yr^{-1}}$ (Hugonnet et al., 2021). We expect higher uncertainties for annual time series of individual glaciers."

l.535: "Lack of regional-wide systematic cryospheric and atmospheric monitoring limits the validation of the available datasets for scientific applications, resulting in an elevated uncertainty for mass balance modelling and interpretation on the underlying processes of glacier response to climate change. We cannot rate the different products."

————

***Downscaling of the reanalysis data:*** *I was surprised to see that the different reanalysis datasets had not been downscaled, especially considering that their respective resolution varies a lot. Is there not a risk that this will introduce elevation biases between subregions? Why has this not been considered in the study?*

We use the datasets intentionally in their original spatial resolution. Any downscaling method will be subject to another level of uncertainty and subjectivity. In this work, we do not want to answer the question of "How should precipitation (temperature) data be downscaled?". We agree that there should be an elevation bias. However, we believe that our two types of analyses and our choice of derived statistics can account for this. First, our temporal analysis investigates the correlation and thus any systematic bias in form of a simple offset (fixed lapse rate) does not affect the outcome. Second, in the spatial analysis, we incorporate statistics like trend, and standard deviation as explanatory variables, which both are in the same way not dependent on the systematic offset. A simple downscaling that we could potentially perform with our expertise would also not help to resolve this problem. A downscaling using a linear function from a coarse to a finer pixel size would keep the trend and variability of the original coarse pixel time-series.

By incorporating the CHELSA data, we provide a means to see how a state-of-the-art downscaling affects the correlation analysis. CHELSA is maybe the most advanced downscaling of ERA5, incorporating various meteorological fields to cover assumptions about precipitation distributions in complex high mountains. The most important message from this exercise is finally that a downscaling can have a severe impact on our interpretation.

Following the critical tenor of this paper, this shows the "danger" of applying downscaling without ground truth data as we create a reality that might fit "better" our story.

We added a paragraph at the beginning of the data section to condense these thoughts and provide a justification for our data choice and our choice to not downscale the data.

l.192: "**CHELSA** time series version 2.1 is a processed and downscaled version of the ERA5 dataset which incorporates directional wind speeds and cloud cover observations to address precipitation biases and wind and leeward distributions (Karger et al., 2017). The final downscaled product presents a $\approx 1\,\mathrm{km}$ spatial and monthly temporal resolution. By incorporating the CHELSA data, we provide a means to determine how a state-of-the-art

downscaling affects the correlation analysis."

l.392: "CHELSA is a processed and downscaled version of the ERA5 dataset. When using these two products, differences appear in the correlation analyses due to different spatial resolutions in the predictor variables; however, these differences only become striking when using the regional subdivision (Fig. 6). This might respond to a low number of data points with high mass balance variability within areas of encompassing meteorological grid cells, while the mass balance variability between the different encompassing grid cells in the regional subdivision is rather low. The nonlinearly downscaled precipitation of CHELSA (Fig. 3) compared with the conserving values for T (Fig. A2) can explain the higher number of significant correlations of this product compared with that of ERA5. As we use two predictor variables, the original P fields of CHELSA can change the significance of T. In the temporal analysis, the effect is even clearer, leading to partly opposing results (Fig. 9 and Fig. A8). We do not answer here how exactly any downscaling method can affect results but we show that advanced downscaling methods (CHELSA's precipitation) can severely impact driver interpretation. Different outcomes can be expected depending on the downscaling technique used, adding subjectivity to the results."

———

***Dependence of snow cover on temperature and precipitation:*** *Does the fact that snow cover is dependent on temperature and precipitation not affect the regressions?*

We strongly assume this, and we think that this is also reflected in our Figures 5 and 6, where we show that the incorporation of snow cover does not increase the number of significant correlations overall. Instead, a shift of important variables from precipitation and temperature to snow cover occurs. And we discuss this using the same argumentation (see Section 5.1, L.355 f.; L.399 ff.). We also mention the collinearity that results from this and is maybe the biggest concern of the comment. While not filtering the results for collinear occasions, the fact that there is this distinct shift from precipitation and temperature to snow cover seems a good proof that the collinearity is not biasing the overall outcome. The referenced paper by Vatcheva et al. (2016) (L. 233) states that the anticipated effect of collinearity results in masking out (removing) significant variables, rather than including more. The results can thus be understood as a more conservative identification of significant variables.

We now directly mention this example in the relevant paragraph.

l.250: "We do not run a separate collinearity test to identify and remove correlating predictors even though this is expected, for example, in the case of different temporal temperature aggregates or snow cover and precipitation data (Fig. 4). As our goal is to show possible inconsistent outcomes depending on the chosen glacier mass balance and meteorological dataset, we focus only on identifying significant predictors rather than model coefficients. A possible impact in case of collinearity is a nonsignificance of one or both correlating predictors (Vatcheva et al., 2016); thus, an anticipated effect of collinearity is the concealing (removing) of significant variables, rather than the inclusion of more. The results can thus be interpreted as a more conservative identification of significant variables when not accounting for collinearity. Due to the large number of combinations for the meteorological dataset ($n = 1480$ for the spatial analysis including MODIS SC), we expect randomly omitted predictors to show up in a different combination and not to be completely omitted."

and

l.325: "Snow cover increases the total amount of significant correlations only for Central Tien Shan for $\mathrm{MB}_{Barandunetal}$ and for Eastern Tien Shan, Pamir-Alay, an Western and Eastern Pamir for $\mathrm{MB}_{Hugonnetetal}$, despite contributing to at least half the significant combinations for most regions. For all other regions, T and P provide more significant correlations when SC is not considered. Combinations with SC barely show significant correlations in Eastern Pamir and Dzungarsky Alatau (Fig. 7). Thus, incorporation of SC does not notably increase the number of significant correlations overall. Instead, a shift of important variables from P and T to SC occurs. This distinct shift from P and T to SC suggests that potential collinearity does not bias the general outcome."

————

*Summarizing the scenarios in the discussion: These potential scenarios are interesting but lengthy and difficult to follow for readers not familiar with the particularities of the region. Could these be synthetized in a figure and streamlined? A short summary of the main differences would also be welcome.*

We provide a figure summarizing the two analyses in the form of a map. The map shows the most important drivers identified for the specific region for each analysis. We shortened the text accordingly and synthesize instead what the figure shows more accessibly. Please refer to the manuscript for the changes.

————

*Moving forward: This is very briefly mentioned in the abstract only (as far as I can tell). I was a bit frustrated that there were not more discussions on this – how should one then proceed to interpret the mass balance patterns? What possible other tools could be used, what additional data should be collected?*

We added a more elaborated part on possible solutions to improve the current situation. It covers in more detail instrumentation, advanced techniques for downscaling, holistic energy and mass conserving modelling approaches, as well as remote sensing validation data from independent data sources.

l.537: "Cryological, hydrological, and meteorological monitoring at high elevation throughout the different subregions of Tien Shan and Pamir need to improve to better understand glacier response to climate change. In several subregions, no glaciers are systematically monitored (Barandun et al., 2020), *in situ* snow monitoring is not established, and hydrological monitoring is very limited (Unger-Shayesteh et al., 2013; Hoelzle et al., 2019). Meteorological measurements are underrepresented at high elevation and, for certain subregions, completely lacking (Unger-Shayesteh et al., 2013; Sorg et al., 2012). Most existing data are inaccessible. The needed long-term systematic monitoring of the different components of the water cycle often lacks financial support, know-how, and man(woman)power (Hoelzle et al., 2019; Barandun et al., 2020). When using only one specific dataset, differences in reanalysis products and elevated uncertainties tied to the available annual mass balance time series for Tien Shan and Pamir underpin the ambiguity in the interpretation of the results of a correlation analysis. As highlighted by the two scenarios presented in Section Discussion, any derived understanding of glacier–climate interactions depends strongly on the dataset used. When a thorough quality assessment of the different reanalysis products for further application cannot be assured, the limitations of these datasets cascade into an increase of uncertainties. This especially affects the modelling of future glacier response, hydrological modelling, and the

equifinality problem related to too many unknowns in the cryohydrological cycle (e.g. Beven, 2006; Farinotti et al., 2015) or reconstruction of glacier mass change without a good calibration dataset. Traditionally, many studies have been based on a single reanalysis dataset, either used for bias correction or directly as model input, and direct calibration data are generally limited to a few individual locations, insufficient for regional applications. Remote sensing can partly bridge the gap in observational data, and progress is made to increase temporal resolution of geodetic mass balance estimates (Beraud et al., 2022). Furthermore, integration of different and sometimes unconventional datasets with information on the atmospheric conditions, the Earth surface energy balance, glacier response, and other water storage changes (Farinotti et al., 2015; Pohl et al., 2017; Naegeli et al., 2022; Key et al., 1997) is valuable to asses and possibly quantify uncertainties related to the presented datasets and, eventually, to rate their quality. Modern downscaling techniques such as those provided in Fiddes and Gruber (2014, 2012) facilitate the inclusion of small-scale topographic effects. Standardizing sophisticated downscaling methods can improve the spatial representations of the reanalysis datasets. Despite potential improvements (remote sensing (Beraud et al., 2022), integration of unconventional datasets (Farinotti et al., 2015; Pohl et al., 2017; Naegeli et al., 2022; Key et al., 1997), and modern downscaling technics Fiddes and Gruber (2014, 2012)), we recommend not to rely on a single product for either correlation analysis or cryohydrological modelling."

————

**1.2 Minor revision**

**1.2.1 Abstract**

*Overall, I find that the abstract could be streamlined and the main message made clearer.*

We changed the abstract to provide a clearer and better streamlined main message.

"We have investigated the drivers behind the observed spatiotemporal mass balance heterogeneity in Tien Shan and Pamir, in High Mountain Asia. To study the consistency of the different interpretations derived from the available meteorological reanalysis and remote sensing products, we used correlation analyses between climatic and static drivers with novel estimates of region-wide annual glacier mass balance time series. These analyses were performed both spatially using different spatial classifications of glaciers and temporally for each individual glacier. Our results show that the importance of the variables studied depends strongly on the dataset used and which spatial classification of glaciers is chosen. This extends to opposing results using the different products. Even supposedly similar datasets lead to different and partly contradicting assumptions on dominant drivers of mass balance variability. The apparent but false consistencies across studies using a single dataset are related, according to our results, to the chosen dataset or spatial classification rather than to the processes or involved environmental variables. Without a glaciological, meteorological, and hydrological *in situ* observation network providing data that allow the direct calibration and validation of extensive datasets, our understanding of the changing cryosphere at the regional scale for Tien Shan and Pamir cannot improve, neither can our understanding of glacier response to climate change or the assessment of water availability for the region's growing population."

————

*L3-4: 'Meteorological analysis, remote sensing products and novel approaches . . . all provide . . . '*

We adjusted this accordingly.

———

*L9: 'only ... do we find'*

We have changed this sentence while reworking the abstract.

———

*L13-14: This feels like a repeat from above*

We shortened the statement.

———

*L16-18: this part is barely mentioned in the discussion and could be developed more.*

We agree with this comment. Please see our answer and suggested modification outlined in the major comment above.

———

Introduction

*In general, the introduction is interesting and well written but I think it would benefit from some reorganization efforts and some streamlining to make the message clearer.*

We restructured the introduction as shown below in the specific comments.

———

*L25-26: This sentence does not bring much and could be removed. Are the two references to Gerlitz et al., 2019, 2020 really needed here?*

We restructured the first paragraph of the introduction.

l.15ff: "Glaciers across Tien Shan and Pamir, in High Mountain Asia, have been observed to change heterogeneously in space (e.g., Barandun et al., 2021; Shean et al., 2020; Brun et al., 2017; Miles et al., 2021). Under the assumption of equal climatology, glacier morphology has been found to explain up to 36% of the mass balance variability for Tien Shan, 20% for Pamir-Alay, and only 8% for Western and Eastern Pamir (Brun et al., 2019). Thus, local topographic and glacier-specific morphological characteristics cannot wholly explain the diverse glacier responses to climate change (Fujita and Nuimura, 2011; Brun et al., 2019); these are also related to sharp contrasts in the local climatological settings, mainly to their different mass balance sensitivities to climate (Sakai and Fujita, 2017; Wang et al., 2019) (responsible for up to 60% of spatially contrasting glacier response in

High Mountain Asia (Sakai and Fujita, 2017)). Mölg et al. (2014) and Farinotti et al. (2020) related a spatially heterogeneous glacier response for selected mountain ranges to different weather pattern constellations. These are reported to have changed in the past Gerlitz et al. (2020), leading to increased climate variability. (de Kok et al., 2020) argues that increased evapotranspiration might explain positive mass balances for solid precipitation sensitive glaciers."

————

*L32-34: This feels out of place.*

We rephrased the statement (see comment above).
————

*L20-36: There are lots of ideas in this first paragraph but the logical links are missing. These need to be reorganized/structured.*

We reorganized this part (see the answer of the two comments earlier).
————

*L46: 'Barandun et al. (2021) have applied'*

We adjusted this accordingly.

————

*L44-48: I am not sure that many details are necessary here, especially as these are described in length in the methods.*

We agree and shortened the introduction.

l.33:"Barandun et al. (2018) highlighted hot spots of spatiotemporal heterogeneity and increased mass balance variability in the different mountain ranges of the Tien Shan and Pamir at annual temporal resolution; however, their results were not purely observation based."

————

*L59-60: Are these details really necessary here? A simple reference to Hugonnet et al. (2021) is likely enough.*

We rewrote this paragraph and the statment was shortened.

l.48:"The spatiotemporal comprehensiveness of reanalysis data facilitates the inclusion of various climatic variables at global scale in correlation analysis that are otherwise not available from simple meteorological stations

or remote sensing/interpolation data products (e.g. Hugonnet et al., 2021)."

————

*L61: something wrong with the English at the end of this sentence*

We rephrased.

l.51: "Many glaciers in Central Asia are heavily debris covered with considerably different debris thickness (Kraaijenbrink et al., 2017; McCarthy et al., 2021)".

————

*L63: There are actually some region-wide debris thicknesses assessments. See Rounce et al. (2021) and McCarthy et al. (2022)*

We rephrased the statement and added the suggested reference.

l.51ff: "Many glaciers in Central Asia are heavily debris covered with considerably different debris thickness (Kraaijenbrink et al., 2017; McCarthy et al., 2021). A scale-dependent debris cover - mass balance relationship and limited region-wide debris thickness assessments restrict the explanatory power of debris cover for region-wide glacier mass balance patterns in Central Asia (Brun et al., 2019; Miles et al., 2022).

————

*L67: references missing. A recent one could be the work by Glasser et al. (2022) - 10.1016/j.geomorph.2022.10829*

We added the reference.

————

*L67-69: Not sure these details are necessary here.*

We shortened the statement.

l.57"Guillet et al. (2022) show that no significant difference exists in mass balance between surge- and nonsurge-type glaciers."

————

*L69: Have the authors considered avalanching as a possible morphological control? It feels like for some of the steeper ranges of the region that could actually play a significant role (Brun et al., 2019)*

This is a good point. Unfortunately, it is not straightforward to accurately quantify the avalanche contribution to mass balance and hence its importance as morphological driver. We added this point to the sections where we introduced the potential morphological drivers and pointed out in the method section that we do not integrate it in our analysis.

l.58ff: "Avalanching represents another nonclimatic factor that influences glacier mass balance through mass redistribution. The quantification of its effect on glacier mass balance at regional scales is, however, not straightforward. As an approximation, Brun et al. (2019) used the avalanche contributing area as potential morphological control. However, the authors found no significant correlation with their mass balance estimates for almost all regions assessed in High Mountain Asia."

l.171: "We do not include other variables such as avalanche processes due to insufficient data availability."

————

Data

*L87: Study site should be plural. No need to capitalize the nouns in the titles.*

We adjusted this accordingly.

————

*L121: I don't think this acronym has been defined before in the main text.*

We introduced the acronym in the text.

l.115: "In the present work, we choose a subdivision of the regions based on the commonly used Hindu Kush Himalayan Monitoring and Assessment Programme (HiMAP) regional division suggested in Bolch et al. (2019)."
————

*L124: Simple reference to RGI v6.0 is enough here (text can be shortened)*

We adjusted this accordingly.

————

*L134: The Barandun et al. (2021) dataset also uses geodetic mass balance products to calibrate their model (second order calibration).*

We reworked and shorten the entire paragraph. Please see to the outlined changes in the following comments.

————

*L135: Suggest adding reference to Figure 1 here.*

We adjusted this accordingly.

————

*L134-159: Are all the details provided here really needed considering that these are already published approaches?*

We agree and shortened this paragraph and add a few lines on the uncertainties.

l.131ff: "In the present study, we use the two existing annually resolved glacier-specific datasets for glacier mass balance estimates covering the entire Tien Shan and Pamir: one based on transient snowline observations and geodetic mass balances; the other, on digital elevation model (DEM) differences (Fig. 1).
The first dataset ($MB_{Barandunetal}$) comprises the annual time series provided by Barandun et al. (2021), who used a mass balance model calibrated simultaneously with transient snowlines (as a proxy for surface mass balance) and geodetic mass changes. The model driven with ERA-interim reanalysis data (Dee et al., 2011) for each glacier and year separately (Barandun et al., 2021, 2018). ERA-interim was chosen because, unlike other reanalysis products (e.g., ERA5), *in situ* observations in mountain regions are assimilated (Orsolini et al., 2019). Barandun et al. (2021) provided annual mass balance time series, closely tied to direct observations, with low sensitivity to meteorological input for roughly 60% of the glaciers larger than $2\,km^2$ in the data-sparse Tien Shan and Pamir. For more details on the methodological approaches for mass balance determination and model sensitivity the reader is referred to Barandun et al. (2018) and Barandun et al. (2021); for the automatic snowline mapping, to Naegeli et al. (2019); and for the geodetic estimates, to Girod et al. (2017) and McNabb et al. (2019). Differences in ERA-interim and ERA5 performance and output at high altitudes are highlighted in Orsolini et al. (2019) and Liu et al. (2021), showcasing the independence of the two datesets.
The second dataset ($MB_{Hugonnetetal}$) comprises the geodetic mass balance estimates by Hugonnet et al. (2021), which rely on DEM differences and filtering techniques. The predominant input is the 20-year-long archive of stereo images from the Advanced Spaceborne Thermal Emission and Reflection Radiometer (ASTER) used to derive time series of DEMs. Their estimates were validated at all glaciers world-wide with intersecting laser altimetry and elevations derived from high resolution optical imagery data. The approach shows good agreement at the global scale although having varying uncertainties at the regional scale (Hugonnet et al., 2021) due to a sometimes limited performance of the method for individual glaciers (Hugonnet et al., 2021). Geodetic methods are more accurate at longer time scales as the signal-to-noise ratio improves. However, Hugonnet et al. (2021) also provide annual estimates connected to much higher uncertainties pointed out by the authors, which we use in this work. We select only the glaciers from this dataset for which estimates from Barandun et al. (2021) are available. This serves for a consistent analysis of dominant drivers and to highlight the differences between the two mass balance estimates (Fig.2).
Both datesets are tied to elevated uncertainties of the annual mass balance estimates. Barandun et al. (2021) adopted mean uncertainties provided in Barandun et al. (2018) ($\pm 0.32\,m\,w.e.\,yr^{-1}$) associated with the snowline-constrained mass balance modelling and combined them with the error estimate from the geodetic surveys. This resulted in a rather conservative uncertainty of $\pm 0.37\,m\,w.e.\,yr^{-1}$ and does not assume independence of the errors from year to year. Hugonnet et al. (2021) reported uncertainties of up to $\pm 0.1\,m\,w.e.\,yr^{-1}$ for mean annual mass balance values of 5-year periods. Hugonnet et al. (2021) provide uncertainties in mass changes for periods shorter than 5 years only for the global or near-global estimates. Global annual mass balance uncertainties are reported to be around $\pm 0.2\,m\,w.e.\,yr^{-1}$ (Hugonnet et al., 2021), and we expect higher ones for the mass balance time series

of individual glaciers."

————

*L141: (Dee et al., 2021) should come after 'data'.*

We adjusted this accordingly.

————

*L148: 'observational' is not really true for Barandun et al. (2021) MB, as it comes from modeling.*

We removed this statement due to the shortening of the paragraph.

————

*L155: can you explain a bit better the sentence 'local and regional scale biases can persist'? A reference might be needed here.*

We reworded this. The sentence is supposed to say that the approach is sometimes performing badly at the single glacier scale (Hugonnet et al., 2021; e.g. Extended Data Fig. 5d) and shows varying uncertainties at the regional scale (e.g. Hugonnet et al., 2021; e.g. Extended Data Fig. 5d).

l.144: "The approach shows good agreement at the global scale although having varying uncertainties at the regional scale (Hugonnet et al., 2021) due to a sometimes limited performance of the method for individual glaciers (Hugonnet et al., 2021)."

————

*L181: Could you also provide the resolution in km for consistency with the other datasets?*

We included this now.

————

*L205-209: I would recommend putting this in a separate subsection.*

We added a subsection.

————

*L240-241: I struggle a bit with this k-mean clustering. Could you give a few more details.*

We added some more details on the k-mean clustering method with a standard reference.

l.264ff: "The latter is determined using a k-means clustering with three fixed classes, where points are classified by minimizing their squared distance to the iteratively determined cluster centers (Fig. 5). The k-means clustering is based on the standard deviation of the mass balance time series of Barandun et al. (2021) and relies on an unsupervised classification that avoids a precise association of a class with, for example, a process. We arbitrarily chose a number of three clusters that would highlight the regions of high mass balance variability (hot spots) identified in Barandun et al. (2021). The division is performed to determine whether different dominant predictors exist for these regions and, in general, to assess the effect of arbitrarily subdividing glaciers into different groups."

————

Results *L276: A3 should come before A4 in the text.*

Figure order was changed.

————

Discussion
*L357: remove the comma*

We adjusted this accordingly.

————

*L359-360: this makes sense as they are likely related*

Yes, we agree. This is also discussed later in the manuscript, as mentioned in one of the earlier comments.

————

*L434: snow cover decrease reported by*

We adjusted this accordingly.

————

*L489-490: missing parenthesis*

We adjusted this accordingly.

———

Conclusion

*L543-544: Use lower cases for Barandunetal and Hugonnetetal*

We adjusted this accordingly.

———

Figures

*The panels of the different figures need to be numbered.*

We added numbering to the figures.

———

*Figure 1*

- *Instead of a globe, a map of HMA would be sufficient and more informative here.*

- *Shouldn't the mass balance should be in m.w.e? Also for consistency with the text.*

- *In general I don't really like the term 'surface' mass balance here – at least for Hugonnet et al., 2021, these are geodetic mass balance measurements. I would suggest sticking to 'glacier' mass balance.*

We agree on all points and adjusted labels in figures, and all occurrences in the text accordingly.

———

*Figure 2*

- *Why are the spatial and temporal analysis linked in the figure? Aren't they done independently from one another?*

- *It would be good to distinguish the data boxes and the methods boxes. Having the seasonal aggregation in the same box as the monthly meteorological time-series is confusing.*

They are independent and we are thankful for the suggestions and changed the figure according to the suggestions.

———

*Figure 3*

- *Could you specify what each cluster corresponds to in the figure? There are likely many ways of clustering this data, which criteria were used here and why?*

As mentioned briefly in the text this is based on a k-means clustering algorithm. It is an unsupervised classification, meaning that there is not a precise association of a class with e.g. a process. The k-means clustering is based on the standard deviation of the mass balance time series of Barandun et al. Therefore, classes represent glaciers with different variability. We clarified this in the figure caption and in the text (see also comment above).

l.264: "The latter is determined using a k-means clustering with three fixed classes, where points are classified by minimizing their squared distance to the iteratively determined cluster centers (Pedregosa et al. (2011); Fig. 5)."

––––––––

*Figure 4,5 & 6*
*Most of the comments here hold for the next figures as well:*

- *Suggest writing out SC, P, T*

- *Do these meteorological correlations relate then to trend, mean, STD?*

- *Which variables do the seasons refer to? This would need to be specified in the caption, and maybe even in a supplementary figure?*

- *Specify in the caption that 'short' and 'long' refer to the 2000-2014 and 2000-2018 periods.*

- *It needs to be specified also that this is an aggregation of all glaciers in the region.*

We wrote out the abbreviations and increase their size in the figure. The correlations are indeed incorporating all derived statistics. As explained also in the downscaling section, this allows to find relationships where there is e.g. an altitudinal bias. It also allows us to use the meteorological data in both the temporal and in the spatial analysis. The latter requires a single value for any given glacier, whereas the former utilizes the time-series as it is. We clarified this and the other points in the text and also in the figure captions. Please refer to the major comments for changes made in the manuscript.

––––––––

*Figure 7 & 8*
*These comments also hold for figure 8:*

- *Stay consistent with acronyms in figures*

- *The explanatory diagram does not need to be repeated in every subplot if it stays the same*

- *Where are the subplots with and without snow cover? I cannot find the legend.*

We removed all explanatory pie diagrams but one and change the labels to be consistent with the other figures. The subplot without the snow cover was moved to the Appendix and we missed changing the caption accordingly. The figures are in A6. We adjusted the figure caption.

————

*Figure A4*

- *Would there not be a way to weight the results of the right panel by the number of glaciers?*

The only thing we want to show with this figure is that we did not identify any different relationship between surge- and non-surge-type glaciers with mass balance among the different subregions. It would certainly be possible to weigh the mass balance estimates but the main aim is really to show that there is not a systematical higher or lower value for surge-type glaciers. We clarified this in the caption.

Figure caption: "Distribution of surge-type glaciers and mass balance range of all glaciers (left) and for the HiMAP regions (right). The regional comparison shows that no systematic difference was found between the relationship of surge- and nonsurge-type glaciers with mass balance."

**2  Reviewer #2**

**2.1  Major revision**

*I think the authors need to restructure their objectives to be something they more clearly address:*

*Objective 1 seems to encompass two separate research questions about the drivers and physical processes themselves, and then the degree to which this is simply a result of which data are used to explore the problem.*

*Objective 2 is related to the latter part of the first objective in my opinion and additionally implies that the authors explicitly compared ground truth data for correlation analysis. I'm not convinced that the authors really test enough the limitations of the gridded data in this manuscript (so objective 1, part 1), or at least not from the data/figures presented.*

We agree that the description of the objectives was not well chosen. As it was written, one would indeed expect a more detailed analysis, and more importantly, conclusive results on the interplay between climate and mass balance. Instead, we finally provided what the reviewer correctly points out as second part of Objective 1. The motivation of our work is, nevertheless, a better understanding of the climate-mass balance relationship. This is why we chose the unfortunate wording in L.70 ff. We have reworded in particular this section to differentiate motivation and resulting objective. The objective will more clearly state that rather than revealing dominant factors, we will explore the relationships using the newly available datasets. This now addresses the question if we can identify consistent relationships between mass balance and climatologic/morphologic drivers.

We addressed the limitations critique by providing a pre-analysis of the datasets used. This was done following the suggested graphs in the style of de Kok et al. (2020), and by comparing the

mass balance datasets using the already introduced aggregated pie chart maps.

l.74ff: "The reasons abovementioned outline a lack of consistent understanding regarding the climatic and nonclimatic drivers of glacier mass balance variability in Tien Shan and Pamir. Therefore, a more comprehensive and rigorous analysis of the available datasets is indispensable to provide more conclusive and accurate results and interpretations on the drivers of glacier response to climate change in Central Asia.

Therefore, we aim in this study for a rigorous analysis of different datasets to identify similarities and differences in the drivers found to explain the glacier mass balance changes in Pamir and Tien Shan, with the ultimate goal of advancing the understanding of the drivers behind heterogeneous mass balance changes in Central Asia. Our analysis benefits from newly available and advanced highly temporally resolved mass balance estimates and new reanalysis products. Given the often unconstrained uncertainties in climatological / meteorological datasets for data-sparse regions, we analyze (1) the consistency of the different meteorological and mass balance datasets and (2) which variables can explain in a statistically significant manner the variability found in mass balance datasets. We follow a systematical approach to test three different reanalysis products that are or have been used extensively in the region. Additionally, due to existing differences in glacier mass balance time series, we incorporate the two available annual mass balance estimates for the region: the snowline-aided estimates by Barandun et al. (2021) and the geodetic mass balances by Hugonnet et al. (2021). Mass balance time series are related to the most commonly used climatic variables temperature (T) and precipitation (P) from the reanalysis datasets, snow cover (SC) from a remote sensing product, and glacier-specific topographic and morphological characteristics. Finally, to account for possible regional data issues or arbitrarily chosen regional divisions, the analyses are performed at different spatial subsets."

and

l.284: "A comparison of the different reanalysis datasets is shown for precipitation in Figure 3 and for temperature in Figure A2. Only CHELSA provides a spatial resolution to resolve intramountain range variability. Differences in precipitation amount and different trends exist independently of the spatial resolution (Fig. 3). However, for temperature, CHELSA is the temperature-conserving downscaled ERA5 product with matching trends and a correlation of one (Fig. A2i). For precipitation, CHELSA and ERA5 show deviating distribution patterns and trends, highlighting the nonlinear downscaling method (Fig. 3). The use of the datasets in their original spatial resolution in consecutive analyses allow for an assessment of how downscaling affects the correlation with mass balance data."

————

*It is clear that different reanalysis datasets will produce a different correlation with the same mass balance product. While the authors perform a nice assessment of these expected differences, I'm left wondering how comparable the forcings really are in the first place. The authors apply ERA5 (approx.. 36km), HAR30 (30km) and CHELSA (1km, using ERA5 to downscale). While the first two are closer in terms of spatial resolution and the processes they can represent at the surface, CHELSA represents a major effort to include finer scale topographical features, such as windward/leeward precipitation adjustments, even though it likely remains with significant seasonal biases. Accordingly, I would not expect to find a similar relationship to glacier mass balance (spatially or temporally) when comparing to ERA5 or HAR30 (e.g. L286-7, L343-4), which cannot represent precipitation dynamics at the scales relevant to glacier processes (e.g. L401).*

*The authors have given some consideration to these limitations in their discussion section, but I'm not convinced that it is a fair test and fully supports the conclusions about an ambiguity*

*to glacier response in the region. I think that redoing analysis with additional datasets would also not necessarily advance the findings of this manuscript and perhaps cloud the main messages. However, I do think that the authors need to address these discrepancies and perhaps even remove CHELSA from the analysis. More importantly, I would like to see some level of comparison between the different products for air temperature and precipitation estimates (especially if CHELSA remains included). What explains the presence or lack of correlations in given subregions, for example? Some supplementary figure(s) would be useful to that end.*

A more detailed answer to this valid and important point regarding the spatial resolution can be found in an answer to reviewer 1 about downscaling. We argue that using the relatively coarse datasets alongside the fine resolution CHELSA dataset is a valuable and valid approach for what we want to show. Reviewer 1 was concerned about the elevation bias that results from not downscaling the data. The coarse data provide ultimately the same data for multiple glaciers. Here we argue twofold: 1) a systematic offset in the independent (meteorological) variable will have no impact on the outcome of the temporal analysis (significant relationship or not) because we perform a correlation analysis, where a determined significance is independent of a linearly scaled variable; 2) this holds true for the spatial analysis because the statistics derived from the meteorological time-series that serve as independent variable reflect temporal components (standard deviation, trend) and thus the same applies as for point (1).

However, this does not eliminate the problem that there are the same time-series or the same derived statistic for multiple glaciers encompassed by a single grid cell. Finding a relationship might be prevented if the glaciers within the extent of a grid cell show a stronger variability in mass balance than between different grid cells. This is, however, not the case for the entire study region, or the larger regional subsets, as we show in our study (e.g. Fig. 4,5,6). For smaller regional subsets, as shown particularly in Fig.5, this might have actually been the reason for the lack of significant correlations. However, because we also use CHELSA, we can see that this is not necessarily a spatial resolution problem. We argue that this can be related to changing variable importance that we do not cover in our study, e.g. a transition into more radiation-dependent processes (Sec. 5.1, L.367 f.) or differences in small scale processes important in different regions. This assessment is supported by having CHELSA as an additional fine spatial resolution dataset in our analysis.

We do agree that having a better idea of how the different datasets compare to each other would provide a much-appreciated basis for any reader. The additional figure about meteorological datasets will show the differences and correlations now.

We added the outlined considerations in the Data, Results and Discussion sections.

l.192ff: "**CHELSA** time series version 2.1 is a processed and downscaled version of the ERA5 dataset which incorporates directional wind speeds and cloud cover observations to address precipitation biases and wind and leeward distributions (Karger et al., 2017). The final downscaled product presents a $\approx 1\,\text{km}$ spatial and monthly temporal resolution. By incorporating the CHELSA data, we provide a means to determine how a state-of-the-art downscaling affects the correlation analysis."

and

l.288ff: "A comparison of the different reanalysis datasets is shown for precipitation in Figure 3 and for temperature in Figure A2. Only CHELSA provides a spatial resolution to resolve intramountain range variability.

Differences in precipitation amount and different trends exist independently of the spatial resolution (Fig. 3). However, for temperature, CHELSA is the temperature-conserving downscaled ERA5 product with matching trends and a correlation of one (Fig. A2i). For precipitation, CHELSA and ERA5 show deviating distribution patterns and trends, highlighting the nonlinear downscaling method (Fig. 3). The use of the datasets in their original spatial resolution in consecutive analyses allow for an assessment of how downscaling affects the correlation with mass balance data."

and

l.391ff: "CHELSA is a processed and downscaled version of the ERA5 dataset. When using these two products, differences appear in the correlation analyses due to different spatial resolutions in the predictor variables; however, these differences only become striking when using the regional subdivision (Fig. 6). This might respond to a low number of data points with high mass balance variability within areas of encompassing meteorological grid cells, while the mass balance variability between the different encompassing grid cells in the regional subdivision is rather low. The nonlinearly downscaled precipitation of CHELSA (Fig. 3) compared with the conserving values for T (Fig. A2) can explain the higher number of significant correlations of this product compared with that of ERA5. As we use two predictor variables, the original P fields of CHELSA can change the significance of T. In the temporal analysis, the effect is even clearer, leading to partly opposing results (Fig. 9 and Fig. A8). We do not answer here how exactly any downscaling method can affect results but we show that advanced downscaling methods (CHELSA's precipitation) can severely impact driver interpretation. Different outcomes can be expected depending on the downscaling technique used, adding subjectivity to the results."

————

*Building on the previous point, I think the authors also need to address any potential circular issues related to the use of ERA-Interim for the derivation of the $MB_{Barandunetal}$ mass balance results and its comparison to the next generation of the ECMWF reanalysis.*

This point has been raised by both reviewers and we understand their concern. However we believe the use of the two datasets is justified in the way they have been used. The approach presented in Barandun et al. (2021) fortunately does not rely on precise meteorological input (which simply do not exist extensively in the high mountains of Central Asia). Barandun et al. (2018) showed that using transient snowlines for model calibration reduces the sensitivity to the meteorological input data. The sensitivity analysis in Barandun et al. (2018) uses an average temperature and precipitation dataset without any year-to-year variability still provided results with an RMSE of less than 0.2 m w.e. $yr^{-1}$ in comparison with the direct measurements. This highlights that the snowline observations used for calibration, for each glacier and year individually, are primarily responsible for the modelled mass balances. For more details, we believe it is sufficient to refer to the previous work Barandun et al. (2018) that provides an in-depth analysis of the model sensitivity which the current manuscript builds upon.

In Barandun et al. (2018) ERA-interim was used becauseOrsolini et al. (2019) showed the superior performance of ERA-interim for mountainous regions like High Mountain Asia over ERA5, mainly due to the assimilation of *in situ* station data in ERA-interim that is omitted in ERA5. The products have thus significant difference in the generated precipitation fields that are independent from each other. This provides the opportunity for further analysis of possible

correlations between the mass balance time series and the climatic drivers with ERA5 or other independent reanalysis datasets. Currently ERA5 is one of the most used reanalysis products due to its finer spatial resolution and often shortcomings as outlines above are not considered in correlation analysis. Despite the low model sensitivity we did not use ERA-interim for the correlation analysis. We added a justification of the dataset in the manuscript, explaining also the insensitivity of the mass balance time series to the meteorological input data.

l.134ff: "The first dataset ($MB_{Barandunetal}$) comprises the annual time series provided by Barandun et al. (2021), who used a mass balance model calibrated simultaneously with transient snowlines (as a proxy for surface mass balance) and geodetic mass changes. The model driven with ERA-interim reanalysis data (Dee et al., 2011) for each glacier and year separately (Barandun et al., 2021, 2018). ERA-interim was chosen because, unlike other reanalysis products (e.g., ERA5), *in situ* observations in mountain regions are assimilated (Orsolini et al., 2019). Barandun et al. (2021) provided annual mass balance time series, closely tied to direct observations, with low sensitivity to meteorological input for roughly 60% of the glaciers larger than $2\,km^2$ in the data-sparse Tien Shan and Pamir. For more details on the methodological approaches for mass balance determination and model sensitivity the reader is referred to Barandun et al. (2018) and Barandun et al. (2021); for the automatic snowline mapping, to Naegeli et al. (2019); and for the geodetic estimates, to Girod et al. (2017) and McNabb et al. (2019). Differences in ERA-interim and ERA5 performance and output at high altitudes are highlighted in Orsolini et al. (2019) and Liu et al. (2021), showcasing the independence of the two datesets."

————

*The temporal analyses produce clearer and more interpretable plots for this manuscript (which I liked), but I have concerns about the robustness of these results given their short temporal scale (14 years for correlations using HAR30) and high uncertainties for mass balances given annual data series. While the ideal minimum sample size required can vary based on the type of observations, it has generally been considered that sample sizes less than 30 can produce imprecise correlation estimates. I'm curious why the authors did not consider a few longer term reanalysis/gridded products of similar spatial resolution (e.g. HARv2, ERA5Land,WRF9km) that all provide data since 1980 at least. I guess there are also temporal limitations for the mass balance data period, but this should at least be discussed more clearly.*

This is a good point and we would have liked to use longer time-series. We agree fully that it would certainly make the analyses more robust. Our work is unfortunately restricted by the length of the mass balance estimates rather than the meteorological datasets. But, we were also keen to use the HAR version 1 dataset as it is one of the best performing datasets in the region that has been shown to actually resolve the discharge variability over the greater Pamir domain (Pohl et al., 2017), and also over the Tibetan plateau (Biskop et al., 2016b). Comparison between HAR version 1 and version 2 (not shown here) show that these datasets are very different. This is not surprising as version 2 uses ERA5 as input data. We certainly intend to raise some awareness about what looks like a tendency to more and more use ERA5 and ERA5-land directly - or as basis for further downscaling (HAR v2, CHELSA)- because it is so convenient with its long temporal coverage and good spatial resolution. With our choice of datasets we were hoping to provide simultaneously two critical views on the climate-glacier interaction topic. As the comparison between CHELSA and ERA5 shows, the downscaling has a significant impact on the obtained results. By incorporating HAR version 1, with a similar spatial resolution as ERA5, we also have a comparison between different forcing datasets for the

downscaling. Without that, the main deductible conclusion would be that downscaling affects the outcome.

We added an additional statement on our justification for using the indeed rather short HAR version 1.

—————

l.196ff: "**HAR30** version 1.4 (Maussion et al., 2011) presents a 30 km spatial and a daily temporal resolution. Data result from a dynamical downscaling of global analysis data (Final Analysis data from the Global Forecasting System (National Centers for Environmental Prediction, National Weather Service, NOAA, 2000); dataset ds083.2) using the Weather Research and Forecasting–Advanced Research WRF (WRF-ARW) model (Skamarock and Klemp, 2008). The dataset has shown high consistency with temperature, precipitation, and snow cover measurements in several regions of High Mountain Asia, where uncertainties are especially large due to limited or nonexistent meteorological stations. This dataset has captured climatic extremes in the greater Pamir region that lead to floods and droughts (Pohl et al., 2015), has been used for glacier studies in the greater Himalayan region (Maussion et al., 2011; Mölg et al., 2013; Curio et al., 2015), and has been applied in hydrological studies (Pohl et al., 2015, 2017; Biskop et al., 2016a). Despite the need to bias-correct HAR precipitation intensities in Pamir (Pohl et al., 2015) and Tibetan Plateau (Biskop et al., 2016a), the dataset has shown superior correlation with *in situ* measurements than interpolated and remote sensing datasets in Pamir (Pohl et al., 2015). Due to large uncertainties in gridded meteorological datasets, we believe that the HAR dataset, validated at least to some degree in High Mountain Asia in several previous studies (e.g. Pohl et al., 2015, 2017; Biskop et al., 2016a; Maussion et al., 2014), should be included in our analysis. We do not use HAR version 2 because ERA5 is used as input to downscale the version 2 output and shows strong differences with the validated HAR version 1 dataset."

—————

*Plotting such large datasets can be challenging and I think the authors have done a great job in many respects. I think there are several areas where they might still be improved for clarity and to help the message of the manuscript. I have given some specific comments below.*

Thank you for the comment. We addressed the specific comments below and refer to the answers given to each specific comment.

—————

*As mentioned, the work is generally well written. However, there are several instances where the wording is unclear or with grammatical errors. I have tried to provide some examples of this below, but the authors should carefully check the entire manuscript at the next submission.*

We went through the specific comments and included the changes in the updated manuscript. We have also decided to ask for a professional proofreader to eliminate unclear wording.

—————

**2.2 Specific Comments**

*L90+ : I think that the authors provide some very valuable and interesting information about the climatic setting in this paragraph, but I do not feel that the discussion reflects enough upon these interesting variations. Building upon my main point about comparisons of the gridded datasets, the authors should present spatial maps of mean (summed) temperature (precipitation) from different products, or better yet, the spatial trends of some of these variables (where are they statistically significant for ERA5/HAR30?). Exact comparisons can be challenging due to differences in spatial scale, but the authors could consider spatially binning the grids as presented, for example, in de Kok et al. (2020).*

As mentioned in other comments, we now provide such a figure.

————

*L185-188: The authors state that snow depth over-estimation in ERA5 renders use of radiation variables "problematic", and rely upon only temperature and precipitation data. However, there are well reported cold biases for air temperature at high elevation regions (such as those in this manuscript) which also result from this albedo-effect (e.g. Wang et al. 2020).*

For any type of correlation analysis, a systematic and linearly scaled over-/underestimation does not pose a problem. Our main reason for the choice of the two main variables temperature and precipitation is, however, that they are widely available. For reanalysis with meteorological station data assimilation of these two variables should also be better constrained than others, which are rarely or not at all measured at meteorological stations or not retrieved by other means.

We clarified this.

l.188: "This renders the parameters for in- and outgoing energy fluxes problematic, nor are they consistently measured at the few meteorological stations in High Mountain Asia, and we focus only on the two variables P and 2 m T. For our correlation analysis, a systematic and linearly scaled over-/underestimation does not pose a problem. Consequently, we focus on P and T for the other datasets too."
————

*L192: Given that the authors test the latest products from ECMWF (ERA5) and CHELSA, I'm surprised that they did not also consider the HARv2 product which is openly accessible and available back to 1980 (thus eliminating the shorter temporal focus and affecting the strength of correlations). I think a line should be added somewhere to justify the use of these three datasets specifically and why they are considered comparable, especially given the points from my main comment above.*

This point is largely answered in a previous comments. We addressed this in the data section with more rigour.

l.196ff: "**HAR30** version 1.4 (Maussion et al., 2011) presents a 30 km spatial and a daily temporal resolution. Data result from a dynamical downscaling of global analysis data (Final Analysis data from the Global Forecasting System (National Centers for Environmental Prediction, National Weather Service, NOAA, 2000); dataset

ds083.2) using the Weather Research and Forecasting–Advanced Research WRF (WRF-ARW) model (Skamarock and Klemp, 2008). The dataset has shown high consistency with temperature, precipitation, and snow cover measurements in several regions of High Mountain Asia, where uncertainties are especially large due to limited or nonexistent meteorological stations. This dataset has captured climatic extremes in the greater Pamir region that lead to floods and droughts (Pohl et al., 2015), has been used for glacier studies in the greater Himalayan region (Maussion et al., 2011; Mölg et al., 2013; Curio et al., 2015), and has been applied in hydrological studies (Pohl et al., 2015, 2017; Biskop et al., 2016a). Despite the need to bias-correct HAR precipitation intensities in Pamir (Pohl et al., 2015) and Tibetan Plateau (Biskop et al., 2016a), the dataset has shown superior correlation with *in situ* measurements than interpolated and remote sensing datasets in Pamir (Pohl et al., 2015). Due to large uncertainties in gridded meteorological datasets, we believe that the HAR dataset, validated at least to some degree in High Mountain Asia in several previous studies (e.g. Pohl et al., 2015, 2017; Biskop et al., 2016a; Maussion et al., 2014), should be included in our analysis. We do not use HAR version 2 because ERA5 is used as input to downscale the version 2 output and shows strong differences with the validated HAR version 1 dataset."

———————

*L205: I'm not convinced that the MOD10CM product at a 5km resolution does much to aid the analysis presented here. Given its spatial resolution, can it reasonably represent anything meaningful at the scale of the glaciers? Please add some additional information*

Even though the resolution of MOD10CM is not matching the individual glacier scale it provides a robust small (5km) scale snow cover evolution time-series. We chose this product for the robustness (averaging out falsely classified pixels and cloud cover issues) with the intention to have a general overview of snow patterns. Similar to the coarse ( 30km) ERA5 and HAR products, we use either the time-series directly in the temporal analysis or the statistics that provide information derived from the temporal domain (standard deviation, trend, and also snow cover change per time step) for the spatial analysis. Many times the high spatial resolution MODIS snow cover products provide either time-invariant signals at the highest resolution (250 m and 500 m), or slightly varying values suggesting changes in winter where these changes are more likely to reflect measurement or classification uncertainties than actual changes.

In our analysis we use snow cover mainly to check the results for plausibility, i.e. if we can substitute an interplay of precipitation and temperature (leading to snow cover) with an actual estimate for snow cover. As such, the resolution closer to the coarse meteorological datasets is likely better suitable than a fine resolution dataset that would capture small scale processes that certainly are not represented in the meteorological datasets.

An advantage of the coarse resolution is certainly that changes over time are more clearly captured. This is because the averaging over a larger area will detect a decline or increase in snow cover due to the more variable evolution at lower elevations. With other words, we obtain a better signal-to-variability ratio. In the end, this is a trade-off. But we think that the use is justified for the presented work and our idea of testing the obtained variable importance for plausibility.

We incorporated these thoughts into the Data and Discussion sections.

l.250ff: "We do not run a separate collinearity test to identify and remove correlating predictors even though this is expected, for example, in the case of different temporal temperature aggregates or snow cover and precipitation data (Fig. 4). As our goal is to reveal possible inconsistent outcomes depending on the chosen glacier mass balance

and meteorological dataset, we focus only on identifying significant predictors rather than model coefficients. A possible impact in case of collinearity is a nonsignificance of one or both correlating predictors (Vatcheva et al., 2016); thus, an anticipated effect of collinearity is the concealing (removing) of significant variables, rather than the inclusion of more. The results can thus be interpreted as a more conservative identification of significant variables when not accounting for collinearity. Due to the large number of combinations for the meteorological dataset ($n = 1480$ for the spatial analysis including MODIS SC), we expect randomly omitted predictors to show up in a different combination and not to be completely omitted."

and

l.325ff: "Snow cover increases the total amount of significant correlations only for Central Tien Shan for $\text{MB}_{Barandunetal}$ and for Eastern Tien Shan, Pamir-Alay, and Western and Eastern Pamir for $\text{MB}_{Hugonnetetal}$, despite contributing to at least half the significant combinations for most regions. For all other regions, T and P provide more significant correlations when SC is not considered. Combinations with SC barely show significant correlations in Eastern Pamir and Dzungarsky Alatau (Fig. 7). Thus, incorporation of SC does not notably increase the number of significant correlations overall. Instead, a shift of important variables from P and T to SC occurs. This distinct shift from P and T to SC suggests that potential collinearity does not bias the general outcome."

————

*L215-216 (and 243-245): I do not understand clearly how the authors relate the spatial means of mass balance with trends of meteorological variables. From the trend you will have just one value per gridded pixel, correct? So the authors combine all glaciers in a given binned region (circles in Figure 1) or just the entire region, and regress all the trends in a given variable (say temperature) at the corresponding pixel of each glacier against the mean or STD of mass balance at that glacier? I believe the authors need to more clearly explain this. Unfortunately, Figure 2 does not help much with this interpretation.*

We explained this in more detail in addition to the proposed changes from the previous comments about the coarse reanalysis resolution. We do not calculate a mass balance mean but instead use each glacier as one data point. This will - explained in the other comments - indeed lead to often multiple glaciers sharing the same meteorological data point. Regarding Figure 2, we do not have a direct solution for how to incorporate these ideas in the figure but we think that with our argumentation of why we think that our approach is sound (explanations in previous comments), we provide a clearer discussion on the this issue.

l.280ff: "We run multiple linear regression analysis on the glacier mass balance time series per glacier to identify significant predictors over the time domain. Annual glacier mass balance time series represent the dependent variable; combinations of meteorological variables from individual reanalysis datasets represent the predictors. As this yields one result per glacier, we aggregate the results visually in form of maps. We identify both significant variables and significant seasons to identify possible patterns and differences resulting from the use of a particular reanalysis or mass balance dataset."

and

l.241: "The coarse spatial resolution of HAR30 and ERA5 can prevent the finding of correlations in the spatial analysis if the mass balance intergrid variability is lower than or equal to the variability within a grid cell. Using ERA5 and its downscaled version CHELSA for comparison provides a means to assess whether this effect occurs."

and

l.394: "This might respond to a low number of data points with high mass balance variability within areas of encompassing meteorological grid cells, while the mass balance variability between the different encompassing grid cells in the regional subdivision is rather low."

————

*L221-222: Please re-write this sentence to improve syntax. It makes sense, but is not well written.*

We rewrote the statement.

l.227ff: "In this study, we focus strictly on the determination of significant predictors in multiple linear regression analysis and do not report, nor discuss, obtained model coefficients or coefficients of determination in detail. This decision responds to the large uncertainties present in the outcomes of the analyses, as shown in the results. Focusing on whether variables show or do not show significant correlations allow for a streamlined highlighting of important variables and seasons."

————

*L225-226: This is also a little unclear from the writing. Do the authors mean that they do not mix variables from different products in their analyses (e.g. temperature from ERA5 and precipitation from CHELSA)?*

Yes, exactly. We clarified this in the text.

l.238: "For the meteorological analysis, we constrain these combinations separately to each individual dataset product, i.e., only combinations from ERA5, or HAR, or CHELSA. However, we also extend this analysis by adding the MODIS SC data to each reanalysis datasets in a second step. This allows to explore whether the observational SC dataset adds more explanatory power than the reanalysis."

————

*L242: Please redefine here what is meant by these 'hotspots'. I note that this comes from the author's previous published work, but it would be better to clarify explicitly for the reader what is meant by a hotspot of mass balance variability. I interpret that as the areas in Figure 1 where STD is greatest, for example.*

We introduced what we mean with "hot spots" in the introduction.

l.33"Barandun et al. (2018) highlighted hot spots of spatiotemporal heterogeneity and increased mass balance variability in the different mountain ranges of the Tien Shan and Pamir at annual temporal resolution; however,

their results were not purely observation based.".

and

l.267: "We arbitrarily chose a number of three clusters that would highlight the regions of high mass balance variability (hot spots) identified in Barandun et al. (2021)."

―――――

*L253: The authors consider the ambiguity of the glacier response here as being largely due to different meteorological or mass balance datasets considered, but what about the uncertainties stemming from the annual-series of geodetic mass balances themselves, which are, to the authors own admission (e.g. L156), with less accuracy, and I assume, less precise? Again, it would be ideal to be able to assess the comparison of the different gridded meteorological and mass balance products independently to identify where the largest differences in the datasets lie, not just the correlations/significance.*

In accordance with the comment of reviewer #1, we have outlined the differences between the annual time series and highlight the usefulness of using the two different mass balance time series. We add also a statement on the uncertainties related to both mass balance time series. For a detailed answer on the uncertainties and proposed changes in the manuscript please refer to the answer to reviewer #1 for the detailed changes (major revision comment #2).

―――――

*L264: I think the authors should attempt to signpost some of the key findings a little better in the figures and simplify some of the main figures where possible (see specific figure comments below). I find it a little hard to navigate from text to figure to follow the story-line.*

We improved the figures of the paper and refer here to the answer of to specific comments.

―――――

*L344: The authors do not consider all available reanalysis datasets or atmospheric data for their analysis. Please rephrase this.*

We rephrased the statement.

l. 381: "According to our correlation analyses between the two mass balance estimates and different reanalysis products, in Pamir and Tien Shan, a clear identification of glacier evolution dominant drivers is impossible."

―――――

*L346: "at the local scale. . . " Please check the text for many of these small grammatical issues or minor mistakes.*

We adjusted this accordingly.

———

*L346-8: This sentence does not make sense to me. Please rephrase to make your point clear.*

We adjusted the entire paragraph to account also for the next three comments.

l.380:"Our temporal and spatial analyses (Section Spatial correlation) and Section Temporal correlation) offer radically different conclusions on the climatic drivers behind glacier mass balance response, independently of the mass balance estimate used or time period considered. According to our correlation analyses between the two mass balance estimates and different reanalysis products, in Pamir and Tien Shan, a clear identification of glacier evolution dominant drivers is impossible. Regional subdivision can influence significantly the outcome of a spatial analysis (Section Spatial correlation), and aggregation or division of the total number of glaciers into regional clusters for comparison, although often required to summarize extensive study results, is somehow arbitrary (e.g. Brun et al., 2019). Use of temporal analysis based on annual mass balance time series can improve both the subregions definition and the aggregation process; however, this analysis depends strongly on the accuracy of the mass balance estimates at annual time scales, which are often highly uncertain, not observation based, or simply not available. In the present study, we discuss the uncertainty and danger of misinterpretations related to correlation analysis relying on individual data products."

———

*L350: but they do not prevent this analysis here, the authors have an annual mass balance value for temporal analysis. I think this first paragraph needs to be restructured slightly to highlight what the authors are able to show, which is normally not considered/available.*

This comment is not fully clear to us. The temporal analysis provides us with the means to avoid any arbitrary aggregation and instead look at each glacier individually. We rephrased the entire paragraph, so that it becomes more clear to the reader (see comment above).

———

*L369-379: I think the authors should rewrite this segment as their point is not particularly clear upon reading it. Do the authors mean that their HiMAP regions encompass multiple sub-climatic regions and associated atmospheric processes which therefore limits the usefulness of these defined regions for such analyses?*

There are various points addressed that encompass spatial and temporal aspects. But yes, this is often the problem when using standard regional divisions for correlation analysis. Apart from this, the heterogeneity of glacier response to changing atmospheric settings that again leads to changes in sensitivity can create a very complex picture within one subregion. This is then difficult to relate to the reanalysis datasets. We tried to be clearer with our statement and rewrite and restructure the section.

l.407ff: "When dividing glaciers regionally, the amount of significant correlations between T, P, SC, and mass balance decreases, and the identified variables are shown to depend strongly on the datasets used (Fig. 7 and 8).

The low number of significant correlations for certain subregions can respond to (1) data availability, (2) data quality and scale issues, and (3) process representation.

(1) Due to the lower number of glaciers with mass balance time series available, especially for Eastern Pamir and Dzhungarsky Alatau, the robustness of the statistical analysis is reduced, making difficult to find a significant correlation with the meteorological variables. (2) The meteorological variables used cannot represent on their own the relevant processes responsible for the mass balance variability. For example, at high elevations, the relevance of temperature and long-wave radiation can decrease in favor of short wave radiation, which could likewise explain lower numbers of significant correlations for T in Eastern Pamir, where the highest glaciers are located (Sicart et al., 2011; Yang et al., 2011). The larger changes in meteorological forcing needed for a glacier response in areas with lower mass balance sensitivity (regions with more continental climate regimes) are easier to capture in reanalysis than the small changes that can make glaciers react in areas with higher sensitivity (subcontinental regions). For example, small scale processes (e.g. changes in pore space) close to the equilibrium line altitude can change accumulation patterns to which such glaciers react sensitively (Kronenberg et al., 2022). This can explain the lower number of significant correlations found for more subcontinental settings. Such scale-related issues can also explain some of the contrasting patterns in the correlation analysis resulting from the different reanalysis products due to their different spatial resolution. (3) Changing mass balance variability, sensitivity, response times, or mass balance gradients and regimes can further complicate finding consistent relationships under climate change at a subregional scale. Based on station data, Wang et al. (2019) showed that subcontinental glaciers react mostly to air temperature variability, whereas continental glaciers, generally located at higher elevations, are sensitive to both air temperature and precipitation. Under ongoing climate change, glacier response to meteorological conditions undergo important changes due to changing mass balance sensitivity and variability (Azisov et al., accepted; Dyurgerov and Dwyer, 2000), partly related to a shift from continental to more subcontinental conditions, thus shifting the dependence on specific meteorological variables. Changes in mass balance gradient such as rotation (simultaneous increase of ablation with decreasing elevation and increase of accumulation with increasing elevation or vice-versa) or parallel shifts of the mass balance gradients (simultaneous increased ablation and decreased accumulation or vice-versa) have been observed for the region (Kronenberg et al., 2022; Dyurgerov and Dwyer, 2000; Azisov et al., accepted; Kuhn, 1980, 1984). Such shifts, in return, influence the mass balance sensitivity (Wang et al., 2017) and variability (Barandun et al., 2021)."

————

*L382: Please define what "rotation" of mass balance gradients means.*

We explained this better.

l.430: "Changes in mass balance gradient such as rotation (simultaneous increase of ablation with decreasing elevation and increase of accumulation with increasing elevation or vice-versa) or parallel shifts of the mass balance gradients (simultaneous increased ablation and decreased accumulation or vice-versa) have been observed for the region (Kronenberg et al., 2022; Dyurgerov and Dwyer, 2000; Azisov et al., accepted; Kuhn, 1980, 1984).."

————

*L385: reword "already slight uncertainties..." It does not make sense. Small uncertainties? What estimates? Which variables?*

We reworked the entire paragraph as outlined above and the statement has been deleted.
* * *
*L387: How "invisible"? Are simply not resolved by the coarse reanalysis grid scale?*

This is indeed what we meant to say. We clarified and reworked the entire paragraph (see comment above).
* * *
*L388-9: Not clear. Do the authors suggest that coarse grid scale products can capture changes in these regions because there is less sub-grid variability in topography and its associated meteorological complexity?*

We meant that due to the different mass balance sensitivities of glaciers in continental or more maritime settings, their reaction to a change will be different in amplitude. For glacier at high altitudes and more continental climate regimes, with lower mass balance sensitivity, larger changes are needed for a glacier response. These more important changes are probably easier to capture in reanalysis datasets than small scale changes and might explain the higher number of significant correlations found for more continental settings.

However for glaciers in sub-continental regions, where sensitivity is highest, small variations at various spatial scales over a glacier might remain unresolved in the coarsely resolved reanalysis products, leading to the lower correlation. This adds to the heterogeneous mass balance response and renders finding direct correlations with climatic drivers difficult.

We try to be more clear and better structured with the argumentation within the entire paragraph (see changes above).
* * *
*L392: What do the authors mean with "outer orogene margins"?*

We adjusted this. The "outer" was not necessary.
* * *
*L399: "remains unclear..."*

We adjusted this accordingly.
* * *
*L406: please provide examples of which physical snow properties that the authors refer to and what variables missing from the analysis might influence the results.*

We adjusted this accordingly.

l.445: "For these cases, glacier response might be more related to radiation, physical snow properties (e.g. albedo, grain size), or the amount of snow, which in total are better represented by simply using T and P than with the qualitative information of SC."

———————

*L410: "extent".*

We adjusted this accordingly.

———————

*L434: Citation requires parenthesis, check formatting.*

We adjusted this accordingly.

———————

*L489: Check formatting.*

We adjusted this accordingly.

———————

*L497-8: Related to the general comments, 14 years does not produce an ideal correlation period. Some discussion on this is needed.*

We added more discussion on this as explained also in an earlier comment.

———————

*LL503: The authors cannot evaluate the biases and patterns of reanalysis datasets due to lack of ground data, but should still highlight how the products relate to each other. Although long term in situ observations are rare, and I understand that it is not the goal of the manuscript to identify the 'best' product to use, but I would be interested to see how the different products compare to even short term measurements from sporadic AWS measurements familiar to the authors from previous work (e.g. Abramov Glacier , Golubin Glacier).*

We now include the dataset comparison. The comparison with AWS or meteorological station data in general comes with a lot of culprits that we do not want to address in this paper. At least for the HAR dataset there is a comparison available in Pohl et al. (2015). For Abramov

there is already the problem that precipitation is not measured by the modern AWS (Kronenberg et al., 2021) and data gaps are very common at other sites as well. We think that trying to find a suitable method for a sound comparison would - although interesting - not change the conclusions in the present work. We hope this is acceptable for the reviewer. We linked the two references mentioned before and included references from other sites where there has been an assessment made. Either way, a good or a bad correlation should only locally increase our trust in any of the products but not over the whole study domain.

l.35ff: "Meteorological measurements are sparse and often discontinuous even for the most monitored glaciers in Central Asia, such as Abramov or Golubin Glacier (Kronenberg et al., 2021; Azisov et al., accepted). Replacement of old meteorological stations with modern sensors often lacks precise homogenization. Regional extrapolation from station data and use of existing time series as validation datasets for gridded products are thus problematic."

and

l.288ff: "A comparison of the different reanalysis datasets is shown for precipitation in Figure 3 and for temperature in Figure A2. Only CHELSA provides a spatial resolution to resolve intramountain range variability. Differences in precipitation amount and different trends exist independently of the spatial resolution (Fig. 3). However, for temperature, CHELSA is the temperature-conserving downscaled ERA5 product with matching trends and a correlation of one (Fig. A2i). For precipitation, CHELSA and ERA5 show deviating distribution patterns and trends, highlighting the nonlinear downscaling method (Fig. 3). The use of the datasets in their original spatial resolution in consecutive analyses allow for an assessment of how downscaling affects the correlation with mass balance data."

————

Figures *Figure 1: The inset map in the upper right panel should ideally be of High Mountain Asia to provide a better spatial context as to where the study site is. If able, the authors should attempt to neaten the location and intersect of the legend numbering as there is overlap in places.*

We included a HMA inset map and neaten the legend.

————

*Figure 4: Please clarify in the caption how variable importance is defined and why this is given as a relative freq. in the plot labels. Also ideally provide indices (a,b,c etc) to aid navigation for the reader from the main text. I don't find a 'short' analysis period as meaningfully different to the long one for the lower panels. Such morphological characteristics (e.g. debris) won't greatly change over 4 years. Consider removing it to streamline and simplify the figure. In general I find it very hard to interpret the information given in these figures (4,5,6, A1, A2). Any attempt to simplify it or remove parts would aid interpretation.*

We use the term "Relative Importance" which at this point is only explained in the methods. We included this in a clearer way in the second paragraph of the methods section. We added this information also in the caption of the workflow overview with a simpler wording and referenced

it to the main text to aid understanding the graphs. The words "short" and "long" always refer to the length of the meteorological respectively glacier mass balance data time series. The explanation, including the exact length of the periods is included now in the Data section and stated in the figure captions.

Fig.4 caption: "[...] All significant correlations of a variable are used to produce the maps for the temporal analyses, and the reported "Relative Importance" in the spatial analysis is defined as the number of significant variables over the total number of tested variable combinations (see also Section Methods).

Fig.6 caption: "[...] Labels "short" and "long" refer to the periods 2000–2014 and 2000–2018, respectively. For the morphological analysis this refers to the glacier mass balance data.

————

*Figure 5: I think this figure has far too much information and becomes difficult to interpret. Please consider streamlining the most crucial information where possible, perhaps just showing the short periods, or those with MOD10. For upper panels, one has to struggle to align the x axis label and interpret what is the take-home information. legends should be increased in size, but could easily be kept for just one panel for the whole left and right side.*

This is a difficult point. We agree that this is a lot of information provided in the figure. However, we also think it is well suited to show the differences per sub-regions, the changing variable importance with and without snow cover, and between the two considered time periods. As we take reference to these figures, we worked on improving the readability by adding helping lines and panel labels.

————

*Figure 6: Similar considerations to the above.*

We addressed this in the same way as pointed out for the previous comment.

————

*Figure 7: These plots are more clear than the previous ones. Perhaps the authors can re-structure the panels to be 3 rows and two columns, so that the size on the page can be enhanced and the reader can more easily interpret the small circles. Why using 'M' in front of the meteorological datasets? I think it would be clearer without. I do not understand from the figure caption the "with or without snow cover. . . " as all show scf as a variable importance coloured on the map. Please clarify or correct this. Please use indices for the subplots to help the reader. For Figure 8 as well.*

We homogenized the different abbreviations and write out the data combinations (M for MOD10) to avoid this confusion. We tried to optimize the figure readability by the proposed 3 x 2 format

to increase the individual panel sizes.

———